# Personalized Collaborative Learning with Affinity-Based Variance Reduction

**Chenyu Zhang**
Massachusetts Institute of Technology
`zcysxy@mit.edu`

**Navid Azizan**
Massachusetts Institute of Technology
`azizan@mit.edu`

## Abstract

Multi-agent learning faces a fundamental tension: leveraging distributed collaboration without sacrificing the personalization needed for diverse agents. This tension intensifies when aiming for full personalization while adapting to unknown heterogeneity levels—gaining collaborative speedup when agents are similar, without performance degradation when they are different. Embracing the challenge, we propose personalized collaborative learning (PCL), a novel framework for heterogeneous agents to collaboratively learn personalized solutions with seamless adaptivity. Through carefully designed bias correction and importance correction mechanisms, our method AffPCL robustly handles both environment and objective heterogeneity. We prove that AffPCL reduces sample complexity over independent learning by a factor of $\max\{n^{-1}, \delta\}$, where $n$ is the number of agents and $\delta \in [0, 1]$ measures their heterogeneity. This *affinity-based* acceleration automatically interpolates between the linear speedup of federated learning in homogeneous settings and the baseline of independent learning, without requiring prior knowledge of the system. Our analysis further reveals that an agent may obtain linear speedup even by collaborating with arbitrarily dissimilar agents, unveiling new insights into personalization and collaboration in the high heterogeneity regime.

## 1 Introduction

Heterogeneity is a defining yet formidable characteristic of multi-agent systems. When agents differ significantly, their incentives to collaborate diminish, as leveraging experience from others can introduce bias and impede their own learning. This challenge intensifies in scenarios where strategic agents seek highly accurate, tailored solutions. Collaborative multi-agent systems commonly adopt a federated learning (FL) setup, where agents communicate via a central server to jointly learn a unified solution. However, in the presence of heterogeneity, such unified solutions often prove suboptimal or even irrelevant for individual agents. Consequently, effective personalization becomes essential for collaborative learning among heterogeneous agents.

This need is evident in real-world applications: personalized recommendations drive user engagement (Good et al., 1999; Anand & Mobasher, 2003; Khribi et al., 2008), autonomous transportation must accommodate local traffic conditions (Huang et al., 2021; You et al., 2024; Jalota et al., 2023), robots need to adapt to different operating environments (Richards et al., 2023; Tang et al., 2025), diverse patient profiles require tailored treatments (Chen et al., 2022; Tang et al., 2024), and agentic language models need to adapt to specific user styles and task contexts (Li et al., 2024; Woźniak et al., 2024; Bose et al., 2025; Nobari et al., 2025).

These considerations motivate the following multi-agent decision-making setup.

1. **Personalized**. Agents are intrinsically heterogeneous, each with arbitrarily distinct environments and objectives, and act strategically to optimize their own goals.
2. **Collaborative**. Agents communicate through a central server that aggregates information from agents and broadcasts back the aggregated result.
3. **Learning**. Agents have no prior knowledge of their systems and interact only with local environments that generate stochastic observations of system parameters.

Such a complex, stochastic, and heterogeneous multi-agent system demands, but also challenges, the design of a *personalized collaborative learning* algorithm that can (1) find fully personalized solutions for all agents, (2) achieve performance gains through collaboration, (3) and adapt to unknown heterogeneity among agents without prior knowledge, automatically harnessing greater collaboration benefits when agents are similar, and falling back to, while ensuring no worse performance than, non-collaborative independent learning when agents are markedly different.

This work reveals that the key to achieving these goals lies in identifying and exploiting *affinity*, i.e., similarity among agents. Formally, we capture agent heterogeneity through a score $\delta \in [0, 1]$, with $\delta = 0$ indicating homogeneous agents and larger values of $\delta$ indicating greater heterogeneity. For any agent, our method finds its personalized solution with a mean squared error of

$$O(t^{-1} \cdot \max\{n^{-1}, \delta\}), \tag{1}$$

where $t$ is the number of samples collected by each agent and $n$ is the number of agents. This finite-sample complexity enjoys federated speedup linear in $n$ when agents are similar, while it gracefully reduces to the baseline rate of independent learning $O(t^{-1})$ when agents are highly heterogeneous, but never worse. In intermediate regimes, *affinity-based* acceleration manifests.

We summarize our main contributions:

1. We formulate a novel multi-agent decision-making paradigm of personalized collaborative learning (PCL), encompassing applications and problems in supervised learning, reinforcement learning (Appendix C.6), and statistical decision-making.
2. We develop a simple yet effective method that realizes the vision of PCL, called AffPCL, which finds fully personalized solutions and adaptively harnesses collaboration benefits when agents are similar while ensuring no worse performance than independent learning when they are highly heterogeneous. Our method robustly handles arbitrary objective and environment heterogeneity through principled personalized bias correction and importance correction mechanisms.
3. We establish finite-sample convergence guarantees for AffPCL, achieving the rate in (1) and thus demonstrating the desired phenomenon of *affinity-based variance reduction*. This rate adaptively interpolates between the linear speedup of FL and the minimax optimal rate of independent learning. This is the first result that proves efficiency gains for learning fully personalized solutions through collaboration among arbitrarily heterogeneous agents.
4. We further enhance AffPCL with features including asynchronous importance estimation and agent-specific update schemes. Our agent-specific analysis reveals that an agent may achieve linear speedup even when it is dissimilar to all others, a phenomenon unattainable in prior frameworks.

## 1.1 RELATED WORK

We focus on the most relevant works in heterogeneous collaborative learning that motivate this study.

**Personalization falls short in federated learning.** Classical FL methods (McMahan et al., 2017) aim for a unified solution for all agents without personalization guarantees. The unified objective mitigates heterogeneity; for instance, bias correction in heterogeneous FL (Karimireddy et al., 2020b; Guo et al., 2023; Karimireddy et al., 2020a; Gao et al., 2022), which prevents local updates from drifting away from the central update direction, is averaged across all agents and thus enjoys federated variance reduction (see also Section 2). In contrast, personalization requires adjusting the central update relative to each agent's unique local direction, which precludes federated variance reduction.

The growing literature on (partially) personalized FL highlights the importance of personalization. A common strategy combines global and local models through regularization or mixtures (Li et al., 2020; Hanzely & Richtárik, 2020; T Dinh et al., 2020; Li et al., 2021; Deng et al., 2020), but such methods offer only *partial* personalization and the trade-offs may be heuristic. Similarly, clustering-based methods (Sattler et al., 2020; Mansour et al., 2020; Ghosh et al., 2020; Briggs et al., 2020; Chai et al., 2020; Grimberg et al., 2021) do not offer personalization within each cluster and may be sensitive to hyperparameter tuning or prior knowledge. In contrast, PCL aims for *full* personalization and seamless adaptivity, requiring neither prior knowledge of heterogeneity nor hyperparameter tuning.

**Slower rates in independent learning.** Other personalized learning approaches combine FL and independent learning. A sequential strategy uses FL as a warm start followed by independent fine-tuning (Fallah et al., 2020; Cheng et al., 2021); while effective in some cases, this approach is

generally rate-suboptimal, as the small initialization error through FL diminishes faster than the variance from independent learning, making its change in finite-time complexity marginal. A parallel approach simultaneously learns a shared global component and a personalized local component (Pillutla et al., 2022; Xiong et al., 2025; Liang et al., 2020); this approach requires certain global-local structures, and similarly, the independent learning component dominates the overall complexity, obscuring collaborative speedup. In contrast, PCL imposes no structural assumptions, accommodates arbitrarily heterogeneous agents, and aims for provably faster rates than independent learning.

**Curse of heterogeneity in collaborative learning.** Closest to our setup, Chayti et al. (2021); Even et al. (2022) also study full personalization with arbitrarily heterogeneous systems, but with fundamentally different approaches from ours in handling heterogeneity to achieve collaborative variance reduction. First, they selectively collaborate with similar agents, effectively reducing to clustering-based methods or low heterogeneity regimes, whereas AffPCL enables collaboration among all agents regardless of similarity. With AffPCL, an agent may attain linear speedup even when it's not similar to any other agent (Section 6.3), which is unattainable in their frameworks. Second, achieving optimal speedup in their setting requires either knowledge of objective heterogeneity (Even et al., 2022) or access to a bias estimation oracle whose variance reduces linearly in the number of agents (Chayti et al., 2021), which is a strong assumption as bias estimation for personalization is inherently agent-specific, and its variance does not reduce with more agents. In contrast, AffPCL requires no prior knowledge or bias estimation oracle, and enjoys affinity-based variance reduction fully adaptively.

## 1.2 PROBLEM FORMULATION

We consider a general multi-agent linear system:
$$\bar{A}^i x_*^i = \bar{b}^i, \quad i = 1, \ldots, n, \tag{2}$$
where $\text{sym}(\bar{A}^i) = \frac{1}{2}(\bar{A}^i + (\bar{A}^i)^T) \succ 0$. Each agent aims to find the fixed point $x_*^i$ of its system with access to only stochastic observations $A(s_t^i) \in \mathbb{R}^{d \times d}$ and $b^i(s_t^i) \in \mathbb{R}^d$ evaluated at its local random state $s_t^i \in \mathcal{S}$ independently sampled from its distinct environment distribution $\mu^i \in \Delta(\mathcal{S})$ at time step $t$. The stochastic observations are unbiased such that $\bar{A}^i = \mathbb{E}_{\mu^i} A(s^i)$ and $\bar{b}^i = \mathbb{E}_{\mu^i} b^i(s^i)$.

**Terminology and notation.** Our system modeling draws inspiration from various fields, including supervised learning, reinforcement learning, and statistical decision-making, where $(A, b, \mu)$ are commonly referred to as (feature, label, covariate distribution), (function approximation, reward, stationary distribution), and (measurement, response, data distribution), respectively. To appeal to a broader audience and align with our setup, we refer to $A$ as the *feature* embedding matrix, $b$ as the *objective* vector, and $\mu$ as the *environment* distribution. As is common in practice, we assume that all agents share the same feature extractor $A$, but may have different objectives $b^i$ and environments $\mu^i$, referred to as *objective heterogeneity* and *environment heterogeneity*, respectively.

Throughout the paper, superscript $i$ denotes quantities related to agent $i$ and superscript $0$ denotes the *averaged* quantity across all agents, i.e., $f^0 = \frac{1}{n} \sum_{i=1}^n f^i$ for any quantity $f$. The averaged quantity may be explicitly aggregated by the central server, or it can represent a virtual quantity only used for analysis. We write $[n] := \{1, \ldots, n\}$. For any function $f^i$ on $\mathcal{S}$, $\bar{f}^i$ denotes the expectation of $f^i$ under the corresponding environment distribution $\mu^i$, i.e., $\bar{f}^i = \mathbb{E}_{\mu^i} f^i(s^i)$. For an unknown quantity $f$, its estimate learned at time step $t$ is denoted by $\hat{f}_t$. The default norm is the Euclidean norm for vectors, operator norm for matrices, and total variation norm for distribution differences. Appendix A contains a complete list of notation.

**Roadmap.** This paper adopts a progressive approach to first develop insights in stylized settings and then incrementally extend to more complex scenarios. We start with a simplified FL setup (Section 2), then gradually introduce personalization (Section 3), adaptivity (Section 4), environment heterogeneity (Section 5.1), and finally arrive at the most general setup (2) in Section 5.2. Several theoretical extensions are discussed in Section 6 and numerical results are presented in Section 7.

## 2 WARM-UP: HETEROGENEOUS FEDERATED LEARNING

We start by reviewing heterogeneous FL, a variant of (2) where agents with distinct objectives collaborate to find a unified solution $x_*^c$ satisfying
$$\bar{A}^0 x_*^c = \bar{b}^0, \tag{3}$$

where $\bar{A}^0 = \frac{1}{n}\sum_{i=1}^n \mathbb{E}_{\mu^i}[A(s)]$ and $\bar{b}^0 = \frac{1}{n}\sum_{i=1}^n \mathbb{E}_{\mu^i}[b^i(s)]$.[1] This warm-up section assumes homogeneous environment distributions $\mu^i \equiv \mu$ for all $i \in [n]$, and thus we can drop the superscript of $\bar{A}$. Since $\mathrm{sym}(\bar{A})$ is positive definite, in a federated stochastic approximation setting, each agent adopts the following fixed-point iteration:

$$x_{t+1}^i = x_t^i - \alpha_t g_t^i(x_t^i), \quad \text{where} \quad g_t^i(x_t^i) := A(s_t^i)x_t^i - b(s_t^i),$$

where $\alpha_t$ is the step size and the update direction $g_t^i$ is the stochastic residual at time step $t$.

To focus on the main ideas, this work considers a simplified communication scheme, where agents communicate with a central server at every time step. In FL, agents send their local updates to the server, which aggregates them to get the central decision variable $x_{t+1}^c$ and broadcasts it back $x_{t+1}^i \leftarrow x_{t+1}^c$. The resultant central update rule is then given by

$$x_{t+1}^c = x_t^c - \alpha_t g_t^0(x_t^c), \text{ where } g_t^0(x_t^c) := \frac{1}{n}\sum_{i=1}^n g_t^i(x_t^c) = \frac{1}{n}\sum_{i=1}^n A(s_t^i)x_t^c - \frac{1}{n}\sum_{i=1}^n b^i(s_t^i). \quad (4)$$

We note that in this FL setting, the local decision variables are always synced with the central one, and thus we have $g_t^i(x_t^i) = g_t^i(x_t^c)$. Moreover, we can write the central decision variable as the average of the local ones: $x_t^c = \frac{1}{n}\sum_{i=1}^n x_t^i = x_t^0$. However, this equivalence becomes obsolete when we introduce heterogeneous environments and personalization.

**Constants.** We define the following constants used throughout. $\lambda := \min_i \lambda_{\min}(\mathrm{sym}(\bar{A}^i)) > 0$ ensures strong monotonicity of the fixed-point iteration and controls the convergence rate; an analogous condition in optimization is $\lambda$-strong convexity or $\lambda$-PL condition of the objective function (Nesterov, 2013). $G_A := \max_i \sup_s \|A^i(s)\|$, $G_b := \max_i \sup_s \|b^i(s)\|$, and $G_x := \max_i \|x_*^i\|$ upper bound the system parameters. Let $\sigma := 2\max\{G_A G_x, G_b\}$ represent the scale of the system, which can also be thought of as the variance proxy of the update direction at the solution point, since $\|g_t^i(x_*^i)\| \le \|A(s_t^i)\|\|x_*^i\| + \|b^i(s_t^i)\| \le G_A G_x + G_b \le \sigma$; its analogy in optimization is the objective function gradient's Lipschitz constant. We then define $\kappa := \sigma/\lambda$ as the condition number of the stochastic system. Without loss of generality, we use 1 as the variance proxy of the environment distributions, in the sense that $\mathrm{tr}\,\mathrm{Var}_\mu(f(s)) = \mathbb{E}_\mu\|f(s)\|^2 \le G_f^2$, which holds for any zero-mean operator $f$ with $\mathrm{ess\,sup}_{s\sim\mu}\|f(s)\| \le G_f$.

We have the following convergence guarantee for heterogeneous FL.[2]

**Proposition 1.** *With a constant step size $\alpha \equiv \ln t/(\lambda t)$, (4) satisfies*

$$\mathbb{E}\|x_t^c - x_*^c\|^2 = \widetilde{O}(\kappa^2 t^{-1} n^{-1}),$$

*where $\widetilde{O}$ suppresses the logarithmic dependence on $\ln t$.*[3]

The mean squared error (MSE) of FL vanishes linearly as $t$ goes to infinity, with the rate scaled by the problem scale $\sigma$ and controlled by $\lambda$. The federated collaboration contributes linear speedup in terms of the number of agents $n$. Proposition 1 is tight in $\kappa$, $t$, and $n$ (Woodworth et al., 2020; Karimireddy et al., 2020b; Glasgow et al., 2022), and serves as a baseline for our subsequent results.

## 3 Introducing personalization: Personalized bias correction

Due to heterogeneity, the unified solution described in Section 2 is generally suboptimal for individual agents, and becomes less relevant as the heterogeneity level grows. More realistically, strategic agents seek *personalized* solutions:

$$\bar{A}x_*^i = \bar{b}^i, \quad i \in [n].$$

---

[1] The unified-solution setting is also related to distributed linear-system solvers (Azizan-Ruhi et al., 2019; Velasevic et al., 2023), but those assume that the equations are partitioned across agents.

[2] All proofs are deferred to Appendices E to G, where we progressively establish the main result Theorem 1 and cover all the propositions in the main text.

[3] The $\ln t$ dependence can be removed by using a linearly diminishing step size and considering a convex combination of the iterates $\{x_\tau^c\}_{\tau=0}^t$, as specified in Lemma D.5. This refinement applies to all results in the main text. For brevity, we defer the related discussion to the appendix and omit this remark in subsequent results.

To build intuition, this section makes two simplifications to be relaxed in the next two sections: agents have the same environment distribution, and the central objective $b^0(s_t^i) = \frac{1}{n}\sum_{i=1}^n b^i(s_t^i)$ is known to agent $i$ upon observing $s_t^i$. With access to the central objective, we propose affinity-aware personalized collaborative learning (AffPCL), a simple yet effective update rule for each agent:

$$x_{t+1}^i = x_t^i - \alpha_t \tilde{g}_t^i, \quad \text{where} \quad \tilde{g}_t^i = g_t^i(x_t^i) + g_t^0(x_t^0) - g_t^{0\to i}(x_t^0), \tag{5}$$

where the update direction consists of three components:

$$g_t^i(x_t^i) = A(s_t^i)x_t^i - b^i(s_t^i), \quad g_t^0(x_t^0) = \frac{1}{n}\sum_{i=1}^n g_t^i(x_t^0), \quad g_t^{0\to i}(x_t^0) = A(s_t^i)x_t^0 - b^0(s_t^i).$$

Recall that $x_t^0 = \frac{1}{n}\sum_{i=1}^n x_t^i$ is synced with the central server. Alternatively, inspired by Section 2, we can replace $x_t^0$ with an explicitly maintained central decision variable $x_t^c$ and update it using (4) within the same communication round for computing the central update direction. Both implementations have the same convergence guarantee in current setting, while the latter proves robust to heterogeneous environment distributions, as detailed in Section 5.2. See Appendix C.1 for further discussion.

Unlike FL, the convergence of AffPCL depends on how *similar* the objectives of agents are.

**Definition 1** (Objective heterogeneity). The objective heterogeneity level is defined as

$$\delta_{\text{obj}} := \max_{i,j\in[n]} \sup_{s\in\mathcal{S}} \|b^i(s) - b^j(s)\|_2/(2G_b) \in [0,1].$$

**Proposition 2.** *With a constant step size $\alpha \equiv \ln t/(\lambda t)$, (5) satisfies*

$$\mathbb{E}\|x_t^i - x_*^i\|^2 = \widetilde{O}(\kappa^2 t^{-1}\cdot\max\{n^{-1},\tilde{\delta}_{\text{obj}}\}), \quad \forall i\in[n],$$

*where $\tilde{\delta}_{\text{obj}} \leq \min\{1,\kappa\delta_{\text{obj}}\}$ is the* effective *objective heterogeneity level.*

The precise definition of the effective heterogeneity level is $\tilde{\delta}_{\text{obj}} = \min\{1,\nu\delta_{\text{obj}}\}$, where $\nu$ is the *stochastic condition number* that is trivially bounded by $\kappa$. We defer the definition of $\nu$ and the discussion of how the stochastic conditioning affects the effective collaboration gain to Section 6.2 and C.5. In most cases of interest, $\nu$ is close to 1, reducing $\tilde{\delta}_{\text{obj}}$ to the raw heterogeneity level $\delta_{\text{obj}}$. Thus, the following discussion of $\tilde{\delta}_{\text{obj}}$ can be understood as applying to $\delta_{\text{obj}}$ as well.

Proposition 2 previews the phenomenon of *affinity-based variance reduction*. Compared to independent learning, Proposition 2 achieves a convergence rate accelerated by a factor of $\max\{n^{-1},\tilde{\delta}_{\text{obj}}\}$, capturing speedup from both federated collaboration and agent similarity. When agents have similar objectives ($\tilde{\delta}_{\text{obj}} \leq n^{-1}$), this factor recovers the linear speedup $n^{-1}$ from FL (Proposition 1); with abundant collaborating agents ($n \geq \tilde{\delta}_{\text{obj}}^{-1}$), objective affinity dominates variance reduction. Importantly, since $\tilde{\delta}_{\text{obj}} \in [0,1]$, AffPCL's worst-case complexity is always upper bounded by that of independent learning, $\widetilde{O}(\kappa^2 t^{-1})$, ensuring collaboration never degrades performance. As agents have markedly different objectives ($\tilde{\delta}_{\text{obj}} \uparrow 1$), collaboration benefits vanish and AffPCL falls back to independent learning, as expected. Proposition 2 showcases that AffPCL seamlessly interpolates between FL and independent learning, offering full adaptivity without imposing artificial restrictions.

To provide intuitions for affinity-based variance reduction, we discuss three interpretations of AffPCL.

**Bias correction.** $g_t^i(x_t^i) - g_t^{0\to i}(x_t^0)$ in (5) corrects the bias in the aggregated update direction $g_t^0(x_t^0)$ to achieve personalization. Specifically, one can verify that $\mathbb{E}_\mu[\tilde{g}_t^i] = \mathbb{E}_\mu[g_t^i(x_t^i)]$. In collaborative learning, agents want to leverage the aggregated update direction for its lower variance, but also need to correct its bias towards the central solution rather than the personalized solution. We remark that this bias correction is fundamentally different from those used in the heterogeneous FL literature (Karimireddy et al., 2020b; Mangold et al., 2024), which correct for local drift away from the central direction. In other words, our novel bias correction term is *personalized* for each agent.

**Control variates.** Although $g_t^i(x_t^i) - g_t^{0\to i}(x_t^0)$ is personalized and thus cannot benefit from federated averaging, it enjoys affinity-based variance reduction via control variates (Defazio et al., 2014; Rubinstein & Kroese, 2016). Specifically, $g_t^{0\to i}(x_t^0)$ serves as a control variate that positively correlates with the local update direction $g_t^i(x_t^i)$, and thus reduces the variance of the overall update. Then, a low-variance version (sample average) of the control variate, $g_t^0(x_t^0)$, is added to correct the introduced

bias. The variance reduction effect scales with the correlation in the control variate, which in turn scales with the affinity between the local and central systems. This control variate perspective offers a clear explanation for affinity-based variance reduction in AffPCL, and motivates our design choice of the bias correction term $g_t^{0 \to i}(x_t^0)$, which correlates with $g_t^i(x_t^i)$ through the underlying sample $s_t^i$, unlike other potential candidates (e.g., $A_t^i x_t^0 - b_t^0$) that would be nearly independent of $s_t^i$.

**Central-local decomposition.** To perceive how this variance reduction scales with affinity, we can view (5) as performing *central* and *local* learning in parallel. The central learning happens at the server side, seeking a unified point $x_*^{\mathrm{cen}}$ that solves the *central system* (3), and the local learning happens at the client side, solving the *local residual system* $\bar{A} x_*^{i,loc} = \bar{b}^i - \bar{b}^0$. Then, $x_*^{\mathrm{cen}} + x_*^{i,loc}$ gives the personalized solution to (2). Specifically, $g_t^0(x_t^0) = A_t^0 x_t^0 - b_t^0$ drives the central learning and $g_t^i(x_t^i) - g_t^{0 \to i}(x_t^0) = A_t^i(x_t^i - x_t^0) - (b^i - b^0)(s_t^i)$ drives the local learning. Intuitively, the local residual system is simpler to solve when agent's objective is close to the central one, leading to affinity-based variance reduction.Identifying this low-complexity local residual system is key to AffPCL's success. If the local learning problem were as complex as the original one (e.g., fine-tuning), such a central-local decomposition would offer only marginal speedup over independent learning.

## 4 INTRODUCING ADAPTIVITY: CENTRAL OBJECTIVE ESTIMATION

Knowing the central objective amounts to knowing other agents' objectives, which may not be realistic in practice. This section removes this assumption by enabling agents to *adaptively* learn the central objective while learning their personalized solutions. A practical challenge is that when the state space $\mathcal{S}$ is large or infinite, $b^0$ becomes high- or infinite-dimensional, and learning it inevitably dominates the overall complexity. To match the dimension of other system parameters, we consider a linear parametrization of the objective function: $b^i(s) = \Phi(s)\theta_*^i$ for all $i \in [n]$, where $\Phi \in \mathbb{R}^{d \times d}$ is a feature embedding function such that $\mathrm{sym}(\mathbb{E}_\mu \Phi(s)) \succ 0$, and $\theta_*^i \in \mathbb{R}^d$ is the weight. This structure covers finite state spaces as a special case; see Appendix C.2 for more discussion.

Interestingly, central objective estimation (COE) is a special case of heterogeneous FL in Section 2:
$$\bar{\Phi}^0 \theta_*^c = \bar{b}^0, \tag{6}$$
where $\bar{\Phi}^0 = \mathbb{E}_\mu \Phi(s)$ and $\bar{b}^0 = \frac{1}{n}\sum_{i=1}^n \mathbb{E}_\mu b^i(s) = \mathbb{E}_\mu b^0(s)$. Therefore, agents can federatedly estimate the central objective using the same algorithm in Section 2:
$$\theta_{t+1}^c = \theta_t^c - \alpha_t g_t^{0,b}(\theta_t^c), \text{ where } g_t^{0,b}(\theta_t^c) := \frac{1}{n}\sum_{i=1}^n \Phi(s_t^i)\theta_t^c - \frac{1}{n}\sum_{i=1}^n b^i(s_t^i). \tag{7}$$

Without loss of generality, we use normalized features $\|\Phi(s)\|_2 \leq 1$ for all $s \in \mathcal{S}$. With linear parametrization, we redefine the objective bound $G_b := \max\{\max_i \|\theta_*^i\|, \|\theta_*^c\|\}$ and heterogeneity level $\delta_{\mathrm{obj}} := \max_{i,j \in [n]} \|\theta_*^i - \theta_*^j\|_2/(2G_b) \in [0,1]$, which imply the original bound and Definition 1. Then, COE directly enjoys the same guarantee as in Proposition 1, with $\lambda$ replaced by $\lambda_{\min}(\mathrm{sym}(\bar{\Phi}^0))$.

We denote $\hat{b}_t^0 := \Phi(s)\theta_t^c$ as the estimated central objective at time $t$. Then, agents use $\hat{b}_t^0(s_t^i)$ in place of $b^0(s_t^i)$ in AffPCL (5), asynchronously with COE in (7). This scheme enjoys the same convergence guarantee as in Proposition 2 as proven in Appendix G.

## 5 INTRODUCING ENVIRONMENT HETEROGENEITY: IMPORTANCE CORRECTION

### 5.1 CENTRAL LEARNING REVISITED

Section 3 discusses the central-local decomposition of AffPCL. With homogeneous environments, central learning happens *implicitly* by considering the dynamics of the averaged decision variable $x_t^0 = \frac{1}{n}\sum_{i=1}^n x_t^i$, which converges to the solution $x_*^c = \frac{1}{n}\sum_{i=1}^n x_*^i$ to (3). However, this is no longer true with heterogeneous environment distributions ($\mu^i \neq \mu^j$), because
$$x_*^0 = \tfrac{1}{n}\sum_{i=1}^n x_*^i = \tfrac{1}{n}\sum_{i=1}^n \left((\bar{A}^i)^{-1}\bar{b}^i\right) \neq \left(\tfrac{1}{n}\sum_{i=1}^n \bar{A}^i\right)^{-1}\left(\tfrac{1}{n}\sum_{i=1}^n \bar{b}^i\right) = (\bar{A}^0)^{-1}\bar{b}^0 = x_*^c.$$
That is, as $x_t^i$ converges to the personalized solution $x_*^i$ for all $i \in [n]$, their average will not converge to $x_*^c$, invalidating the implicit central learning through $x_t^0$.

Fortunately, we manage to show that if agents *explicitly* maintain a unified *central* decision variable $x_t^c (\not\equiv x_t^0)$ and update it federatedly using (4), then $x_t^c$ still converges to $x_*^c$ with the same convergence rate in Proposition 1, even in the presence of environment heterogeneity. We refer to this explicit approach as central decision learning (CDL). The same argument applies to COE in Section 4, i.e., (7) converges to the solution to (6) with the same rate under heterogeneous environment distributions. We defer the proof to Appendices E and F. Intuitively, this works because a sample from the mixture distribution $\mu^0$ is equivalent to first sampling an index $i$ uniformly and then sampling $s$ from $\mu^i$. Therefore, a federated update direction that equally weights local sample information from all agents is unbiased towards the central solution.

Beyond algorithmic implications, we remark that in Sections 2 and 4, the averaged decision variable naturally corresponds to a *virtual* system with parameters $\mu^0 = \frac{1}{n} \sum_{i=1}^n \mu^i$ and $b^0 = \frac{1}{n} \sum_{i=1}^n b^i$. The affinity among agents directly translates to the affinity between each agent and this "central agent" with index $0$. Environment heterogeneity perplexes this concept: who is the "central agent" now, and does it inherit the affinity among agents? Our central objective characterization in (6) helps answer the first question by defining $b^c(s) := \Phi(s)\theta_*^c$, and then the "central system" (3) corresponds to a virtual system with environment distribution $\mu^0$ and objective $b^c (\not\equiv b^0)$. Pinpointing this relocated central agent is crucial for deriving agent-specific affinity-based variance reduction in Section 6.3.

The second question is more subtle, as now the central agent, unlike the "averaged agent", can have a drastically different objective $b^c$ from all actual agents, even when the latter have similar objectives. For instance, an ill-conditioned system can amplify a small $\delta_{\text{obj}}$ (Definition 1) such that $\|b^c - b^i\|_\infty$ reaches its maximum possible value $2G_b$ for some agent $i$. This divergence, which also applies to the relationship between the central and personalized decision variables, presents a fundamental challenge introduced by environment heterogeneity in achieving affinity-based variance reduction. Fortunately, our analysis reveals that what is crucial is the affinity in *feature space*, e.g., terms like $\|A(s)(x_*^i - x_*^c)\|$ and $\|\bar{A}^i(x_*^i - x_*^c)\|$, which are well controlled by the raw affinity among actual agents. Please refer to Lemmas D.1 and D.2 in Appendix F for more discussion.

## 5.2 PCL with importance correction

We now arrive at the most general setup (2), where agents have heterogeneous environment distributions and objectives and seek personalized solutions. In addition to the challenges posed by environmental heterogeneity discussed in Section 5.1, a further obstacle emerges in the design of AffPCL: the bias correction mechanism in Section 3 alone is no longer sufficient. To overcome this, we propose integrating a novel *importance correction* to the central update direction before it gets sent to each agent, resulting in the following AffPCL update rule:

$$x_{t+1}^i = x_t^i - \alpha_t \tilde{g}_t^i, \quad \text{where} \quad \tilde{g}_t^i = g_t^i(x_t^i) + g_t^{c \rightrightarrows i}(x_t^c) - g_t^{c \rightarrow i}(x_t^c), \tag{8}$$

where the bias correction term $g_t^{c \rightarrow i}(x) = A(s_t^i)x - \hat{b}_t^c(s_t^i)$ now uses the estimated central objective $\hat{b}_t^c(s_t^i)$ from COE (7), and $g_t^{c \rightrightarrows i}$ is the importance-corrected central update direction:

$$g_t^{c \rightrightarrows i}(x) := \frac{1}{n} \sum_{j=1}^n \rho^i(s_t^j) g_t^{c \rightarrow j}(x) := \frac{1}{n} \sum_{j=1}^n \frac{\mu^i(s_t^j)}{\mu^0(s_t^j)} \left( A(s_t^j)x - \hat{b}_t^c(s_t^j) \right).$$

AffPCL (8) with asynchronous COE (7) and CDL (4) gives the complete algorithm for solving (2). We provide the pseudocode and discuss implementation details in Appendix C.1.

AffPCL effectively handles environment heterogeneity by (i) correcting bias: $\mathbb{E}[g_t^{c \rightrightarrows i}(x) - g_t^{c \rightarrow i}(x)] = 0$ (Lemma G.1), and (ii) reducing variance based on agents' *environment affinity* (Lemmas G.2 and G.3).

**Definition 2** (Environment heterogeneity)**.** The environment heterogeneity level is defined as

$$\delta_{\text{env}} := \max_{i,j \in [n]} \|\mu^i - \mu^j\|_{\text{TV}} \in [0, 1].$$

The interpretations discussed in Section 3 still account for a portion of the variance reduction effect, especially w.r.t. objective affinity. For the newly introduced importance-corrected central update direction $g_t^{c \rightrightarrows i}$, its variance has three key properties: (i) it decomposes into a federated term, an affinity-dependent term similar to the variance of the bias correction term, and a term that characterizes the environment heterogeneity: $\frac{\sigma^2}{n} \chi^2(\mu^i, \mu^0)$, where $\chi^2$ is the chi-square divergence;

(ii) the chi-square divergence is bounded by the total variation distance, which defines the environment heterogeneity: $\chi^2(\mu^i, \mu^0) \leq \max\{\|\rho^i\|_\infty, 1\}\delta_{\mathrm{env}}$; (iii) the density ratio $\rho^i$ has a natural upper bound: $\rho^i(s) = \mu^i(s)/(\frac{1}{n}\sum_{j=1}^n \mu^j(s)) \leq \mu^i(s)/(\frac{1}{n}\mu^i(s)) = n$. Combining the three observations gives an upper bound of the additional variance from environment heterogeneity: $\frac{\sigma^2}{n}\chi^2(\mu^i, \mu^0) \leq \sigma^2\delta_{\mathrm{env}}$. This means that our method automatically adapts to the level of environment heterogeneity, enjoys affinity-based variance reduction, and never performs worse than independent learning, since $\delta_{\mathrm{env}} \leq 1$.

These observations motivate the design of *server-side* importance correction. If this correction were performed on the client side, the additional variance term in (i) would lack the mitigating $n^{-1}$ factor, and the density ratio $\mu^0(s)/\mu^i(s)$ would not be bounded by $n$ as in (iii), which could result in potentially unbounded variance that degrades performance.

We are now ready to present the main result, which shows that AffPCL achieves affinity-based variance reduction characterized by both environment and objective affinities, generalizing Section 3.

**Theorem 1.** *With a constant step size $\alpha \equiv \ln t/(\lambda t)$, AffPCL (8) with COE (7) and CDL (4) satisfies*

$$\mathbb{E}\|x_t^i - x_*^i\|^2 = \widetilde{O}(\kappa^2 t^{-1} \cdot \max\{n^{-1}, \tilde{\delta}_{\mathrm{env}}, \tilde{\delta}_{\mathrm{obj}}\}), \quad \forall i \in [n],$$

*where $\tilde{\delta}_{\mathrm{env}} \leq \min\{1, \kappa\delta_{\mathrm{env}}\}$, $\tilde{\delta}_{\mathrm{obj}} \leq \min\{1, \kappa\delta_{\mathrm{obj}}\}$ are effective environment and objective heterogeneity.*

## 6 DISCUSSION

### 6.1 DENSITY RATIO ESTIMATION

Section 5.2 requires that the density ratios $\rho^i(s) := \frac{\mu^i(s)}{\mu^0(s)}$ of environment distributions are known to the central server. This is a common assumption in supervised learning (Cortes et al., 2010; Ma et al., 2023), controlled sampling (Rubinstein & Kroese, 2016), and off-policy reinforcement learning (Precup et al., 2000; Thomas & Brunskill, 2016). It is satisfied, for example, when data are pre-collected or the covariate shift is induced by known mechanisms. When $\rho^i$ is unknown, we can incorporate asynchronous density ratio estimation (DRE) into AffPCL. Similar to COE in Section 4, DRE with linear parametrization (Sugiyama et al., 2012; Zhang et al., 2025) is also a special variant of (2) (see Appendix C.4). However, unlike COE, which enjoys affinity-based variance reduction without importance correction, DRE seeks personalized solutions, which, according to our previous analysis, requires a known density ratio for importance correction to achieve affinity-based variance reduction. This creates a chicken-and-egg problem, settled by the following information-theoretic lower bound.

**Theorem 2.** *Let $\hat{\rho}_t^i$ be any estimator of the true density ratio $\rho^i$, given $n$ agents, $t$ independent samples per agent, and no communication or computation constraint. There exists a system such that*

$$\inf_{\hat{\rho}_t^i} \mathbb{E}_{\mu^0}|\hat{\rho}_t^i(s) - \rho^i(s)|^2 \geq \min\left\{(96t)^{-1}, \delta_{\mathrm{env}}^2\right\}.$$

Theorem 2 rules out the possibility of achieving variance reduction linear in the environment heterogeneity level $\delta_{\mathrm{env}}$ without knowing the density ratio a priori. This hardness result can be circumvented if additional structure presents, such as sparsity (environment distributions differ only in a few dimensions) or coupling (environment distributions are dependent). That is, the key difference from previous problems is that affinity in DRE should be measured by criteria other than total variation distance. Our analysis of AffPCL assumes access to a DRE oracle capable of exploiting such structure to achieve affinity-based variance reduction, thereby proving Theorem 1 in full generality and further showcasing the adaptivity of AffPCL. Appendix C.3 proves Theorem 2, and Appendix C.4 contains an extended discussion on DRE in our setting.

### 6.2 NOISE ALIGNMENT

We formally define the effective heterogeneity levels in Proposition 2 and Theorem 1 as $\tilde{\delta}_{\mathrm{env}} = \min\{1, \nu\delta_{\mathrm{env}}\}$ and $\tilde{\delta}_{\mathrm{obj}} = \min\{1, \nu\delta_{\mathrm{obj}}\}$, where $\nu$ characterizes the system's "stochastic conditioning".

**Definition 3** (Stochastic condition number). $\nu := \max_i \|\bar{D}^i(\bar{A}^i)^{-1}\|$ where $D(s) := \sqrt{A(s)^T A(s)}$.

$\nu$ is trivially upper bounded by $\kappa$ (see Appendix C.5), and we refer to $\nu^{-1} \geq \kappa^{-1}$ as the *noise alignment* constant. Note that the polar decomposition gives $A(s) = U(s)D(s)$, where the positive

semidefinite matrix $D(s)$ defined as above stretches the vector it acts on, and the orthogonal matrix $U(s)$ rotates it. If $U(s)$ maintains a similar orientation for almost all $s \in \mathcal{S}$, $A(s)$ is "well-aligned" and one can see that $\nu^{-1}$ is large. Conversely, if $U(s)$ varies significantly, $\nu^{-1}$ tends to be small. The impact of affinity on variance reduction is thus modulated by this noise alignment (Lemma D.2).

We remark that while $\nu$ bears resemblance to the matrix condition number and is upper bounded by $\kappa$, the condition number $\kappa$ pertains solely to the deterministic parameters of the system, whereas $\nu$ captures the conditioning of the system's stochastic structure. Consequently, a large $\kappa$ does not necessarily imply a large $\nu$, and vice versa. Fortunately, in many cases of interest, the noise alignment constant $\nu^{-1}$ is large. A particularly relevant example is a positive semidefinite $A(s)$, a property often imposed on feature embedding matrices by design. In this case, $\nu^{-1} = 1$. See Appendix C.5 for three more examples with large $\nu^{-1}$ and further discussion on noise alignment.

### 6.3 Agent-specific affinity-based variance reduction

For ease of exposition, previous sections use the *worst-agent* heterogeneity levels (Definitions 1 and 2) to characterize the *worst-agent* performance (Theorem 1). Intuitively, agents closer to the "center" should enjoy greater affinity-based variance reduction. In Appendix G, we analyze AffPCL in full generality and obtain an *agent-specific* convergence guarantee:

$$\mathbb{E}\|x_t^i - x_*^i\|^2 = \widetilde{O}((\kappa^i)^2 t^{-1} \cdot \max\{n^{-1}, \tilde{\delta}_{\text{cen}}^i\}), \quad \forall i \in [n], \tag{9}$$

where $\kappa^i = \sigma/\lambda^i$, $\lambda^i = \lambda_{\min}(\text{sym}(\bar{A}^i))$, $\tilde{\delta}_{\text{cen}}^i = \min\{1, \nu \delta_{\text{cen}}^i\}$, and $\delta_{\text{cen}}^i$ is a more natural measure of agent $i$'s closeness to the "center agent", defined as

$$\delta_{\text{cen}}^i := \max\{\|\mu^i - \mu^0\|_{\text{TV}}, \|\bar{b}^i - \bar{b}^0\|/(2G_b)\} \in [0, 1].$$

Notably, $\delta_{\text{cen}}^i$ is affected by both objective and environment heterogeneity, and admits a trivial bound $\delta_{\text{cen}}^i \leq \min\{1, \delta_{\text{env}} + \delta_{\text{obj}}\}$ (Lemma D.1). (9) confirms that AffPCL inherently offers agent-specific affinity-based variance reduction, with agents closer to the center benefiting more from collaboration. An interesting consequence is that in the high heterogeneity regime, an agent that is not close to any other actual agent ($\delta_{\text{env}} \approx \delta_{\text{obj}} \approx 1$) may still get a "free ride" by being close to the virtual central agent ($\tilde{\delta}_{\text{cen}}^i \ll 1$), thereby gaining significant speedup. Taking this a step further, an agent can collaborate with agents that are arbitrarily heterogeneous to it but still benefit from collaboration maximally and obtain linear speedup when $\tilde{\delta}_{\text{cen}}^i \leq n^{-1}$. These insights are not captured by works that focus on linear speedup only in the low heterogeneity regime (Chayti et al., 2021; Even et al., 2022).

## 7 Numerical simulations

**Synthetic data.** We first compare AffPCL, independent learning, federated averaging (McMahan et al., 2017, FedAvg), fine-tuning (FedAvg followed by local independent learning), regularized (T Dinh et al., 2020; Li et al., 2021, pFedMe and Ditto), and clustered (Ghosh et al., 2020) FL methods, in a synthetic system with 20 agents at different heterogeneity levels $\delta_{\text{env}} = \delta_{\text{obj}} \in \{0, 0.05, 0.3, 0.8\}$, with results presented in Figure 1. The average $\text{MSE}^0 = \frac{1}{n}\sum_{i=1}^n \text{MSE}^i$ is reported. For AffPCL, we also report the MSE of the agent closest to the center to highlight the agent-specific speedup effect.

**Real-world data.** We further evaluate AffPCL on the real-world FEMNIST dataset. For clarity, we use only independent learning and FedAvg as baselines. To introduce objective heterogeneity, we consider a task where each user determines if a handwritten character is a digit and if it is a curved letter, with different users potentially having different preferences on these two objectives. We train 10 users across four levels of objective heterogeneity and report the average test MSE in Figure 2.

**Reinforcement learning.** We extend our method to the fundamental reinforcement learning algorithm SARSA, a temporal difference method that encompasses TD(0); see Appendix C.6 for details. SARSA solves a **non-linear** policy optimization problem, showcasing the versatility of AffPCL beyond linear systems. We consider 10 agents with different reward functions (objectives) and transition kernels (environments). We incorporate in this experiment the asynchronous DRE module discussed in Section 6.1. Specifically, we have $\hat{\rho}_t^i(s, a) = \frac{\hat{\mu}_t^i(s)\pi_t^i(a \mid s)}{\frac{1}{n}\sum_{j=1}^n \hat{\mu}_t^j(s)\pi_t^j(a \mid s)}$, where $\hat{\mu}_t^i(s)$ is the estimated state distribution of agent $i$ at time $t$ via naive Monte Carlo, and $\pi_t^i(a \mid s)$ is the behavior policy of agent $i$ at time $t$. The average MSE with respect to the optimal Q-function is reported in Figure 3.

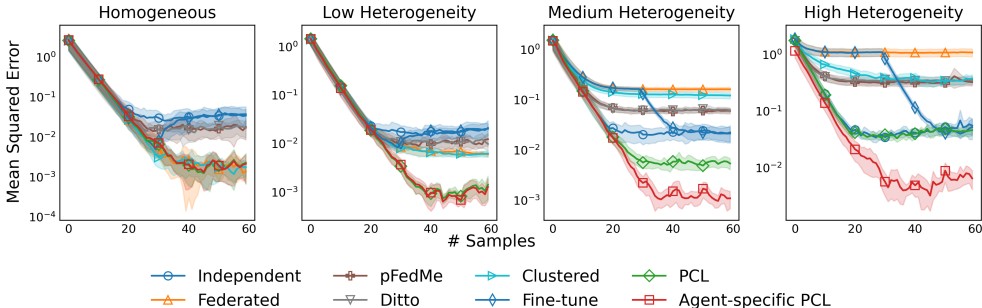

Figure 1: AffPCL matches FedAvg in the homogeneous setting and independent learning in the high heterogeneity regime. Across all scenarios, AffPCL consistently achieves the lowest MSE, while other methods' relative performance varies with the heterogeneity level. In the high heterogeneity regime where all agents are dissimilar, the agent closest to the center still enjoys significant speedup.

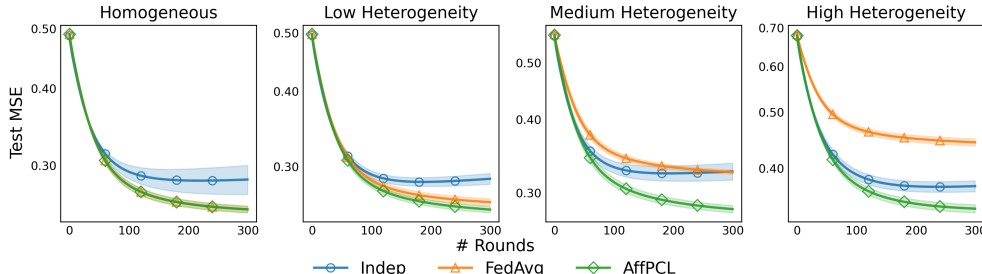

Figure 2: AffPCL matches FedAvg in the homogeneous setting and consistently achieves the lowest test MSE across all heterogeneity levels. The relative performance of FedAvg and independent learning varies with the heterogeneity level.

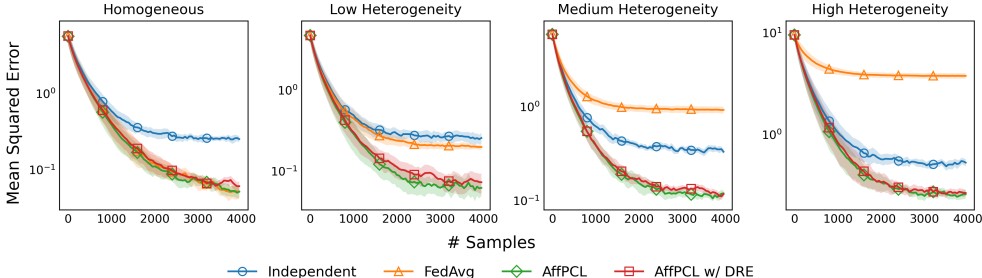

Figure 3: Consistent with other experiments, AffPCL achieves the lowest MSE across all heterogeneity levels in the reinforcement learning setting. Incorporating asynchronous DRE does not hinder the performance of AffPCL, suggesting that density estimation is of relatively low complexity compared to policy optimization.

Please refer to Appendix B for the detailed setup and additional results. These simulations validate the superiority and practicality of AffPCL and our theory of adaptive affinity-based variance reduction.

## 8 CONCLUSION AND FUTURE DIRECTIONS

AffPCL affirms that collaboration among arbitrarily heterogeneous agents can yield fully personalized solutions with adaptive affinity-based speedup, opening new avenues for harmonizing personalization and collaboration in multi-agent learning. We advocate for future endeavors in the following topics: (i) personalized feature embeddings; (ii) trade-offs between collaboration benefit and communication sophistication such as cost, privacy, and security; (iii) lower bounds on information exchange to achieve collaborative speedup; (iv) nonlinear systems, regret minimization, and stochastic optimization problems; and (v) other affinity structures such as sparsity, correlation, and low-rankness.

ACKNOWLEDGMENTS

This work was supported in part by Jane Street, the MIT-IBM Watson AI Lab, the MIT-Amazon Science Hub, the MIT-Google Program for Computing Innovation, and MathWorks.

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

# Appendix

## Table of Contents

ORGANIZATION OF APPENDIX

The appendix is organized as follows. Notation and symbols are summarized in Appendix A. The detailed setup and additional results of numerical simulations are provided in Appendix B. Appendix C supplements omitted discussions in the main text.

The remaining sections are dedicated to proving Theorem 1 in full generality, incorporating asynchronous density ratio estimation and agent-specific step sizes, and subsuming all other results in the main text. We first collect several useful lemmas in Appendix D to facilitate later analysis. The three components of AffPCL are then examined in sequence: central objective estimation (Appendix E), central decision learning (Appendix F), and personalized local learning (Appendix G).

## A NOTATION

We summarize the key notation and symbols used throughout this paper in Tables 1 and 2. We reiterate that the superscript 0 denotes the *averaged* quantity across all agents, i.e., $f^0 = \frac{1}{n}\sum_{i=1}^n f^i$ for any quantity $f$. Due to symmetry, $f^0$ usually satisfies the same property as $f^i$ for all $i \in [n]$. Therefore, in addition to the notation $[n]$, we also use $[n^0]$ to denote $\{0, 1, \ldots, n\}$. The overline denotes the *mean* quantity under the corresponding environment distribution, i.e., $\bar{f}^i = \mathbb{E}_{\mu^i} f^i(s^i)$ for any operation $f^i$. We remark that the aggregation of the mean values is not necessarily equal to the mean under the aggregated environment distribution:

$$\bar{f}^0 = \frac{1}{n}\sum_{i=1}^n \mathbb{E}_{\mu^i} f^i(s^i) \not\equiv \mathbb{E}_{\mu^0}\left[\frac{1}{n}\sum_{i=1}^n f^i(s)\right].$$

However, we have two special cases where the equality holds: (i) $f^i = f^j$ for all $i, j \in [n]$; or (ii) $\mu^i = \mu^j$ for all $i, j \in [n]$. The superscript $c$ denotes the explicitly maintained *central* quantity that aims to bridge the above discrepancy, e.g., the central objective $b^c$ and central decision variable $x_t^c$. Generally, $f^c \not\equiv f^0$ for any quantity $f$, but the equality may hold in the two special cases above.

For the ease of presentation, we use the following shorthand notation throughout the analysis: $\Delta z_t$ represents the optimality gap $z_t - z_*$ at time step $t$ for any decision variable $z$; for a function $f$ on $\mathcal{S}$, $f_t^i$ represents its evaluation at the observation $s_t^i$, and $f_t^0 = \frac{1}{n}\sum_{i=1}^n f(s_t^i)$; $\mathbb{E}_t^i := \mathbb{E}_{s_t^i \sim \mu^i}$ and $\mathbb{E}_t := \mathbb{E}_{s_t^i \sim \mu^i, i \in [n]}$; $\mathbb{E}_{\mathcal{F}_{t-1}}$ represents the conditional expectation given the history filtration $\mathcal{F}_{t-1}$ that contains all the randomness up to time step $t-1$.

We use $\succ, \succeq, \preceq, \prec$ to denote the Loewner order and $\gtrsim, \asymp, \lesssim$ to denote the asymptotic order as $t \to \infty$.

Table 1: Notation.

| Notation | Description |
|---|---|
| $[n], [n^0]$ | $\{1, 2, \ldots, n\}, \{0, 1, \ldots, n\}$ |
| $\Delta z_t$ | decision variable optimality gap $z_t - z_*$ |
| $f^i, f_t^i$ | $i$-th agent's quantity, and its realization at time step $t$ (if a random variable) or evaluation at observation $s_t^i$ (if a function) |
| $f^0$ | averaged quantity across agents $\frac{1}{n}\sum_{i=1}^n f^0$ |
| $f^c$ | explicitly maintained central quantity |
| $\bar{f}^i$ | expected quantity under agent $i$'s environment distribution $\mathbb{E}_{\mu^i} f^i(s)$ |
| $\bar{f}^0$ | aggregated expected quantity $\frac{1}{n}\sum_{i=1}^n \mathbb{E}_{\mu^i} f^i(s)$ |
| $\hat{f}_t$ | estimation of $f$ at time step $t$ |
| $g^{0 \to i}, g^{c \to i}$ | bias correction from aggregated/central update direction to agent $i$ |
| $g^{c \rightrightarrows i}$ | importance-corrected update direction from central to agent $i$ |

Table 2: Symbols.

| Symbol | Description | Symbol | Description |
|---|---|---|---|
| $A$ | feature matrix | $\alpha$ | step size |
| $b$ | objective | $\beta$ | Young's inequality parameter |
| $C$ | constant | $\chi^2$ | chi-square divergence |
| $d$ | system dimension | $\delta$ | heterogeneity level |
| $\mathcal{E}$ | Estimation error | $\Delta(\mathcal{S})$ | probability measure space |
| $\mathcal{F}$ | Filtration | $\eta$ | density ratio weight |
| $g$ | update direction | $\gamma$ | reward discount factor |
| $G$ | system parameter bound | $\kappa$ | condition number |
| $i, j, k$ | agent index | $\lambda$ | minimal eigenvalue |
| $\mathcal{L}$ | Lyapunov function | $\mu$ | environment distribution |
| $n$ | number of agents | $\nu$ | stochastic condition number |
| $s, \mathcal{S}$ | state, state space | $\phi, \Phi, \psi, \Psi$ | feature map |
| $t, \tau$ | time step | $\rho$ | density ratio |
| $w$ | weight | $\sigma$ | problem scale/ variance proxy |
| $x, z$ | decision variable | $\theta$ | objective weight |

## B  ADDITIONAL NUMERICAL SIMULATIONS

**Synthetic setup.**  We run our simulations in a synthetic multi-agent linear system with $n = 20$ agents in a $d = 5$ dimensional space. Agents possess distinct multivariate Gaussian distributions $\mu^i = \mathcal{N}(m_i, I_d)$ as their environments, and their personalized objectives are given by linear models $b^i(s) = \Phi(s)\theta_*^i$. We construct the stochastic feature embedding matrices $A(s)$ and $\Phi(s)$ to have a multiplicative noise structure (Example 7): $A(s) = (I_d + \epsilon_A \cdot ss^T)A_{\text{base}}$ and $\Phi(s) = (I_d + \epsilon_b \cdot ss^T)\Phi_{\text{base}}$, where $A_{\text{base}}$ and $\Phi_{\text{base}}$ are randomly generated positive definite matrices for each problem instance with condition numbers of $O(1)$. We set $\sigma_A = 1$ and $\sigma_b = 0.5$ to control the level of stochastic noise alignment. The reference personalized solutions $x_*^i$ are calculated using Monte Carlo estimation with 5000 samples.

To generate heterogeneous environments, we set $m_i = \delta_{\text{env}} C_A v_i$, where $v_i$ is a random unit vector and $C_A = 4$ satisfies that $\|\mathcal{N}(0, I_d) - \mathcal{N}(C_A \mathbf{1}, I_d)\|_{\text{TV}} \geq 0.9$. This ensures that the environment heterogeneity level goes to 1 as $\delta_{\text{env}}$ approaches 1 (Definition 2). Similarly, heterogeneous objectives are generated by setting $\theta_*^i = \theta_{\text{base}} + \delta_{\text{obj}} u_i$, where $u_i$ is a random unit vector and $\theta_{\text{base}} \sim \mathcal{N}(0, I_d)$. This construction ensures that the objective heterogeneity level is of $O_P(\delta_{\text{obj}})$ (Definition 1). For reference, we fix the first agent's environment as $\mu^1 = \mathcal{N}(0, I_d)$ and $\theta_*^1 = \theta_{\text{base}}$, making it close to the "center" when the number of agents $n$ is large.

We compare our proposed AffPCL algorithm against the following baselines:

1. **Independent learning**, where each agent learns its own solution using its local data without communication.
2. **Federated averaging** (FedAvg), where all agents collaboratively learn a unified solution by averaging their update directions.
3. **Fine-tuning**, where agents first run FedAvg for 30 steps to learn a common model, and then fine-tune their personalized models independently for another 30 steps;
4. **Regularized FL**, pFedMe (T Dinh et al., 2020) and Ditto (Li et al., 2021) specifically, where agents collaboratively learn a global model, and then learn personalized models by solving a regularized objective that penalizes the distance to the global model. The regularization parameter is set to 15 for both methods.
5. **Clustered FL** (Ghosh et al., 2020), where agents are iteratively clustered based on the similarity of their local systems, and then collaboratively learn a model within each cluster. The number of clusters is set to 10.

The results are reported in Figure 1. All algorithms are run for $t = 60$ steps with a fixed learning rate of $\alpha = 0.01$. All experiments are repeated for 10 runs, and we report the mean squared error

averaged over all agents $\mathrm{MSE}^0 = \frac{1}{n}\sum_{i=1}^{n}\|x_t^i - x_*^i\|^2$, along with the 90% confidence region. To showcase the agent-specific affinity-based variance reduction, we also report the error of the first agent $\mathrm{MSE}^1 = \|x_t^1 - x_*^1\|^2$.

**Comparison with baselines.** We evaluate the performance of all algorithms under different heterogeneity levels $(\delta_{\mathrm{env}}, \delta_{\mathrm{obj}})$. Figure 1 reports the homogeneous setting $(0.0, 0.0)$, low heterogeneity $(0.05, 0.05)$, medium heterogeneity $(0.2, 0.2)$, and high heterogeneity $(0.5, 0.5)$. Results of exhaustive sweeps over $(\delta_{\mathrm{env}}, \delta_{\mathrm{obj}})$ are presented in Figure 4, where we report the improvement of AffPCL over independent learning and FedAvg, measured by the average $\mathrm{MSE}^0$ over the last 10 time steps. We remark that when agents' environment distributions vary greatly $(\delta_{\mathrm{env}} \geq 0.9)$, all algorithms experience high variance and thus the results may not be statistically significant. Figure 4a demonstrates that AffPCL consistently outperforms independent learning, with the affinity-based speedup increasing as the heterogeneity level decreases. Figure 4b shows that AffPCL matches FedAvg in the homogeneous setting, while FedAvg fails to provide any personalization in the presence of heterogeneity.

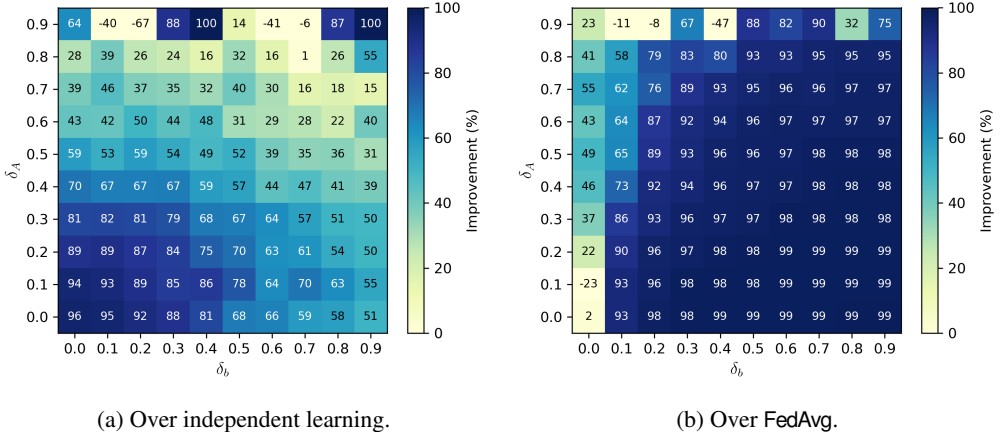

(a) Over independent learning.  (b) Over FedAvg.

Figure 4: Improvement of AffPCL.

**Federated vs. affinity-based speedup.** Our theory identifies two factors in the variance reduction of AffPCL: federated speedup $n^{-1}$ and heterogeneity level $\delta$. We conduct exhaustive sweeps over the number of agents $n \in [2, 50]$ and heterogeneity level $\delta = \delta_{\mathrm{env}} = \delta_{\mathrm{obj}} \in [0.02, 0.5]$. Iso-performance contours of AffPCL are plotted in Figure 5, where each curve represents the combinations of $(n^{-1}, \delta)$ that yield the same average $\mathrm{MSE}^0$ over the last 10 steps. As expected, the contours form Pareto-type curves, confirming that $\max\{n^{-1}, \delta\}$ characterizes the trade-off between collaboration and affinity in AffPCL.

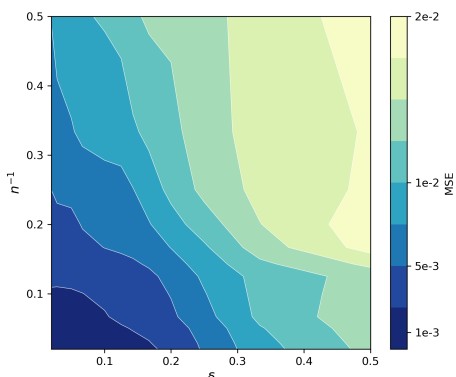

Figure 5: Iso-performance contours of AffPCL.

**Agent-specific performance.** Another highlight of our theory is the agent-specific affinity-based variance reduction effect, where agents closer to the center benefit more from collaboration (Section 6.3). We examine this phenomenon in a high heterogeneity setting $(\delta_{\text{env}}, \delta_{\text{obj}}) = (0.7, 0.7)$ and report the performance of independent learning and AffPCL for a generic agent and for the agent closest to the center in Figure 6. Looking at the performance of generic agent, AffPCL performs similarly to independent learning, as the collaboration benefit gets diminished by the high heterogeneity. However, the agent closest to the center still gets a "free ride" and achieves significant speedup, compared to learning on its own, through collaborating with other agents. We remark that in the high heterogeneity regime, the agent closest to the center may not be close to any other agents, yet the collaboration benefit remains.

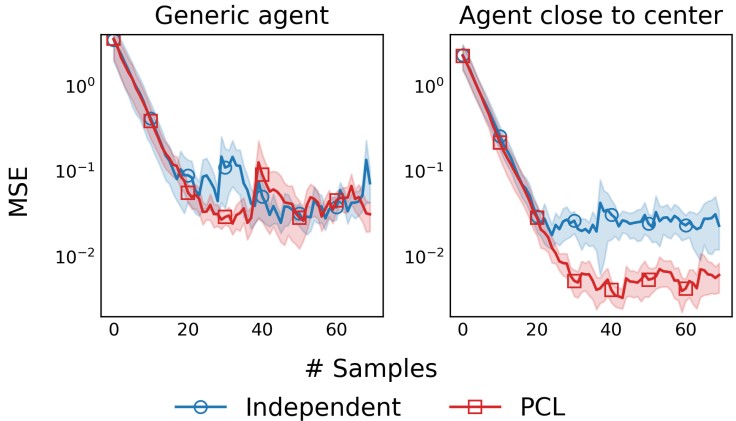

Figure 6: Agent-specific performance.

**Real-world setup.** For the real-world FEMNIST dataset, we first pre-train a numerical network using some training data as the feature extractor $\phi$ shared across all users. The dimensionality of the extracted feature is $d = 84$. To introduce varying levels of objective heterogeneity, we consider the following label for input character image $z$:

$$y(z; \lambda) = \lambda \cdot \mathbb{1}\{\text{character } z \text{ is a digit}\} + (1 - \lambda) \cdot \mathbb{1}\{\text{character } z \text{ is a curved letter}\}.$$

Then, the least squares linear regression problem corresponds to the following linear system:

$$\mathbb{E}_{\mu^i} \left[ \phi(z)\phi(z)^T \right] x^i = \mathbb{E}_{\mu^i} \left[ \phi(z)y(z; \lambda^i) \right], \quad i = 1, \ldots, n,$$

where $\mu^i$ is the data distribution of user $i$. In our implementation of AffPCL for this task, we omit the importance correction. We test four levels of objective heterogeneity by setting the range of $\lambda$ as $[0.5 - \delta_{\text{obj}}/2, 0.5 + \delta_{\text{obj}}/2]$ with $\delta_{\text{obj}} \in \{0, 0.2, 0.6, 1\}$. 10 users have evenly distributed $\lambda^i$ in the specified range. All three algorithms share the same hyperparameters: learning rate $\alpha = 0.002$, batch size is 32, and the number of samples between two communication rounds is 70. The test MSE averaged over all users is reported in Figure 2. The results are consistent with those in the synthetic experiments, validating the practicality of AffPCL.

**Reinforcement learning setup.** We follow the derivation in Appendix C.6 and setup in Zhang et al. (2024) to implement AffPCL, independent learning, and FedAvg versions of SARSA. We consider 10 agents, 10 states, 5 actions, and a feature dimension of $d = 20$ for the state-action space. The other RL hyperparameters are set as follows: reward discount factor $\gamma = 0.1$, behavior policy is softmax with temperature 10, and step size $\alpha = 0.1$. We perturb a nominal reward function and transition kernel to generate heterogeneous objectives and environments, with relative perturbation levels $\delta_{\text{obj}}, \delta_{\text{env}} \in \{0.0, 0.2, 0.5, 1.0\}$. The reference optimal Q-function is calculated using offline value iteration with the true model. For the asynchronous DRE module, we use the following density ratio estimator:

$$\hat{\rho}_t^i(s, a) = \frac{\hat{\mu}_t^i(s)\pi_t^i(a \mid s)}{\frac{1}{n}\sum_{j=1}^n \hat{\mu}_t^j(s)\pi_t^j(a \mid s)},$$

where $\hat{\mu}_t^i(s)$ is estimated using the naive Monte Carlo, i.e., average state visitation frequency up to time step $t$ for agent $i$, and $\pi_t^i(a \mid s)$ is the behavior policy of agent $i$ corresponding to its current Q-function estimate. The average MSE with respect to the reference optimal Q-functions is reported in

Figure 3. The results are consistent with other experiments and show that incorporating asynchronous DRE does not hinder the performance of AffPCL.

## C  FURTHER DISCUSSIONS

### C.1  IMPLEMENTATION DETAILS

We present in Algorithm 1 the pseudocode of AffPCL with asynchronous COE and CDL.

---

**Algorithm 1:** Personalized collaborative learning (AffPCL)

**initialize:** $x_0^c, \theta_0^c, x_0^i$ for $i \in [n]$.

1 **for** $t = 0, 1, \ldots$ **do**
2     **foreach** agent $i \in [n]$ in parallel **do**
3         sample $s_t^i \sim \mu^i$
4         evaluate residuals $g_t^i(x_t^i), g_t^i(x_t^c), g_t^{i,b}(\theta_t^c), g_t^{c \to i}(x_t^c)$
5         send $(s_t^i, g_t^i(x_t^c), g_t^{i,b}(\theta_t^c), g_t^{c \to i}(x_t^c))$ to the server
6     **at central server:**
7         aggregate central residuals $g_t^{0,c}(x_t^c), g_t^{0,b}(\theta_t^c), g_t^{c \rightleftarrows i}(x_t^c)$ for $i \in [n]$
8         send central residuals back to agents
9     **foreach** agent $i \in [n]$ in parallel **do**
10         $x_{t+1}^c = x_t^c - \alpha_t g_t^{0,c}(x_t^c)$
11         $\theta_{t+1}^c = \theta_t^c - \alpha_t g_t^{0,b}(\theta_t^c)$
12         $x_{t+1}^i = x_t^i - \alpha_t(g_t^i(x_t^i) + g_t^{c \rightleftarrows i}(x_t^c) - g_t^{c \to i}(x_t^c))$

---

We provide several remarks on central decision learning (CDL). First, if the central server has memory, the central decision variable $x_t^c$ and central objective parameter $\theta_t^c$ can also be maintained and updated (Lines 10-11) at the server side.

Second, when agents share the same environment distribution, the central decision variable can be replaced by the average decision variable $x_t^0$, since the solution to the central system (3) coincides with the average of personalized solutions, i.e., $x_*^c = x_*^0$. In this way, CDL happens implicitly without executing (4), reducing computation complexity. However, this implementation requires an additional communication round to compute $x_{t+1}^0$ after each local update, increasing communication complexity.

Third, when agents have different environment distributions, note that the central update direction $g_t^{c \rightleftarrows i}(x_t^c)$ in (8) now involves $g_t^{c \to i}(x_t^c)$ (which uses the estimated central objective $\hat{b}_t^c(s_t^i)$), instead of $g_t^i(x_t^c)$ (which uses personalized objective $b^i(s_t^i)$) as in (4). This modification is necessary for the importance correction to work. That said, it should be unsurprising that if we also use $g_t^{c \to i}$ to compute the central update direction in central learning, i.e., using $g_t^0(x_t^c) = \frac{1}{n}\sum_{i=1}^n g_t^{c \to i}(x_t^c)$ in (4), the convergence guarantee still holds. For completeness, we also prove convergence for both implementations in Appendix F.

*Remark* 1 (Communication and computation complexity). To provide a clearer picture of scalability in our stylized setup, we compare the communication and computation complexity of AffPCL with federated averaging (FedAvg), both with immediate communication after each local update. In FedAvg, each round involves the communication of all local residuals and one averaged residual, resulting in a communication complexity of $\mathrm{comm}(\mathsf{FedAvg}) = \Theta(2nd)$ per round, where $d$ is the system dimension. Updating the local decision variable results in a computation complexity of $\mathrm{comp}(\mathsf{FedAvg}) = \Theta(d)$ per agent per round. In AffPCL with COE, CDL, and DRE, as detailed in Algorithm 1, each round has a communication complexity $\mathrm{comm}(\mathsf{AffPCL}) \leq 4\,\mathrm{comm}(\mathsf{FedAvg})$. Suppose we calculate the central update direction using $g_t^{c \to i}$ as discussed above, and suppose the density ratio is known a priori, then the communication complexity reduces to $\mathrm{comm}(\mathsf{AffPCL}) \leq 2.5\,\mathrm{comm}(\mathsf{FedAvg}) = \Theta(5nd)$. Similarly, the computation complexity per agent per round in AffPCL (Algorithm 1) is $\mathrm{comp}(\mathsf{AffPCL}) = 6\,\mathrm{comp}(\mathsf{FedAvg}) = \Theta(6d)$, where the extra factor comes from CDL, COE, importance correction, and composing the personalized update direction.

The above analysis shows that, although sharing the same asymptotic communication and computation complexity as FedAvg, AffPCL incurs a larger constant factor (specifically, up to 4 times for communication and 6 times for computation per iteration) due to the additional components and personalization. This inherent trade-off calls for future research into developing more sophisticated communication schemes to reduce the communication and computation overhead of AffPCL, and deriving lower bounds on the information exchange to achieve collaborative benefits in personalized learning. Additionally, we remark that numerous techniques from federated and decentralized learning that reduce the number of communication rounds, such as multiple local updates, compression, and partial participation, can be readily incorporated into AffPCL. We consider the stylized setting of immediate communication to focus on the main ideas.

## C.2  LINEAR PARAMETRIZATION OF OBJECTIVE AND DENSITY RATIO

We first note that linear parametrization covers finite state spaces as a special case. For DRE, $\psi(s) = e_s$ is the one-hot encoding of state $s$, and then $\eta_*^i = (\rho^i(s_1), \ldots, \rho^i(s_{|\mathcal{S}|}))^T$ simply records the density ratio for all states. For COE, we transform the original objective as $b^i(s) \leftarrow e_s \otimes b^i(s) \in \mathbb{R}^{d|\mathcal{S}|}$, where $\otimes$ denotes the Kronecker product. Then, $\Phi(s) = e_s e_s^T \otimes I_d \in \mathbb{R}^{d|\mathcal{S}| \times d|\mathcal{S}|}$, and $\theta_*^i$ similarly records the objective vector for all states.

Linear parametrization is also widely used in supervised learning (parametric regression) and reinforcement learning (linear value function approximation). COE performs parametric estimation with a linear function class: $\theta_*^i = \arg\min_{\theta \in \mathbb{R}^d} \|\hat{b}_\theta^i - b^i\|$, where $\hat{b}_\theta^i(s) = \Phi(s)\theta$, and we omit the discussion of approximation error when the model is misspecified, i.e., $b^i(s) \notin \{\Phi(s)\theta : \theta \in \mathbb{R}^d\}$. See Sugiyama et al. (2012) for more details on DRE with linear parametrization.

When applied to reinforcement learning, our linear parametrization subsumes linear Markov reward processes (Bhandari et al., 2018), where $b^i(s) = \phi_p(s)r^i(s) = \phi_p(s)\phi_r(s)^T \theta_*^i$, with $r^i$ as agent $i$'s reward function and $\phi_p, \phi_r$ as feature maps. See Appendix C.6 for more details on this application.

We choose linear parametrization for its simplicity, allowing us to focus on the main ideas. Our method readily extends to other (non)parametric models, provided the function class has complexity polynomial in $d$ rather than in $|\mathcal{S}|$.

## C.3  PROOF OF THEOREM 2

We restate and establish the lower bound of DRE MSE. Our proof generally follows the standard Le Cam's method with two special treatments: (i) we show that DRE is lower bounded by a special density estimation problem, (ii) we show that collaborating with other agents does not help in estimating one agent's own density ratio.

**Theorem 2.** *Let $\hat{\rho}_t^i(s)$ be the estimate of true density ratio $\rho^i$, given by any algorithm with $n$ agents and $t$ independent samples per agent, with no communication or computation constraint. There exists a problem instance such that*

$$\inf_{\hat{\rho}_t^i} \mathbb{E}_{\mu^0} |\hat{\rho}_t^i(s) - \rho^i(s)|^2 \geq \min\left\{(96t)^{-1}, \delta_{\text{env}}^2\right\}.$$

*Proof.* In this proof, we omit the agent index $i$ and thus $\rho = \mu/\mu^0$. We build the hard problem instance step by step. We first consider a finite state space $\mathcal{S} = [d]$ and thus $\rho$ and $\hat{\rho}_t$ are vectors in $\mathbb{R}^d$ (this is equivalent to a linear function approximation with feature map $\psi(s) = e_s$, the $s$-th standard basis vector). Then, the minimax risk w.r.t. the MSE loss is defined as

$$R^* = \inf_{\hat{\rho}_t} \sup_{0 \leq \rho \leq n} \sup_{\{\mu, \mu^0 \in \Delta(\mathcal{S}) : \mu/\mu^0 = \rho\}} \mathbb{E}_{(\mu \times \mu^0)^{\otimes t}} \mathbb{E}_{\mu^0} |\hat{\rho}_t(s) - \rho(s)|^2,$$

where $\hat{\rho}_t$, a random vector, is the estimate learned from the sample drawn from $(\mu \times \mu^0)^{\otimes t}$. We use the convention that if any constraint set is empty, the risk is zero. We note that although samples across time steps are i.i.d., the samples from $\mu$ and $\mu^0$ at the same time step are correlated as $\mu^0 = \frac{1}{n}\sum_{j=1}^n \mu^j$. However, this correlation scales as $O(n^{-1})$ and diminishes as $n$ increases. We then apply several reductions. By set equivalence,

$$R^* = \inf_{\hat{\rho}_t} \sup_{\mu^0 \in \Delta(\mathcal{S})} \sup_{\mu = \rho\mu^0, 0 \leq \rho \leq n} \mathbb{E}_{(\mu \times \mu^0)^{\otimes t}} \mathbb{E}_{\mu^0} |\hat{\rho}_t(s) - \rho(s)|^2.$$

We now fix $\mu^0 = d^{-1}\mathbf{1} \in \Delta(\mathcal{S})$ and define a convex set $\mathcal{M} := \{0 \leq \rho \leq n : \rho\mu^0 \in \Delta(\mathcal{S})\}$.[4] Then,

$$R^* \geq \inf_{\hat{\rho}_t} \sup_{\mu=\rho\mu^0, \rho\in\mathcal{M}} \mathbb{E}_{(\mu\times\mu^0)^{\otimes t}}\mathbb{E}_{\mu^0}|\hat{\rho}_t(s) - \rho(s)|^2$$
$$= d^{-1}\inf_{\hat{\rho}_t}\sup_{\mu\in\mathcal{M}}\mathbb{E}_{(\mu\times\mu^0)^{\otimes t}}\|\hat{\rho}_t - \rho\|_2^2.$$

For a sufficiently small $\epsilon$ to be determined, we can choose $\rho_1, \rho_2 \in \mathcal{M}$ such that $\|\rho_1 - \rho_2\|_2 \geq 2\epsilon$. Following Le Cam's method, we construct a test $\hat{\varphi}_t = \operatorname{argmin}_{m\in[2]}\|\hat{\rho}_t - \rho_m\|_2$. Then,

$$R^* \geq d^{-1}\inf_{\hat{\rho}_t}\frac{1}{2}\sum_{m=1}^{2}\epsilon^2 P_m(\hat{\varphi}_t \neq m)$$

$$\geq d^{-1}\epsilon^2\inf_{\hat{\varphi}_t}\frac{1}{2}\sum_{m=1}^{2}P_m(\hat{\varphi}_t \neq m)$$

$$\geq \frac{\epsilon^2}{2d}\left(1 - \|P_1 - P_2\|_{\text{TV}}\right), \tag{10}$$

where $P_m = (\mu_m \times \mu^0)^{\otimes t}$. By Pinsker's inequality and properties of KL divergence,

$$2\|P_1 - P_2\|_{\text{TV}}^2 \leq D_{\text{KL}}(P_1\|P_2)$$
$$= tD_{\text{KL}}(\mu_1 \times \mu^0\|\mu_2 \times \mu^0)$$
$$= t(\underbrace{D_{\text{KL}}(\mu_1\|\mu_2)}_{H_1} + \underbrace{D_{\text{KL}}(\mu_1^0\|\mu_2^0\,|\,\mu_1)}_{H_2}), \tag{11}$$

where $\mu_m^0$ is conditional distribution of $s^0$ given $s \sim \mu_m$ and $(s, s^0) \sim \mu_m \times \mu^0$. Recall that $\mu^0$ is the aggregated distribution of agents' distributions.

We proceed to construct $\rho_1, \rho_2$ and bound the KL divergence terms. Suppose $d$ is even. Let

$$\mu_m(s) = d^{-1} + d^{-3/2}\epsilon(-1)^{s+m}, \quad s \in [d], m \in [2].$$

Let $\epsilon \leq \sqrt{d}$. One can verify that $\mu_m \in \Delta(\mathcal{S})$ and

$$\|\rho_1 - \rho_2\|_2 = d\|\mu_1 - \mu_2\|_2 = d \cdot 2d^{-3/2}\epsilon \cdot \sqrt{d} = 2\epsilon.$$

Further, let $\epsilon \leq \sqrt{d}/2$. Then, for the first KL divergence term, we bound it using the chi-square divergence:

$$H_1 \leq \chi^2(\mu_1\|\mu_2) = \sum_{s=1}^{d}\frac{(\mu_1(s) - \mu_2(s))^2}{\mu_2(s)} \leq \sum_{s=1}^{d}\frac{4d^{-3}\epsilon^2}{d^{-1}/2} = 8d^{-1}\epsilon^2.$$

For the second KL divergence term, we first have the decomposition of $\mu^0 = \frac{1}{n}\mu_m + \frac{n-1}{n}\mu_m'$, for $m \in [2]$, where $\mu_m'$ is the aggregated distribution of all agents except agent $i$ and is independent of $\mu_m$. Then, the conditional distribution given $s \sim \mu_m$ is simply $\mu_m^0 = \frac{1}{n}\delta_s + \frac{n-1}{n}\mu_m'$, for $m \in [2]$. If $n \leq 1$, then $\mu_1^0 = \mu_2^0$ and $H_2 = 0$. Thus, we consider $n \geq 2$. By the convexity of KL divergence and Jensen's inequality,

$$H_2 \leq \frac{1}{n}D_{\text{KL}}(\delta_s\|\delta_s\,|\,\mu_1) + \frac{n-1}{n}D_{\text{KL}}(\mu_1'\|\mu_2'\,|\,\mu_1) = \frac{n-1}{n}D_{\text{KL}}(\mu_1'\|\mu_2').$$

Again, bounding it by chi-square divergence gives

$$H_2 \leq \frac{n-1}{n}\sum_{s=1}^{d}\frac{(\mu_1'(s) - \mu_2'(s))^2}{\mu_2'(s)}.$$

Notice that

$$\mu^0 = \frac{1}{n}\mu_1 + \frac{n-1}{n}\mu_1' = \frac{1}{n}\mu_2 + \frac{n-1}{n}\mu_2' \implies (\mu_1' - \mu_2')^2 = \left(\frac{\mu_1 - \mu_2}{n-1}\right)^2 = \frac{4d^{-3}\epsilon^2}{(n-1)^2}.$$

[4]Here $\rho\mu^0$ is the element-wise product as we treat them as functions on $\mathcal{S}$.

Together with $\mu_2'(s) = d^{-1} - \frac{1}{n-1}d^{-3/2}\epsilon(-1)^s \geq d^{-1}/2$, we have

$$H_2 \leq \frac{8d^{-1}\epsilon^2}{n(n-1)} \leq 4d^{-1}\epsilon^2.$$

Plugging $H_1$ and $H_2$ back into (11) and combining (10) gives

$$R^* \geq \frac{\epsilon^2}{2d}\left(1 - \epsilon\sqrt{\frac{6t}{d}}\right). \tag{12}$$

We are left to determine $\epsilon$. There are two cases. The first is that $\delta_{\text{env}} \leq \frac{1}{4\sqrt{6t}}$, i.e., $\mu$ is close to $\mu^0$, or we do not have many samples. In this case, one would constrain the estimator to get a smaller error. Note that

$$\|\mu_m - \mu^0\|_{\text{TV}} = \frac{1}{2}d \cdot d^{-3/2}\epsilon, \quad m \in [2].$$

Thus, pushing $\mu_1$ and $\mu_2$ to the boundary of the ball $\{\mu : \|\mu - \mu^0\|_{\text{TV}} \leq \delta_{\text{env}}\}$ gives $\epsilon = 2\delta_{\text{env}}\sqrt{d}$. Plugging this into (12) gives

$$R^* \geq 2\delta_{\text{env}}^2(1 - 2\delta_{\text{env}}\sqrt{6t}) \geq \delta_{\text{env}}^2.$$

The second case is that $\delta_{\text{env}} > \frac{1}{4\sqrt{6t}}$, where we need to make $\mu_1$ and $\mu_2$ closer to make their discrimination harder. In this case, we set $\epsilon = \frac{1}{2}\sqrt{\frac{d}{6t}}$. Then $\|\mu_m - \mu^0\|_{\text{TV}} = \frac{1}{4\sqrt{6t}} < \delta_{\text{env}}$ so $\mu_1$ and $\mu_2$ are feasible. Plugging $\epsilon$ into (12) gives

$$R^* \geq \frac{1}{96t}.$$

Combining the two cases gives

$$R^* \geq \min\{\delta_{\text{env}}^2, (96t)^{-1}\}.$$

$\square$

## C.4 DENSITY RATIO ESTIMATION

This subsection first shows that DRE with linear parametrization is a special variant of (2) and then discusses several environment affinity structures that help circumvent the lower bound in Theorem 2 and enable affinity-based variance reduction in DRE.

A linear parametrization of density ratio (Sugiyama et al., 2012) takes the form $\rho^i(s) = \psi(s)^T\eta_*^i$ for all $i \in [n]$, where $\psi(s) \in \mathbb{R}_+^d$ is a measure basis and $\eta_*^i \in \mathbb{R}_+^d$ is the true weight. Let $\Psi(s) \coloneqq \psi(s)\psi(s)^T$. Then,

$$\mathbb{E}_{\mu^0}\Psi(s)\eta_*^i = \int_{\mathcal{S}}\psi(s)\rho^i(s)\mu^0(s)\mathrm{d}s = \int_{\mathcal{S}}\psi(s)\mu^i(s)\mathrm{d}s = \mathbb{E}_{\mu^i}\psi(s).$$

Therefore, a simple stochastic fixed point iteration for (2) described in the paper (see also Example 2) finds $\eta_*^i$ with an MSE of $O(t^{-1})$, but the affinity-based variance reduction is unattainable because of Theorem 2.

Alternatively, notice that $\rho^i(s) - 1$ directly measures the affinity between $\mu^i$ and $\mu^0$ and is the quantity through which $\rho^i$ enters the analysis. Thus, we can directly apply linear parametrization to $\rho^i(s) - 1 = \psi(s)^T\eta_*^i$. Then at time step $t$, $\hat{\rho}_t^i = 1 + \psi(s_t^i)^T\eta_t^i$. The DRE problem becomes

$$\mathbb{E}_{\mu^0}\Psi(s)\eta_*^i = \int_{\mathcal{S}}\psi(s)\left(\rho^i(s) - 1\right)\mu^0(s)\mathrm{d}s = \int_{\mathcal{S}}\psi(s)\left(\mu^i(s) - \mu^0(s)\right)\mathrm{d}s =: \mathbb{E}_{\mu^i - \mu^0}\psi'(s).$$

This problem formulation is easier to work with because $\eta_*^i \approx 0$ when $\mu^i \approx \mu^0$; in such cases, regularization can be applied if it is known a priori that agents' environments are similar.

Theoretically, this regularization will not work if our prior knowledge of environment similarity is measured in total variation distance, i.e., $\delta_{\text{env}}$, due to Theorem 2. Nonetheless, several other affinity structures can help.

**Example 1** (Sparsity). $\eta_*^i$ encodes the difference between $\mu^i$ and $\mu^0$ (recall that in the tabular case, $\eta_*^i(s) = \rho^i(s) - 1 = (\mu^i(s) - \mu^0(s))/\mu^0(s)$; see Appendix C.2). If $\mu^i$ and $\mu^0$ differ only in a few dimensions, i.e., $\eta_*^i$ is $\delta_{\text{env}}d$-sparse such that $\|\eta_*^i\|_0 \leq \delta_{\text{env}}d$, then we can use $\ell_0$-constrained or $\ell_1$-regularized least squares to estimate $\eta_*^i$, which can be calculated efficiently online and has a standard MSE of $\widetilde{O}(\kappa^2 t^{-1} \cdot \delta_{\text{env}}d)$.

**Example 2** (Coupling). Suppose DRE uses coupled samples from $\mu^i$ and $\mu^0$, such that $P(s_t^i = s_t^0) = 1 - \delta_{\text{env}}$. Note that $P(s_t^i = s_t^0) \leq 1 - \|\mu^i - \mu^0\|_{\text{TV}}$ and the equality is attained when $\mu^i$ and $\mu^0$ are optimally coupled. Then, a simple fixed-point iteration

$$\eta_{t+1}^i = \eta_t^i - \alpha_t^\rho g_t^{i,\rho}(\eta_t^i) := \eta_t^i - \alpha_t^\rho \left( \Psi(s_t^0)\eta_t^i - (\psi(s_t^i) - \psi(s_t^0)) \right)$$

with a properly chosen step size $\alpha_t^\rho = \widetilde{O}(t^{-1})$ has an MSE of $\widetilde{O}(\kappa^2 t^{-1} \cdot \delta_{\text{env}})$.

*Proof.* We only need to show that the update is monotone and its variance at the fixed point enjoys affinity-based variance reduction; then the result follows from standard stochastic approximation analysis (see e.g., Appendix E). First, we have

$$\mathbb{E}g_t^{i,\rho}(\eta_t^i) = \mathbb{E}_{\mu^0}\Psi(s)\eta_t^i - \mathbb{E}_{\mu^i - \mu^0}\psi'(s) = \bar{\Psi}^0(\eta_t^i - \eta_*^i).$$

The montonicity follows from

$$\left\langle \Delta\eta_t^i, \bar{\Psi}^0\Delta\eta_t^i \right\rangle \geq \lambda_{\min}(\bar{\Psi}^0)\|\Delta\eta_t^i\|^2.$$

For the variance, without loss of generality, we assume normalized feature $\|\psi(s)\| \leq 1$. Then,

$$\begin{aligned}
\mathbb{E}\|g_t^{i,\rho}(\eta_*^i)\|^2 &= \mathbb{E}\|\Psi(s_t^0)\eta_*^i - (\psi(s_t^i) - \psi(s_t^0))\|^2 \\
&= \mathbb{E}\|\psi(s_t^0)(\rho^i(s_t^i) - 1) - (\psi(s_t^i) - \psi(s_t^0))\|^2 \\
&\leq 2\mathbb{E}\|\psi(s_t^0)(1 - \rho^i(s_t^0))\|^2 + 2\mathbb{E}\|\psi(s_t^i) - \psi(s_t^0)\|^2 \\
&\leq 2\chi^2(\mu^i, \mu^0) + 2 \cdot 4P(s_t^i \neq s_t^0) \\
&\leq 2\|\rho^i\|_\infty\|\mu^i - \mu^0\|_{\text{TV}} + 8(1 - P(s_t^i = s_t^0)) \\
&= O(\delta_{\text{env}}).
\end{aligned}$$

$\square$

These examples indicate that DRE requires a stricter affinity measure to achieve affinity-based variance reduction. To enable maximum generality, we make the following assumption.

**Assumption 1** (DRE oracle). We assume access to a DRE oracle that returns an estimate weight $\eta_t^i$ or density ratio $\hat{\rho}_t^i$ such that $|\hat{\rho}_t^i(s) - \rho^i(s)| = O(1)$ throughout the learning process and

$$\mathbb{E}\|\hat{\rho}_t^i(s) - \rho^i(s)\|^2 = \widetilde{O}((\kappa^\rho)^2 t^{-1} \cdot \max\{n^{-1}, \tilde{\delta}_{\text{cen}}^i\}), \tag{13}$$

where $\kappa^\rho$ captures the conditioning of the DRE problem.

Assumption 1 ensures that DRE does not become a bottleneck for achieving affinity-based speedup. Our analysis incorporates asynchronous DRE through Assumption 1; see Appendix G.

## C.5 NOISE ALIGNMENT

We first establish the trivial bound of the stochastic condition number (noise alignment constant) defined in Definition 3. In this subsection, we omit the agent index $i$ for simplicity and generality. For a general stochastic matrix $A(s)$, recall its stochastic condition number is

$$\nu := \|\bar{D}\bar{A}^{-1}\|,$$

where $\bar{A} = \mathbb{E}A(s)$, $\bar{D} = \mathbb{E}D(s)$, and $D(s) = \sqrt{A(s)^T A(s)}$. For the upper bound, we have

$$\nu \leq \|\bar{D}\|\|\bar{A}^{-1}\|.$$

The "numerator" satisfies

$$\|\bar{D}\| = \|\mathbb{E}D(s)\| \leq \mathbb{E}\|D(s)\| = \mathbb{E}\sigma_{\max}(D(s)) = \mathbb{E}\sigma_{\max}(A(s)) \leq G_A,$$

where we use the polar decomposition $A(s) = U(s)D(s)$, where $U(s)$ is an orthogonal matrix, and thus $A(s)$ and $D(s)$ share the same singular values. The "denominator" satisfies $\|\bar{A}^{-1}\| = \sigma_{\min}^{-1}(\bar{A})$, and we have

$$\sigma_{\min}(\bar{A}) = \min_{\|x\|=1} \|\bar{A}x\| = \min_{\|x\|=1} \|\bar{A}x\|\|x\| \geq \min_{\|x\|=1} x^T \bar{A}x = \min_{\|x\|=1} x^T \operatorname{sym}(\bar{A})x = \lambda_{\min}(\operatorname{sym}(\bar{A})). \tag{14}$$

Note that $\lambda \geq \lambda_{\min}(\operatorname{sym}(\bar{A}))$ and $G_A \leq \sigma$. Thus,

$$\nu \leq \sigma/\lambda = \kappa.$$

To illustrate the idea that $\nu^{-1}$ measures the alignment of noise in $A(s)$, we describe one example where $\nu^{-1}$ is small.

**Example 3** (Misaligned noise). Suppose $\mathcal{S} = [0, 2\pi - \epsilon] \subset \mathbb{R}$ and $A(s) = U(s)I \in \mathbb{R}^{2\times2}$, where $U(s)$ is a rotation matrix with angle $s$. Then, $D(s) = I$, $\bar{D} = I$, and

$$\bar{A} = \int_0^{2\pi-\epsilon} \begin{pmatrix} \cos s & -\sin s \\ \sin s & \cos s \end{pmatrix} \mathrm{d}s = \begin{pmatrix} -\sin\epsilon & \cos\epsilon - 1 \\ 1 - \cos\epsilon & -\sin\epsilon \end{pmatrix} = 2\sin\tfrac{\epsilon}{2} \begin{pmatrix} -\cos\tfrac{\epsilon}{2} & -\sin\tfrac{\epsilon}{2} \\ \sin\tfrac{\epsilon}{2} & -\cos\tfrac{\epsilon}{2} \end{pmatrix},$$

whose smallest singular value is $2\sin\tfrac{\epsilon}{2} \approx \epsilon$ when $\epsilon > 0$ is small. Thus, $\nu \to \infty$ and $\nu^{-1} \to 0$ as $\epsilon \to 0$. This is an example where the orientation of $A(s)$ is uniformly random, and thus the noise is completely misaligned.

We then give several examples where the noise is well-aligned and the stochastic condition number $\nu$ equals or is close to 1.

**Example 4** (Constant orientation). Suppose $A(s) = UD(s)$ for all $s \in \mathcal{S}$, where $U$ is a constant orthogonal matrix and $D(s) \succeq 0$. Then, $\bar{A} = U\bar{D}$ and $\nu = 1$.

**Example 5** (Positive semi-definite matrix). Suppose $A(s) \succeq 0$ for all $s \in \mathcal{S}$. Then, $D(s) = A(s)$ and $\nu = 1$.

**Example 6** (Low rank feature embedding). Suppose the feature embedding matrix $A(s)$ has a low-rank structure: $A(s) = (\phi(s) - \gamma\psi(s))\phi^T(s)$, where $\phi(s), \psi(s) \in \mathbb{R}^d$ are two normalized feature maps such that $\|\phi(s)\| = \|\psi(s)\| = 1$ and $\phi(s) \stackrel{d}{=} \psi(s)$ for all $s \in \mathcal{S}$. This is the case of temporal difference learning with linear function approximation (Bhandari et al., 2018), where $\gamma$ is the reward discount factor. Then, $\nu \leq \frac{1+\gamma}{1-\gamma}$.

*Proof.* We have

$$\begin{aligned} D(s)^2 &= \phi(s)\phi(s)^T \left(\phi(s)^T\phi(s) - 2\gamma\phi(s)^T\psi(s) + \gamma^2\psi(s)^T\psi(s)\right) \\ &= \phi(s)\phi(s)^T(\phi(s) - \gamma\psi(s))^T(\phi(s) - \gamma\psi(s)), \end{aligned}$$

which implies

$$D(s) = \frac{\|\phi(s) - \gamma\psi(s)\|}{\|\phi(s)\|}\phi(s)\phi(s)^T \preceq (1+\gamma)\phi(s)\phi(s)^T.$$

On the other hand,

$$A(s) \succeq \phi(s)\phi(s)^T - \frac{\gamma}{2}(\phi(s)\phi(s)^T + \psi(s)\psi(s)^T) \stackrel{d}{=} (1-\gamma)\phi(s)\phi(s)^T.$$

Thus,

$$\bar{A} \succeq (1-\gamma)\mathbb{E}[\phi(s)\phi(s)^T] \succeq \frac{1-\gamma}{1+\gamma}\bar{D},$$

which gives

$$\bar{D}\bar{A}^{-1} \preceq \frac{1+\gamma}{1-\gamma}I.$$

Thus, $\nu \leq \frac{1+\gamma}{1-\gamma}$. $\qquad\square$

*Remark* 2. While strict normalization $\|\phi(s)\| = \|\psi(s)\| = 1$ is required to show $\nu \leq \frac{1+\gamma}{1-\gamma}$, Lemma D.2, the only place where $\nu$ is directly used, also holds with $\|\phi(s)\|, \|\psi(s)\| \leq 1$. That is, we define $\nu$ for interpretability and ease of calculation, but it can be relaxed in certain cases.

**Example 7** (Multiplicative noise). Suppose the noise is multiplicative: $A(s) = (I + U(s))\bar{A}$, where $U(s)$ is zero-mean and $\|U(s)\| \le \epsilon$ for all $s \in \mathcal{S}$. Then, $\nu \le 1 + \epsilon$.

*Proof.* We have

$$D(s)^2 = \bar{A}^T (I + U(s))^T (I + U(s)) \bar{A} \preceq (1 + \epsilon)^2 \bar{A}^T \bar{A}$$
$$\implies \bar{A}^{-T} D(s)^2 \bar{A}^{-1} \preceq (1 + \epsilon)^2 I$$
$$\implies \nu = \|\mathbb{E} D(s) \bar{A}^{-1}\| \le 1 + \epsilon,$$

$\square$

Finally, we refer readers to Lemma D.2, which illustrates how noise alignment affects the translation of the raw affinity into an *effective* affinity for variance reduction. Intuitively, when $\bar{A}$ is ill-conditioned, a small perturbation in the objective can lead to a large perturbation in the solution. Then, if $A(s)$ does not align well with $\bar{A}$, the large perturbation in the solution may be further amplified by the large eigenvalues of $A(s)$, leading to a large variance in the update direction. On the other hand, when $A(s)$ aligns well with $\bar{A}$, the perturbation in the solution is only stretched by the small eigenvalues in $A(s)$, maintaining a similar magnitude to the objective perturbation and preventing noise amplification in the ill-conditioned subspace.

### C.6 APPLICATION TO REINFORCEMENT LEARNING

This subsection gives a concrete application of our method to the policy evaluation problem in reinforcement learning (RL) resulting in personalized collaborative temporal difference (TD) learning.

Heterogeneous federated RL has garnered traction recently (Zhang et al., 2024; Wang et al., 2024; Xiong et al., 2025; Li & Azizan, 2024) due to its practicality by accommodating heterogeneity in multi-agent decision-making. However, existing works either fail to personalize and hence only work well in low heterogeneity regimes (Wang et al., 2024; Zhang et al., 2024), or deliver slower convergence rates (Xiong et al., 2025). Our framework encompasses the setting of heterogeneous federated RL and our method provides the first personalized collaborative reinforcement learning algorithm that accommodates arbitrary heterogeneous agents while achieving affinity-based variance reduction.

Consider $n$ agents with distinct Markov reward processes $(\mathcal{O}, P^i, R^i, \gamma)$, where $\mathcal{O}$ is the state space, $P^i$ is the transition kernel induced by agent $i$'s behavior policy, $R^i : \mathcal{O} \times \mathcal{O} \to \mathbb{R}$ is the reward function, and $\gamma \in [0, 1)$ is the discount factor. Following (Bhandari et al., 2018), we write $R^i(o) = \mathbb{E}[R^i(o_h^i, o_{h+1}^i) \mid o_h^i = o]$. Agents want to evaluate their behavior policies by calculating their infinite horizon value functions $V^i(s) = \mathbb{E}[\sum_{h=0}^{\infty} \gamma^h R^i(o_h^i) \mid o_0^i = 0]$, where $o_{h+1}^i \sim P^i(\cdot \mid o_h^i)$. With a linear function approximation $V^i(o) \approx \phi(o)^T x_*^i$ for some $x_*^i \in \mathbb{R}^d$, the expected projected Bellman equation can be cast into (2) as

$$\mathbb{E}^i[\underbrace{\phi(s)(\phi(s) - \gamma \phi(s'))^T}_{A(s,s')}] x_*^i = \mathbb{E}^i[\underbrace{\phi(s) R^i(s, s')}_{b^i(s,s')}], \qquad (15)$$

where $\mathbb{E}^i = \mathbb{E}_{s \sim \mu^i, s' \sim P^i(\cdot \mid s)}$. The stochastic residual of (15) is the TD error, and the corresponding fixed point iteration gives the TD(0) algorithm. Specifically within our framework, each observation tuple is $s_t^i = (o_{h_t}^i, o_{h_t+1}^i)$,[5] $A(s_t^i) = \phi(o_{h_t}^i)(\phi(o_{h_t}^i)^T - \gamma \phi(o_{h_t+1}^i)^T)$, $b^i(s_t^i) = \phi(o_{h_t}^i) R^i(o_{h_t}^i, o_{h_t+1}^i)$, and the environment distribution is $\mu^i(o, o') = \pi^i(o) \times P^i(o' \mid o)$, where $\pi^i$ is the stationary distribution of the agent $i$'s transition kernel. Then $g_t^i(x_t^i)$ represents the TD error and AffPCL (8) gives personalized collaborative TD(0).

With a normalized feature map $\|\phi(o)\| \le 1$ and constants $G_x \ge \max\{\max_{i \in [n]} \|x_*^i\|, \|x^c\|\}$, $G_b \ge \max_{i \in [n]} \|R^i\|_\infty$, we have $G_A \le 1 + \gamma \le 2$ and $\sigma \le 2 \max\{(1 + \gamma) G_x, G_b\} \lesssim G_x + G_b$. As shown in Bhandari et al. (2018, Lemma 3), $\lambda^i \ge (1 - \gamma) \lambda_{\min}(\mathbb{E}_{\pi^i}[\phi(s)\phi(s)^T])$. As shown in

---

[5]Here we assume an offline RL setting where we have i.i.d. samples from a pre-collected dataset consisting of observation tuples $(o_h^i, o_{h+1}^i \sim P^i(\cdot \mid o_h^i), R^i(o_h^i, o_{h+1}^i))$.

Example 6, the stochastic condition number in this example is $\nu \leq \frac{1+\gamma}{1-\gamma} \leq 2(1-\gamma)^{-1}$. Therefore, by Theorem 1, the sample complexity of personalized collaborative TD(0) reads

$$O\left(\frac{(G_x + G_b)^2}{(1-\gamma)^3 (w^i)^2 t} \cdot \max\{n^{-1}, \delta_{\text{env}}, \delta_{\text{obj}}\}\right),$$

where $w^i := \lambda_{\min}(\mathbb{E}_{\pi^i}[\phi(s)\phi(s)^T])$ and $\delta_{\text{env}}, \delta_{\text{obj}}$ represent kernel and reward heterogeneity levels, respectively. This complexity matches the best known result for homogeneous federated TD learning (Wang et al., 2024), while offering new insights in high heterogeneity regimes.

## D PRELIMINARY LEMMAS

**Lemma D.1** (Affinity). *Given the universal scores* $\delta_{\text{env}} = \max_{i,j} \|\mu^i - \mu^j\|_{\text{TV}}$ *and* $\delta_{\text{obj}} := \max_{i,j} \|\theta_s^i - \theta_s^j\|_2/(2G_b)$, *along with the agent-specific scores* $\delta_{\text{env}}^i = \|\mu^i - \mu^0\|_{\text{TV}}$ *and* $\delta_{\text{obj}}^i = \|\theta_s^i - \theta_s^0\|/(2G_b)$, *we establish bounds on various parameter differences in terms of these scores.*

(a) $\|b^i(s) - b^j(s)\| \leq 2G_b \delta_{\text{obj}}$, *for any* $i, j \in [n^0]$.

(b) $\|b^i(s) - b^0(s)\| \leq 2G_b \delta_{\text{obj}}^i$, *for any* $i \in [n]$.

(c) $\|\bar{b}^i - \bar{b}^j\| \leq 2G_b(\delta_{\text{env}} + \delta_{\text{obj}})$, *for any* $i, j \in [n^0]$.

(d) $\|\bar{b}^i - \bar{b}^0\| \leq 2G_b \min\{1, \delta_{\text{env}}^i + \delta_{\text{obj}}, \delta_{\text{env}} + \delta_{\text{obj}}^i\}$, *for any* $i \in [n]$.
   *We thus define* $\delta_{\text{cen}}^i := \max\{\delta_{\text{env}}^i, \|\bar{b}^i - \bar{b}^0\|/(2G_b)\} \leq \min\{1, \delta_{\text{env}}^i + \delta_{\text{obj}}, \delta_{\text{env}} + \delta_{\text{obj}}^i\}$.

(e) $\|\mathbb{E}_{\mu^j}[b^i(s) - b^c(s)]\| \leq 2\sigma\delta_{\text{cen}}^i$, *where* $j \in \{0, i\}$, *for any* $i \in [n]$.

(f) *(Naive)* $\|\theta_*^i - \theta_*^c\| \leq 2\lambda^{-1}\sigma(\delta_{\text{env}} + \delta_{\text{obj}})$, *for any* $i \in [n]$.

(g) $\|\bar{A}^i - \bar{A}^j\| \leq 2G_A \delta_{\text{env}}$, *for any* $i, j \in [n^0]$.

(h) $\|\bar{A}^i - \bar{A}^0\| \leq 2G_A \delta_{\text{env}}^i$, *for any* $i \in [n]$.

(i) $\|\mathbb{E}_{\mu^j}[A(s)(x_*^i - x_*^c)]\| \leq 2\sigma\delta_{\text{cen}}^i$, *where* $j \in \{0, i\}$, *for any* $i \in [n]$.

(j) *(Naive)* $\|x_*^i - x_*^c\| \leq 2\lambda^{-1}\sigma(\delta_{\text{env}} + \delta_{\text{obj}})$, *for any* $i \in [n]$.

*Proof.* For Item (a), by the linear parametrization of the objective,

$$\|b^i(s) - b^j(s)\| = \|\Phi(s)(\theta_*^i - \theta_*^j)\| \leq \|\theta_*^i - \theta_*^j\| \leq 2G_b\delta_{\text{obj}}, \tag{16}$$

where we use the fact that $\|\Phi(s)\| \leq 1$ and the definition of $\delta_{\text{obj}}$ in Section 4. Item (b) follows from the same argument with agent-specific score $\delta_{\text{obj}}^i$ used.

For any function $f$ such that $\|f(s)\| \leq G_f$ for all $s \in \mathcal{S}$, and for all $i \in [n]$, by Definition 2,

$$\begin{aligned}
\|\mathbb{E}_{\mu^i} f(s) - \mathbb{E}_{\mu^j} f(s)\| &= \left\|\int_{\mathcal{S}} f(s)(\mu^i(s) - \mu^j(s))\mathrm{d}s\right\| \\
&\leq 2G_f \|\mu^i - \mu^j\|_{\text{TV}} \\
&\leq \begin{cases} 2G_f\delta_{\text{env}}, & j \in [n] \\ 2G_f\delta_{\text{env}}^i, & j = 0 \end{cases}.
\end{aligned} \tag{17}$$

This bound first gives Item (g) and Item (h) by letting $f(s) = A(s)$ and $G_f = G_A$.

Then, combining (16) and (17) with $f(s) = b^i(s)$ and $G_f = G_b$ gives Item (c):

$$\|\bar{b}^i - \bar{b}^j\| = \|\mathbb{E}^i b^i(s) - \mathbb{E}^i b^j(s) + \mathbb{E}^i b^j(s) - \mathbb{E}^j b^j(s)\| \leq 2G_b\delta_{\text{obj}} + 2G_b\delta_{\text{env}} = 2G_b(\delta_{\text{obj}} + \delta_{\text{env}}).$$

Specifically for the difference between the personalized and central expected objectives, we have

$$\|\bar{b}^i - \bar{b}^0\| = \left\| \mathbb{E}^i b^i(s) - \frac{1}{n} \sum_{j=1}^n \mathbb{E}^j b^j(s) \right\|$$

$$= \left\| \mathbb{E}^i b^i(s) - \frac{1}{n} \sum_{j=1}^n \left( \mathbb{E}^i b^j(s) - \mathbb{E}^i b^j(s) + \mathbb{E}^j b^j(s) \right) \right\|$$

$$\leq \|\mathbb{E}^i [b^i(s) - b^0(s)]\| + \frac{1}{n} \sum_{j=1}^n \|(\mathbb{E}^i - \mathbb{E}^j)[b^j(s)]\|$$

$$\leq 2 G_b \delta^i_{\text{obj}} + 2 G_b \delta_{\text{env}}.$$

Similarly, we have

$$\|\bar{b}^i - \bar{b}^0\| = \left\| \mathbb{E}^i b^i(s) - \frac{1}{n} \sum_{j=1}^n \left( \mathbb{E}^j b^i(s) - \mathbb{E}^j b^i(s) + \mathbb{E}^j b^j(s) \right) \right\|$$

$$\leq \|(\mathbb{E}^i - \mathbb{E}^0)[b^i(s)]\| + \frac{1}{n} \sum_{j=1}^n \|\mathbb{E}^j [b^i(s) - b^j(s)]\|$$

$$\leq 2 G_b \delta_{\text{obj}} + 2 G_b \delta^i_{\text{env}}.$$

The above two bounds give Item (d).

We then look at the naive bounds Items (f) and (j) on the difference between optimal solutions. For any $i, j \in [n]$, we have

$$\bar{A}^i(x^i_* - x^j_*) + (\bar{A}^i - \bar{A}^j)x^j_* - (\bar{b}^i - \bar{b}^j) = 0,$$

which gives

$$\|x^i_* - x^j_*\|_2 \leq \|(\bar{A}^i)^{-1}\|_2 \left( \|\bar{A}^i - \bar{A}^j\|_2 \|x^j_*\|_2 + \|\bar{b}^i - \bar{b}^j\|_2 \right).$$

Combining the previous bounds on system parameter differences gives

$$\|x^i_* - x^j_*\|_2 \leq \sigma^{-1}_{\min}(\bar{A}^i)(2 G_A \delta_{\text{env}} \cdot G_x + 2 G_b (\delta_{\text{obj}} + \delta_{\text{env}})) \leq \sigma^{-1}_{\min}(\bar{A}^i) \cdot 2\sigma(\delta_{\text{obj}} + \delta_{\text{env}}).$$

The above bound also holds for the difference between the personalized solution and the central solution satisfying $\bar{A}^0 x^c_* = \bar{b}^0$. Specifically, the same argument gives

$$\|x^i_* - x^c_*\|_2 \leq \sigma^{-1}_{\min}(\bar{A}^0) \cdot 2\sigma(\delta_{\text{obj}} + \delta_{\text{env}}).$$

Let $\lambda := \min_{i \in [n]} \min\{\lambda_{\min}(\text{sym}(\bar{A}^i)), \lambda_{\min}(\text{sym}(\bar{\Phi}^i))\}$; (14) gives Item (j), and a similar argument gives Item (f). Notably, the upper bound of the optimal solution difference scales with $\lambda^{-1}$, which can be large when $\bar{A}^i$ is ill-conditioned. This indicates that the affinity in objectives or environments do not translate well to the affinity in optimal solutions.

Fortunately, the bound is tamer when the optimal solutions are left-applied by $\bar{A}^i$:

$$\|\bar{A}^i(x^i_* - x^c_*)\|_2 = \|\bar{A}^i x^i_* - \bar{A}^0 x^c_* + (\bar{A}^i - \bar{A}^0)x^c_*\|_2 = \|\bar{b}^i - \bar{b}^0 + (\bar{A}^i - \bar{A}^0)x^c_*\|_2$$
$$\leq 2 G_b \delta^i_{\text{cen}} + 2 G_A \delta^i_{\text{env}} \cdot G_x \leq 2\sigma \delta^i_{\text{cen}}.$$

Left-applying $\bar{A}^0$ gives the same bound, thus giving Item (i). Item (e) can be derived similarly. Items (e) and (i) are saying that the affinity is well-preserved in the *feature space*, i.e., the image of the feature embedding matrix. This is also a key to our analysis: we will never directly bound the difference between optimal solutions, but always inspect them in the feature space. □

**Lemma D.2** (Effective affinity). *Denote $\tilde{\delta}^i_{\text{cen}} = \min\left\{1, \nu \delta^i_{\text{cen}}\right\}$. Then,*

$$\mathbb{E}_{\mu^i} \|A(s)(x^i_* - x^c_*)\|^2 \leq 2\sigma^2 \tilde{\delta}^i_{\text{cen}}, \quad \forall i \in [n].$$
$$\mathbb{E}_{\mu^i} \|\Phi(s)(\theta^i_* - \theta^c_*)\|^2 \leq 2\sigma^2 \tilde{\delta}^i_{\text{cen}},$$

*Proof.* We only prove the first inequality; the second one can be proved similarly. We have

$$
\begin{aligned}
\mathbb{E}^i \|A(s)(x_*^i - x_*^c)\|^2 &= (x_*^i - x_*^c)^T \mathbb{E}^i [A(s)^T A(s)](x_*^i - x_*^c) \\
&= (x_*^i - x_*^c)^T \mathbb{E}^i [D(s)^2](x_*^i - x_*^c) \\
&\leq \|x_*^i - x_*^c\| \|D(s)\|_\infty \|\mathbb{E}^i [D(s)(x_*^i - x_*^c)]\| \\
&\leq 2 G_A G_x \|\bar{D}^i (x_*^i - x_*^c)\| \\
&= 2 G_A G_x \|\bar{D}^i (\bar{A}^i)^{-1} \bar{A}^i (x_*^i - x_*^c)\| \\
&\leq 2 G_A G_x \nu \|\bar{A}^i (x_*^i - x_*^c)\| \\
&\leq 2 G_A G_x \nu \cdot 2 \sigma \delta_{\text{cen}}^i,
\end{aligned}
$$

where the last inequality follows from Item (i) in Lemma D.1. On the other hand, by the trivial bound, we have

$$
\mathbb{E}^i \|A(s)(x_*^i - x_*^c)\|^2 \leq (2 G_A G_x)^2.
$$

Combining the two bounds gives

$$
\mathbb{E}^i \|A(s)(x_*^i - x_*^c)\|^2 \leq 2\sigma^2 \min \left\{ 1, \nu \delta_{\text{cen}}^i \right\} \leq 2\sigma^2 \tilde{\delta}_{\text{cen}}^i.
$$

$\square$

**Lemma D.3.** *For any two distributions $\mu, \mu'$ over $\mathcal{S}$, we have*
$$
\chi^2(\mu, \mu') \leq \max \left\{ \|\mu/\mu'\|_\infty, 1 \right\} \|\mu - \mu'\|_{\text{TV}}.
$$

*Proof.* Let $\mathcal{S}_0 := \{s \in \mathcal{S} : \mu(s) \geq \mu'(s)\}$. We have

$$
\begin{aligned}
\chi^2(\mu, \mu') &= \int_{\mathcal{S}} \left( 1 - \frac{\mu(s)}{\mu'(s)} \right)^2 \mu'(s) \, ds \\
&= \left( \int_{\mathcal{S}_0} + \int_{\mathcal{S}_0^c} \right) \left( 1 - \frac{\mu(s)}{\mu'(s)} \right)^2 \mu'(s) \, ds \\
&\leq \int_{\mathcal{S}_0} \underbrace{\left( \frac{\mu(s)}{\mu'(s)} - 1 \right)}_{\geq 0} \underbrace{(\mu(s) - \mu'(s))}_{\geq 0} \, ds + \int_{\mathcal{S}_0^c} \underbrace{\left( 1 - \frac{\mu(s)}{\mu'(s)} \right)}_{\geq 0, \leq 1} \underbrace{(\mu'(s) - \mu(s))}_{\geq 0} \, ds \\
&\leq (\max \left\{ \|\mu/\mu'\|_\infty, 1 \right\} - 1) \|\mu - \mu'\|_{\text{TV}} + \|\mu - \mu'\|_{\text{TV}} \\
&= \max \left\{ \|\mu/\mu'\|_\infty, 1 \right\} \|\mu - \mu'\|_{\text{TV}}.
\end{aligned}
$$

$\square$

**Lemma D.4** (Uni-timescale Lyapunov analysis for asynchronous learning). *Consider asynchronous learning of multiple decision variables $z_t^k$, $k \in [K]$, which satisfies the following one-step contraction:*

$$
\mathbb{E}\|\Delta z_t^k\|^2 \leq (1 - \tfrac{3}{2}\alpha_t^k \lambda^k)\mathbb{E}\|\Delta z_t^k\|^2 + (\alpha_t^k)^2 C^k + \alpha_t^k \sum_{k'=1}^{k-1} C^{k,k'} \mathbb{E}\|\Delta z_t^{k'}\|^2, \quad k = 1, \dots, K.
$$

*That is, the convergence of $z_t^k$ also depends on the other decision variables $z_t^{k'}$, $k' < k$. $\alpha_t^k$ is the step size for $z_t^k$; we set them using a unified effective step size $\alpha_t$:*

$$
\alpha_t^k \lambda^k = \alpha_t < 1, \quad k = 1, \dots, K.
$$

*Let*

$$
w^K = 1, \quad w^k = 2 \sum_{k'=k+1}^K w^{k'} C^{k',k} (\lambda^{k'})^{-1}, \quad k = 1, \dots, K-1.
$$

*Consider the following overall Lyapunov function:*

$$
\mathcal{L}_t = \sum_{k=1}^K w^k \mathbb{E}\|\Delta z_t^k\|^2.
$$

*Then, we have*

$$
\mathcal{L}_{t+1} \leq (1 - \alpha_t)\mathcal{L}_t + \alpha_t^2 \sum_{k=1}^K w^k C^k (\lambda^k)^{-2}.
$$

*Proof.* By definition, we have

$$\mathcal{L}_{t+1} = \sum_{k=1}^{K} w^k \mathbb{E}\|\Delta z_{t+1}^k\|^2$$

$$\leq \sum_{k=1}^{K} w^k \left( (1 - \tfrac{3}{2}\alpha_t^k \lambda^k)\mathbb{E}\|\Delta z_t^k\|^2 + (\alpha_t^k)^2 C^k + \alpha_t^k \sum_{k'=1}^{k-1} C^{k,k'} \mathbb{E}\|\Delta z_t^{k'}\|^2 \right)$$

$$= \sum_{k=1}^{K} \left( \left( w^k(1 - \tfrac{3}{2}\alpha_t^k \lambda^k) + \sum_{k'=k+1}^{K} w^{k'}\alpha_t^{k'} C^{k',k} \right) \mathbb{E}\|\Delta z_t^k\|^2 + w^k(\alpha_t^k)^2 C^k \right) \quad (18)$$

$$= \sum_{k=1}^{K} \left( \left( w^k(1 - \tfrac{3}{2}\alpha_t) + \tfrac{1}{2}w^k\alpha_t \mathbb{1}_{\{k<K\}} \right) \mathbb{E}\|\Delta z_t^k\|^2 + w^k(\alpha_t^k)^2 C^k \right) \quad (19)$$

$$\leq \sum_{k=1}^{K} \left( (1 - \alpha_t) w^k \mathbb{E}\|\Delta z_t^k\|^2 + w^k(\alpha_t^k)^2 C^k \right)$$

$$= (1 - \alpha_t)\mathcal{L}_t + \alpha_t^2 \sum_{k=1}^{K} w^k C^k (\lambda^k)^{-2},$$

where (18) follows from rearranging the summation and grouping the coefficients of $\mathbb{E}\|\Delta z_t^k\|^2$, and (19) follows from the definition of $w^k$ and $\alpha_t^k$. $\qquad\square$

**Lemma D.5** (Constant and diminishing step size)**.** *Suppose we have the following one-step contraction:*

$$\mathcal{L}_{t+1} \leq (1 - \alpha_t)\mathcal{L}_t + \alpha_t^2 C.$$

*Then, with a constant step size $\alpha = \ln t/t$, we have*

$$\mathcal{L}_t = O\left( \frac{C \ln t}{t} \right).$$

*With a linearly diminishing step size $\alpha_\tau = 4/((\tau + t_0 + 1))$, $\tau = 0, \ldots, t$, the following convex combination*

$$\tilde{\mathcal{L}}_t = \sum_{\tau=0}^{t} \frac{\tau + t_0}{\sum_{\tau=0}^{t}(\tau + t_0)} \mathcal{L}_\tau$$

*satisfies*

$$\tilde{\mathcal{L}}_t = O\left( \frac{C}{t} \right).$$

*Proof.* With a constant step size $\alpha = \ln t/t$, telescoping the one-step contraction gives

$$\mathcal{L}_t \leq (1 - \alpha)^t \mathcal{L}_0 + \alpha^{-1} \cdot \alpha^2 C \leq e^{-\alpha t}\mathcal{L}_0 + \alpha C = \frac{\mathcal{L}_0 + C \ln t}{t} = O\left( \frac{C \ln t}{t} \right).$$

With a linearly diminishing step size $\alpha_\tau = 4/((\tau + t_0 + 1))$, $\tau = 0, \ldots, t$, the one-step contraction first gives

$$\tfrac{1}{2}\mathcal{L}_\tau \leq \left( \frac{1}{\alpha_\tau} - \frac{1}{2} \right)\mathcal{L}_\tau - \frac{1}{\alpha_\tau}\mathcal{L}_{\tau+1} + \alpha_\tau C = \frac{t_0 + \tau - 1}{4}\mathcal{L}_\tau - \frac{t_0 + \tau + 1}{4}\mathcal{L}_{\tau+1} + \frac{4C}{t_0 + \tau + 1}.$$

Thus, the convex combination satisfies

$$
\begin{aligned}
\tilde{\mathcal{L}}_t =& \frac{2}{(t+1)(t+2t_0)} \sum_{\tau=0}^{t} (\tau+t_0) \left( \frac{t_0+\tau-1}{4}\mathcal{L}_\tau - \frac{t_0+\tau+1}{4}\mathcal{L}_{\tau+1} + \frac{4C}{t_0+\tau+1} \right) \\
=& \frac{1}{2(t+1)(t+2t_0)} \sum_{\tau=0}^{t} \left( (t_0+\tau-1)(t_0+\tau)\mathcal{L}_\tau - (t_0+\tau)(t_0+\tau+1)\mathcal{L}_{\tau+1} \right) \\
& + \frac{8C}{(t+1)(t+2t_0)} \sum_{\tau=0}^{t} \frac{t_0+\tau}{t_0+\tau+1} \\
=& \frac{1}{2(t+1)(t+2t_0)} \left( (t_0-1)t_0\mathcal{L}_0 - (t_0+t)(t_0+t+1)\mathcal{L}_{t+1} \right) \\
& + \frac{8C}{(t+1)(t+2t_0)} \sum_{\tau=0}^{t} \frac{t_0+\tau}{t_0+\tau+1} \\
\leq& \frac{t_0^2\mathcal{L}_0}{2t^2} + \frac{8Ct}{t^2} \\
=& O\left( \frac{C}{t} \right).
\end{aligned}
$$

The convex combination removes the logarithmic dependence, and the $t_0$ dependency diminishes quadratically. $\qquad\square$

## E ANALYSIS OF CENTRAL OBJECTIVE ESTIMATION

This section directly considers central objective estimation (COE) with environment heterogeneity in Section 5, which covers Section 4 as a special case. We restate the learning problem in (6):

$$
\bar{\Phi}^0 \theta_*^c = \bar{b}^0,
$$

where $\bar{\Phi}^0 = \mathbb{E}_{\mu^0}[\Phi(s)]$, $\mu^0 = \frac{1}{n}\sum_{i=1}^n \mu^i$, and $\bar{b}^0 = \frac{1}{n}\sum_{i=1}^n \mathbb{E}_{\mu^i} b^i$. Recall that $\|\Phi(s)\|_2 \leq 1$ for all $s \in \mathcal{S}$. The COE algorithm is

$$
\theta_{t+1}^c = \theta_t^c - \alpha_t^b g_t^{0,b}(\theta_t^c), \tag{20}
$$

where

$$
g_t^{0,b}(\theta_t^c) = \frac{1}{n}\sum_{i=1}^n g_t^{i,b}(\theta_t^c), \quad g_t^{i,b}(\theta_t^c) = \Phi(s_t^i)\theta_t^c - b_t^i.
$$

The additional superscript $b$ distinguishes the objective estimation parameters from other learning modules. We denote $\Delta\theta_t^c = \theta_t^c - \theta_*^c$.

The one-step mean squared error (MSE) dynamics of (20) can be decomposed as

$$
\mathbb{E}\|\Delta\theta_{t+1}^c\|^2 = \mathbb{E}\|\Delta\theta_t^c\|^2 - 2\alpha_t^b\mathbb{E}\langle g_t^{0,b}(\theta_t^c), \Delta\theta_t^c\rangle + (\alpha_t^b)^2\mathbb{E}\|g_t^{0,b}(\theta_t^c)\|^2. \tag{21}
$$

We first analyze the cross term, then the variance term, and finally combine them to give the one-step progress. The analysis of other learning modules follows a similar pattern.

**Lemma E.1** (COE descent)**.** *Let $\lambda^b := \lambda_{\min}(\mathrm{sym}(\bar{\Phi}^0))$. The cross term in (21) satisfies*

$$
\mathbb{E}\langle \Delta\theta_t^c, g_t^{0,b}(\theta_t^c)\rangle \geq \lambda^b\mathbb{E}\|\Delta\theta_t^c\|^2.
$$

*Proof.* We use the following shorthand notation: $\mathbb{E}_t := \mathbb{E}_{s_t^j \sim \mu^j, j \in [n]}$, $\mathbb{E}_t^i := \mathbb{E}_{s_t^i \sim \mu^i}$, and $\mathbb{E}_{\mathcal{F}_{t-1}} := \mathbb{E}[\cdot \mid \mathcal{F}_{t-1}]$, where $\mathcal{F}_{t-1}$ is the history filtration up to time step $t-1$. The cross term satisfies

$$
\begin{aligned}
\mathbb{E}\langle \Delta\theta_t^c, g_t^{0,b}(\theta_t^c) \rangle &= \mathbb{E}_{\mathcal{F}_{t-1}}\left[ \left\langle \mathbb{E}_t[g_t^{0,b}(\theta_t^c)], \Delta\theta_t^c \right\rangle \right] \\
&= \mathbb{E}_{\mathcal{F}_{t-1}}\left[ \left\langle \frac{1}{n}\sum_{i=1}^n \mathbb{E}_t^i[\Phi(s)\theta_t^c - b^i(s)], \Delta\theta_t^c \right\rangle \right] \\
&= \mathbb{E}_{\mathcal{F}_{t-1}}\left[ \left\langle \mathbb{E}_{\mu^0}[\Phi(s)]\theta_t^c - \frac{1}{n}\sum_{i=1}^n \mathbb{E}_{\mu^i}[b^i(s)], \Delta\theta_t^c \right\rangle \right] \\
&= \mathbb{E}_{\mathcal{F}_{t-1}}\left[ \langle \bar{\Phi}^0\theta_t^c - \bar{b}^0, \Delta\theta_t^c \rangle \right].
\end{aligned}
$$

Note that the solution $\theta_*^c$ satisfies $\bar{\Phi}^0\theta_*^c - \bar{b}^0 = 0$. Thus,

$$
\begin{aligned}
\mathbb{E}\langle \Delta\theta_t^c, g_t^{0,b}(\theta_t^c) \rangle &= \mathbb{E}_{\mathcal{F}_{t-1}}\left[ \langle (\bar{\Phi}^0\theta_t^c - \bar{b}^0) - (\bar{\Phi}^0\theta_*^c - \bar{b}^0), \Delta\theta_t^c \rangle \right] \\
&= \mathbb{E}_{\mathcal{F}_{t-1}}\left[ \langle \bar{\Phi}^0\Delta\theta_t^c, \Delta\theta_t^c \rangle \right] \\
&\geq \lambda_{\min}(\mathrm{sym}(\bar{\Phi}^0)) \mathbb{E}\|\Delta\theta_t^c\|^2.
\end{aligned}
$$

$\square$

**Lemma E.2** (COE variance). *The variance term in (21) satisfies*

$$
\mathbb{E}\|g_t^{0,b}(\theta_t^c)\|^2 \leq 2\mathbb{E}\|\Delta\theta_t^c\|^2 + 2\sigma^2 n^{-1}.
$$

*Proof.* The variance term can be first decomposed as

$$
\mathbb{E}\|g_t^{0,b}(\theta_t^c)\|^2 = \mathbb{E}\|\tfrac{1}{n}\sum_{i=1}^n \Phi_t^i(\theta_t^c - \theta_*^c) + g_t^{0,b}(\theta_*^c)\|^2 \leq 2\mathbb{E}\|\Delta\theta_t^c\|^2 + 2\mathbb{E}\|g_t^{0,b}(\theta_*^c)\|^2,
$$

where we use the fact that $\|\Phi_t^i\| \leq 1$. The second term can be further decomposed as

$$
\mathbb{E}\|g_t^{0,b}(\theta_*^c)\|^2 = \underbrace{\frac{1}{n^2}\sum_{i=1}^n \mathbb{E}_t\|g_t^{i,b}(\theta_*^c)\|^2}_{H_1} + \underbrace{\frac{1}{n^2}\sum_{i \neq j}\langle \mathbb{E}_t^i g_t^{i,b}(\theta_*^c), \mathbb{E}_t^j g_t^{j,b}(\theta_*^c) \rangle}_{H_2}.
$$

$H_1$ enjoys linear variance reduction:

$$
H_1 = \frac{1}{n^2}\sum_{i=1}^n \mathbb{E}_t\|\Phi_t^i\theta_*^c - b_t^i\|^2 \leq \frac{1}{n^2}\cdot n(2G_b)^2 \leq \frac{\sigma^2}{n}.
$$

The cross term $H_2$ involves all pairs of independent local update directions. However, since each local update direction in $H_2$ is evaluated at the central solution, its expectation is not zero. One solution is to notice that $g_t^{i,b}$ is Lipschitz continuous in its argument. Thus, we have $\|\mathbb{E}_t^i g_t^{i,b}(\theta_*^c)\| = \|\mathbb{E}_t^i[g_t^{i,b}(\theta_*^c) - g_t^{i,b}(\theta_*^i)]\| = O(\|\theta_*^i - \theta_*^c\|) = O(\delta_{\mathrm{env}} + \delta_{\mathrm{obj}})$. However, this will introduce an affinity-dependent term in the variance. We adopt a more "federated" approach:

$$
\begin{aligned}
H_2 &= \frac{1}{n^2}\sum_{i=1}^n \left\langle \mathbb{E}_t^i g_t^{i,b}(\theta_*^c), \sum_{j=1, j \neq i}^n \mathbb{E}_t^j g_t^{j,b}(\theta_*^c) \right\rangle \\
&= \frac{1}{n^2}\sum_{i=1}^n \left\langle \mathbb{E}_t^i g_t^{i,b}(\theta_*^c), \sum_{j=1}^n \mathbb{E}_t^j g_t^{j,b}(\theta_*^c) - \mathbb{E}_t^i g_t^{i,b}(\theta_*^c) \right\rangle \\
&= \frac{1}{n^2}\left\langle \sum_{i=1}^n \mathbb{E}_t^i g_t^{i,b}(\theta_*^c), \sum_{j=1}^n \mathbb{E}_t^j g_t^{j,b}(\theta_*^c) \right\rangle - \frac{1}{n^2}\sum_{i=1}^n \|\mathbb{E}_t^i g_t^{i,b}(\theta_*^c)\|^2 \\
&= \underbrace{\|\bar{\Phi}^0\theta_*^c - \bar{b}^0\|^2}_{=0} - \underbrace{\frac{1}{n^2}\sum_{i=1}^n \|\mathbb{E}_t^i g_t^{i,b}(\theta_*^c)\|^2}_{=O(n^{-1}), \geq 0} \\
&\leq 0.
\end{aligned}
$$

We see that by analyzing the cross terms collectively, we obtain a much tighter bound that does not depend on the affinity. Plugging the bounds of $H_1$ and $H_2$ back gives the desired result. $\square$

Combining Lemmas E.1 and E.2 with (21) gives the one-step progress of COE.

**Corollary E.1** (COE one-step progress). *Let $\alpha_t^b \leq \lambda^b/4$. Then, for any time step $t$, (20) satisfies*

$$\mathbb{E}\|\Delta\theta_{t+1}^c\|_2^2 \leq (1 - \tfrac{3}{2}\alpha_t^b\lambda^b)\mathbb{E}\|\Delta\theta_t^c\|^2 + 2(\alpha_t^b)^2\sigma^2 n^{-1}.$$

Combining Corollary E.1 and Lemmas D.4 and D.5 gives the convergence guarantee of COE.

**Corollary E.2** (COE convergence). *With a constant step size $\alpha^b = \ln t/(t\lambda^b)$, for any time step $t > 0$, (20) satisfies*

$$\mathbb{E}\|\theta_t^c - \theta_*^c\|^2 = O\left(\frac{\sigma^2\ln t}{(\lambda^b)^2 nt}\right).$$

*With a linearly diminishing step size $\alpha_\tau^b = 4/((\tau + t_0 + 1)\lambda^b)$, $\tau = 0,\ldots,t$, where $t_0 > 0$ ensures that $\alpha_0^b \leq \lambda^b/4$, (20) satisfies*

$$\mathbb{E}\|\tilde{\theta}_t^c - \theta_*^c\|^2 \leq \mathbb{E}\|\tilde{\Delta\theta}_t^c\|^2 = O\left(\frac{\sigma^2}{(\lambda^b)^2 nt}\right),$$

*where $\tilde{f}_t$ represents the convex combination specified in Lemma D.5, and we use Jensen's inequality.*

## F    ANALYSIS OF CENTRAL DECISION LEARNING

This section directly considers central decision learning (CDL) with environment heterogeneity and asynchronous COE (20). CDL without COE is covered as a special case with zero estimation error. We restate the learning problem (3) in Section 5:

$$\bar{A}^0 x_*^c = \bar{b}^0,$$

where $\bar{A}^0 = \frac{1}{n}\sum_{i=1}^n \mathbb{E}_{\mu^i} A(s) = \bar{A}^0$ and $\bar{b}^0 = \frac{1}{n}\sum_{i=1}^n \mathbb{E}_{\mu^i} b^i(s) = \mathbb{E}_{\mu^0} b^c(s)$. We consider two variants of CDL:

$$x_{t+1}^c = x_t^c - \alpha_t^c g_t^{0,c}(x_t^c), \tag{22}$$

where

$$g_t^{0,c}(x_t^c) = \frac{1}{n}\sum_{i=1}^n g_t^i(x_t^c), \quad g_t^{i,c}(x_t^c) = A_t^i x_t^c - b_t^i; \tag{22-1}$$

$$\text{or} \quad g_t^{0,c}(x_t^c; \theta_t^c) = \frac{1}{n}\sum_{i=1}^n g_t^{c\to i}(x_t^c; \theta_t^c), \quad g_t^{c\to i}(x_t^c; \theta_t^c) = A_t^i x_t^c - \hat{b}_t^c(s_t^i). \tag{22-2}$$

The first variant (22-1) corresponds to the CDL algorithm (4) in the main text, where $g_t^{0,c}$ is different from the central update direction used in the personalized local learning module. In the second variant (22-2), $\hat{b}_t^c(s) = \Phi(s)\theta_t^c$ is the estimated central objective function at time step $t$, and we highlight this dependence by including $\theta_t^c$ in the arguments. As remarked in Appendix C.1, $g_t^{0,c}$ in the second variant (22-2) is consistent with the central update direction in the personalized local learning, and thus saves some server-side computation and communication. We will show that both variants enjoy the same convergence rate. The additional superscript $c$ distinguishes the central learning parameters from other learning modules. We denote $\Delta x_t^c = x_t^c - x_*^c$.

The one-step MSE dynamics of (22) can be decomposed as

$$\mathbb{E}\|\Delta x_{t+1}^c\|^2 = \mathbb{E}\|\Delta x_t^c\|^2 - 2\alpha_t^c\mathbb{E}\langle g_t^{0,c}(x_t^c), \Delta x_t^c\rangle + (\alpha_t^c)^2\mathbb{E}\|g_t^{0,c}(x_t^c)\|^2. \tag{23}$$

We first analyze the first variant (22-1), which is similar to the analysis of COE in Appendix E as it does not involve the asynchronous COE error.

**Lemma F.1** (CDL descent). *Let $\lambda^c := \lambda_{\min}(\text{sym}(\bar{A}^0))$. With (22-1), the cross term in (23) satisfies*

$$\mathbb{E}\langle \Delta x_t^c, g_t^{0,c}(x_t^c)\rangle \geq \lambda^c\mathbb{E}\|\Delta x_t^c\|^2.$$

*Proof.* Similar to the proof of Lemma E.1, the cross term satisfies

$$
\begin{aligned}
\mathbb{E}\langle \Delta x_t^c, g_t^{0,c}(x_t^c)\rangle =& \mathbb{E}_{\mathcal{F}_{t-1}}\left[\left\langle \frac{1}{n}\sum_{i=1}^{n}\mathbb{E}_t^i[A(s)x_t^c - b^i(s)], \Delta x_t^c\right\rangle\right] \\
=& \mathbb{E}_{\mathcal{F}_{t-1}}\left[\langle \bar{A}^0 x_t^c - \bar{b}^0, \Delta x_t^c\rangle\right] \\
=& \mathbb{E}_{\mathcal{F}_{t-1}}\left[\langle (\bar{A}^0 x_t^c - \bar{b}^0) - (\bar{A}^0 x_*^c - \bar{b}^0), \Delta x_t^c\rangle\right] \\
\geq & \lambda_{\min}(\mathrm{sym}(\bar{A}^0))\mathbb{E}\|\Delta x_t^c\|^2.
\end{aligned}
$$

$\square$

**Lemma F.2** (CDL variance). *With (22-1), the variance term in (23) satisfies*

$$
\mathbb{E}\|g_t^{0,c}(x_t^c)\|^2 \leq 2G_A^2 \mathbb{E}\|\Delta x_t^c\|^2 + 2\sigma^2 n^{-1}.
$$

*Proof.* Similar to the proof of Lemma E.2, the variance term can be first decomposed as

$$
\mathbb{E}\|g_t^{0,c}(x_t^c)\|^2 = \mathbb{E}\|\tfrac{1}{n}\sum_{i=1}^{n}A_t^i(x_t^c - x_*^c) + g_t^{0,c}(x_*^c)\|^2 \leq 2G_A^2\mathbb{E}\|\Delta x_t^c\|^2 + 2\mathbb{E}\|g_t^{0,c}(x_*^c)\|^2,
$$

where the second term can be further decomposed as

$$
\begin{aligned}
\mathbb{E}\|g_t^{0,c}(x_*^c)\|^2 =& \frac{1}{n^2}\sum_{i=1}^{n}\mathbb{E}_t\|g_t^i(x_*^c)\|^2 + \frac{1}{n^2}\sum_{i\neq j}\langle \mathbb{E}_t^i g_t^i(x_*^c), \mathbb{E}_t^j g_t^j(x_*^c)\rangle \\
\leq & \frac{1}{n}(G_A G_x + G_b)^2 + \left\|\frac{1}{n}\sum_{i=1}^{n}\mathbb{E}_t^i g_t^i(x_*^c)\right\|^2 \\
\leq & \sigma^2 n^{-1} + 0.
\end{aligned}
$$

$\square$

Combining Lemmas F.1 and F.2 with (23) gives the one-step progress of the first variant of CDL.

**Corollary F.1** (CDL one-step progress). *Let $\alpha_t^c \leq \lambda^c/(4G_A^2)$. Then, for any time step t, (22-1) satisfies*

$$
\mathbb{E}\|\Delta x_{t+1}^c\|_2^2 \leq (1 - \tfrac{3}{2}\alpha_t^c\lambda^c)\mathbb{E}\|\Delta x_t^c\|_2^2 + 2(\alpha_t^c)^2\sigma^2 n^{-1}.
$$

Next, we analyze the second variant (22-2), which involves the asynchronous COE error.

**Lemma F.3** (CDL +COE descent). *With (22-2), the cross term in (23) satisfies*

$$
\mathbb{E}\langle \Delta x_t^c, g_t^{0,c}(x_t^c; \theta_t^c)\rangle \geq \tfrac{7}{8}\lambda^c\mathbb{E}\|\Delta x_t^c\|^2 - 2(\lambda^c)^{-1}\mathbb{E}\|\Delta\theta_t^c\|^2.
$$

*Proof.* The cross term can be further decomposed as

$$
\begin{aligned}
& \mathbb{E}\left\langle \Delta x_t^c, g_t^{0,c}(x_t^c; \theta_t^c)\right\rangle \\
=& \mathbb{E}_{\mathcal{F}_{t-1}}\left\langle \mathbb{E}_t g_t^{0,c}(x_t^c; \theta_t^c), \Delta x_t^c\right\rangle \\
=& \mathbb{E}_{\mathcal{F}_{t-1}}\left\langle \mathbb{E}_t\left[\frac{1}{n}\sum_{i=1}^{n}(g^{c\to i}(x_t^c, \theta_*^c) - \Phi_t^i(\theta_t^c - \theta_*^c))\right], \Delta x_t^c\right\rangle \\
=& \mathbb{E}_{\mathcal{F}_{t-1}}\left\langle \mathbb{E}_{\mu^0}[A(s)]x_t^c - \mathbb{E}_{\mu^0}[b^c(s)], \Delta x_t^c\right\rangle - \mathbb{E}_{\mathcal{F}_{t-1}}\left\langle \mathbb{E}_{\mu^0}[\Phi(s)]\Delta\theta_t^c, \Delta x_t^c\right\rangle \\
=& \mathbb{E}\left\langle \bar{A}^0 x_t^c - \bar{b}^0, \Delta x_t^c\right\rangle - \mathbb{E}\left\langle \bar{\Phi}^0\Delta\theta_t^c, \Delta x_t^c\right\rangle.
\end{aligned} \tag{24}
$$

The first term in (24) follows a descent direction; by the definition of $x_*^c$,

$$
\begin{aligned}
\mathbb{E}\left\langle \bar{A}^0 x_t^c - \bar{b}^0, \Delta x_t^c\right\rangle =& \mathbb{E}\left\langle (\bar{A}^0 x_t^c - \bar{b}^0) - (\bar{A}^0 x_*^c - \bar{b}^0), \Delta x_t^c\right\rangle \\
=& \mathbb{E}\left\langle \bar{A}^0\Delta x_t^c, \Delta x_t^c\right\rangle \\
\geq & \lambda_{\min}(\mathrm{sym}(\bar{A}^0))\mathbb{E}\|\Delta x_t^c\|_2^2.
\end{aligned} \tag{25}
$$

The second term in (24) involves the estimation error from COE; by the Cauchy-Schwarz inequality and Young's inequality,

$$|\mathbb{E}\langle \bar{\Phi}^0 \Delta\theta_t^c, \Delta x_t^c\rangle| \leq \mathbb{E}[\|\Delta\theta_t^c\|\|\Delta x_t^c\|] \leq \frac{\beta_c}{2}\mathbb{E}\|\Delta x_t^c\|^2 + \frac{1}{2\beta_c}\mathbb{E}\|\Delta\theta_t^c\|^2, \tag{26}$$

where $\beta_c > 0$ is a constant to be determined. Plugging (25) and (26) into (24) gives

$$\mathbb{E}\left\langle \Delta x_t^c, g_t^{0,c}(x_t^c)\right\rangle \geq \left(\lambda^c - \frac{\beta_c}{2}\right)\mathbb{E}\|\Delta x_t^c\|^2 - \frac{1}{2\beta_c}\mathbb{E}\|\Delta\theta_t^c\|^2.$$

Setting $\beta_c = \lambda^c/4$ gives the desired result. $\qquad\square$

**Lemma F.4** (CDL +COE variance). *With (22-2), the variance term in (23) satisfies*

$$\mathbb{E}\|g_t^{0,c}(x_t^c;\theta_t^c)\|^2 \leq 2G_A^2\mathbb{E}\|\Delta x_t^c\|^2 + 4\mathbb{E}\|\Delta\theta_t^c\|^2 + 4\sigma^2 n^{-1}.$$

*Proof.* Similar to the proof of Lemma E.2, the variance term can be first decomposed as

$$\mathbb{E}\|g_t^{0,c}(x_t^c;\theta_t^c)\|^2 = \mathbb{E}\|g_t^{0,c}(x_*^c, \theta_*^c) + A_t^0(x_t^c - x_*^c) - \Phi_t^0(\theta_*^c - \theta_*^c)\|^2,$$

where we write $A_t^0 = \frac{1}{n}\sum_{i=1}^n A_t^i$ and $\Phi_t^0 = \frac{1}{n}\sum_{i=1}^n \Phi_t^i$. Therefore,

$$\mathbb{E}\|g_t^{0,c}(x_t^c;\theta_t^c)\|^2 \leq 2G_A^2\mathbb{E}\|\Delta x_t^c\|^2 + 4\mathbb{E}\|\Delta\theta_t^c\|^2 + 4\mathbb{E}\|g_t^{0,c}(x_*^c, \theta_*^c)\|^2.$$

Similarly, for the variance term at the stationary point, we have

$$\begin{aligned}
\mathbb{E}\|g_t^{0,c}(x_*^c, \theta_*^c)\|^2 =& \frac{1}{n^2}\sum_{i=1}^n \mathbb{E}_t\|g_t^{c\to i}(x_*^c, \theta_*^c)\|^2 + \frac{1}{n^2}\sum_{i\neq j}\langle\mathbb{E}_t^i g_t^{c\to i}(x_*^c, \theta_*^c), \mathbb{E}_t^j g_t^{c\to j}(x_*^c, \theta_*^c)\rangle \\
\leq& \sigma^2 n^{-1} + \|\bar{A}^0 x_*^c - \bar{\Phi}^0\theta_*^c\|^2 \\
=& \sigma^2 n^{-1}.
\end{aligned}$$

Plugging this back gives the desired result. $\qquad\square$

Combining Lemmas F.3 and F.4 with (23) gives the one-step progress of the second variant of CDL.

**Corollary F.2** (CDL +COE one-step progress). *Let $\alpha_t^c \leq \lambda^c/(8G_A^2)$. Then, for any time step $t$, (22-2) satisfies*

$$\mathbb{E}\|\Delta x_{t+1}^c\|_2^2 \leq (1 - \tfrac{3}{2}\alpha_t^c\lambda^c)\mathbb{E}\|\Delta x_t^c\|_2^2 + 8\alpha_t^c(\lambda^c)^{-1}\mathbb{E}\|\Delta\theta_t^c\|_2^2 + 4(\alpha_t^c)^2\sigma^2 n^{-1},$$

*where we use the fact that $\lambda^c/G_A \leq 1$, which implies $\alpha_t^c \leq (\lambda^c/G_A)^2/(8\lambda^c) \leq (\lambda^c)^{-1}$, and thus $\alpha_t^c(\lambda^c)^{-1} + (\alpha_t^c)^2 \leq 2\alpha_t^c(\lambda^c)^{-1}$.*

Combining Corollaries E.1 and F.2 and Lemmas D.4 and D.5 gives the convergence guarantee of CDL with asynchronous COE.

**Corollary F.3** (CDL convergence). *With a constant step size $\alpha^c\lambda^c = \alpha^b\lambda^b = \ln t/t$, for any time step $t > 0$, (22) satisfies*

$$\mathbb{E}\|x_t^c - x_*^c\|^2 = \begin{cases} O\left(\dfrac{\sigma^2\ln t}{(\lambda^c)^2 nt}\right) & \text{for (22-1);} \\[3mm] O\left(\dfrac{\sigma^2\ln t}{(\lambda^b\lambda^c)^2 nt}\right) & \text{for (22-2) with (20).} \end{cases}$$

*With a linearly diminishing step size $\alpha_\tau^c\lambda^c = \alpha_\tau^b\lambda^b = 4/(\tau + t_0 + 1)$, $\tau = 0,\ldots,t$, where $t_0 > 0$ ensures that $\alpha_0^b \leq \lambda^c/(8G_A^2)$ and $\alpha_0^b \leq \lambda^b/4$, (22) satisfies*

$$\mathbb{E}\|\tilde{x}_t^c - x_*^c\|^2 = \begin{cases} O\left(\dfrac{\sigma^2}{(\lambda^c)^2 nt}\right) & \text{for (22-1);} \\[3mm] O\left(\dfrac{\sigma^2}{(\lambda^b\lambda^c)^2 nt}\right) & \text{for (22-2) with (20),} \end{cases}$$

*where $\tilde{x}_t$ represents the convex combination specified in Lemma D.5.*

*Proof.* Corollaries E.1 and F.2 fit into Lemma D.4 with $z_t^1 = \theta_t^c$ and $z_t^2 = x_t^c$, along with $\alpha_t = \alpha_t^b \lambda^b = \alpha_t^c \lambda^c$ and

$$C^1 = 2\sigma^2 n^{-1}, \quad C^2 = 4\sigma^2 n^{-1}, \quad C^{2,1} = 8(\lambda^c)^{-1}.$$

Thus,

$$\mathbb{E}\|\Delta x_{t+1}^c\|^2 + 16(\lambda^c)^{-2}\mathbb{E}\|\Delta\theta_{t+1}^c\|^2$$

$$\leq (1 - \alpha_t)\left(\mathbb{E}\|\Delta x_t^c\|^2 + 16(\lambda^c)^{-2}\mathbb{E}\|\Delta\theta_t^c\|^2\right) + \alpha_t^2\left(\frac{2\sigma^2}{(\lambda^c)^2 n} + \frac{64\sigma^2}{(\lambda^c \lambda^b)^2 n}\right)$$

$$\leq (1 - \alpha_t)\left(\mathbb{E}\|\Delta x_t^c\|^2 + 16(\lambda^c)^{-2}\mathbb{E}\|\Delta\theta_t^c\|^2\right) + \alpha_t^2 \frac{66\sigma^2}{(\lambda^b \lambda^c)^2 n},$$

where the last inequality uses the fact that $\lambda^b \leq \|\bar{\Phi}^0\| \leq 1$. Plugging the above Lyapunov function into Lemma D.5 gives the desired results. □

## G  ANALYSIS OF PERSONALIZED COLLABORATIVE LEARNING

This section analyzes the local component of personalized collaborative learning (AffPCL), with environment heterogeneity, asynchronous COE, and asynchronous DRE that satisfies Assumption 1. The learning problem is the most general form in (2):

$$\bar{A}^i x_*^i = \bar{b}^i, \quad \forall i \in [n],$$

where $\bar{A}^i = \mathbb{E}_{\mu^i} A(s)$ and $\bar{b}^i = \mathbb{E}_{\mu^i} b^i(s)$. We restate the local update rule for agent $i$:

$$x_{t+1}^i = x_t^i - \alpha_t \tilde{g}_t^i = x_t^i - \alpha_t(g_t^i(x_t^i) + g_t^{c\rightrightarrows i}(x_t^c; \theta_t^c, \eta_t^i) - g_t^{c\to i}(x_t^c; \theta_t^c)), \tag{27}$$

where

$$g_t^{c\to i}(x) = A(s_t^i)x - \hat{b}_t^c(s_t^i).$$

and

$$g_t^{c\rightrightarrows i}(x) = \frac{1}{n}\sum_{j=1}^{n} \hat{\rho}_t^i(s_t^j) g_t^{c\to j}(x).$$

Recall that $\hat{b}_t^c$ and $\hat{\rho}_t^i$ are estimated objective and density ratio functions at time step $t$; and with linear parametrization, they satisfy

$$\hat{b}_t^c(s) = \Phi(s)\theta_t^c, \quad \hat{\rho}_t^i(s) = \psi(s)^T \eta_t^i.$$

Thus, we highlight the dependence on estimation weights by including $\theta_t^c$ and $\eta_t^i$ in the arguments of update directions. We denote $\Delta x_t^i = x_t^i - x_*^i$.

We first show that the local update rule follows an unbiased direction towards the local solution plus estimation errors.

**Lemma G.1** (Correction). *The expected local update direction satisfies*

$$\mathbb{E}[\tilde{g}_t^i] = \mathbb{E}[g_t^i(x_t^i)] + \mathbb{E}[\mathcal{E}(\Delta\eta_t^i, \Delta x_t^c, \Delta\theta_t^c)],$$

*where*

$$\|\mathbb{E}[\mathcal{E}(\Delta\eta_t^i, \Delta x_t^c, \Delta\theta_t^c)]\| = O(\sigma\|\Delta\eta_t^i\| + G_A\|\Delta x_t^c\| + \|\Delta\theta_t^c\|).$$

*Proof.* We first inspect the importance-corrected term:

$$\mathbb{E}[g_t^{c \rightleftarrows i}(x_t^c; \theta_t^c, \eta_t^i)]$$

$$=\mathbb{E}\left[\frac{1}{n}\sum_{j=1}^{n}\hat{\rho}_t^i(s_t^j)(A(s_t^j)x_t^c - \hat{b}_t^c(s_t^j))\right]$$

$$=\mathbb{E}_{\mathcal{F}_{t-1}}\left[\frac{1}{n}\sum_{j=1}^{n}\mathbb{E}_{\mu^j}\left[\hat{\rho}_t^i(s)(A(s)x_t^c - \hat{b}_t^c(s))\right]\right]$$

$$=\mathbb{E}_{\mathcal{F}_{t-1}}\left[\mathbb{E}_{\mu^0}\left[\hat{\rho}_t^i(s)(A(s)x_t^c - \hat{b}_t^c(s))\right]\right]$$

$$=\mathbb{E}_{\mathcal{F}_{t-1}}\left[\mathbb{E}_{\mu^0}\left[(\rho^i(s) + \psi(s)^T\Delta\eta_t^i)(A(s)x_t^c - \hat{b}_t^c(s))\right]\right]$$

$$=\mathbb{E}_{\mathcal{F}_{t-1}}\mathbb{E}_{\mu^0}\left[\rho^i(s)(A(s)x_t^c - \hat{b}_t^c(s))\right] + \mathbb{E}_{\mathcal{F}_{t-1}}\mathbb{E}_{\mu^0}\left[\underbrace{\psi(s)^T\Delta\eta_t^i(A(s)x_t^c - \Phi(s)\theta_t^c)}_{\mathcal{E}}\right]. \quad (28)$$

We notice the bias correction term exactly removes the bias in the first term above:

$$\mathbb{E}_{\mathcal{F}_{t-1}}\left[\mathbb{E}_{\mu^0}\left[\rho^i(s)(A(s)x_t^c - \hat{b}_t^c(s))\right]\right]$$

$$=\mathbb{E}_{\mathcal{F}_{t-1}}\left[\int_{\mathcal{S}}\frac{\mu^i(s)}{\mu^0(s)}(A(s)x_t^c - \hat{b}_t^c(s))\,\mu^0(s)\mathrm{d}s\right]$$

$$=\mathbb{E}_{\mathcal{F}_{t-1}}\left[\int_{\mathcal{S}}(A(s)x_t^c - \hat{b}_t^c(s))\,\mu^i(s)\mathrm{d}s\right]$$

$$=\mathbb{E}_{\mathcal{F}_{t-1}}\left[\mathbb{E}_{\mu^i}[A(s)x_t^c - \hat{b}_t^c(s)]\right]$$

$$=\mathbb{E}\left[A(s_t^i)x_t^c - \hat{b}_t^c(s_t^i)\right]$$

$$=\mathbb{E}[g_t^{c \rightarrow i}(x_t^c)].$$

The additional $\mathcal{E}$ encompasses all the estimation error:

$$\|\mathcal{E}(\Delta\eta_t^i, \Delta x_t^c, \Delta\theta_t^c; s)\| = \left\|\psi(s)^T\Delta\eta_t^i(A(s)\Delta x_t^c - \Phi(s)\Delta\theta_t^c + A(s)x_*^c - b^c(s))\right\|$$

$$\leq |\hat{\rho}_t^i(s) - \rho^i(s)|(\|A(s)\Delta x_t^c\| + \|\Phi(s)\Delta\theta_t^c\| + \|A(s)x_*^c - b^c(s)\|)$$

$$\leq |\hat{\rho}_t^i(s) - \rho^i(s)|(G_A\|\Delta x_t^c\| + \|\Delta\theta_t^c\| + \sigma)$$

$$\lesssim \sigma\|\Delta\eta_t^i\| + G_A\|\Delta x_t^c\| + \|\Delta\theta_t^c\|,$$

where the last inequality uses Assumption 1 that $|\hat{\rho}_t^i(s) - \rho^i(s)| = O(1)$. Therefore, we have

$$\mathbb{E}[\tilde{g}_t^i] = \mathbb{E}[g_t^i(x_t^i)] + \mathbb{E}[g_t^{c \rightleftarrows i}(x_t^c)] - \mathbb{E}[g_t^{c \rightarrow i}(x_t^c)] = \mathbb{E}[g_t^i(x_t^i)] + \mathbb{E}[\mathcal{E}(\Delta\eta_t^i, \Delta x_t^c, \Delta\theta_t^c; s)].$$

$$\square$$

**Corollary G.1** (AffPCL descent). *Let $\lambda^i := \lambda_{\min}(\mathrm{sym}(\bar{A}^i))$. The expected local update direction satisfies*

$$\mathbb{E}\left\langle \tilde{g}_t^i, \Delta x_t^i \right\rangle \geq \tfrac{7}{8}\lambda^i \mathbb{E}\|\Delta x_t^i\|^2 - 2(\lambda^i)^{-1}\mathbb{E}\|\mathcal{E}\|^2.$$

*Proof.* By Lemma G.1 and Young's inequality,

$$\mathbb{E}\left\langle \tilde{g}_t^i, \Delta x_t^i \right\rangle = \mathbb{E}_{\mathcal{F}_{t-1}}\left\langle \mathbb{E}_{\mu^i}[g_t^i(x_t^i)] + \mathcal{E}, \Delta x_t^i \right\rangle$$

$$=\mathbb{E}_{\mathcal{F}_{t-1}}\left\langle \mathbb{E}_{\mu^i}[A(s)x_t^i - b^i(s)], \Delta x_t^i \right\rangle + \mathbb{E}\left\langle \mathcal{E}, \Delta x_t^i \right\rangle$$

$$=\mathbb{E}_{\mathcal{F}_{t-1}}\left\langle \bar{A}^i x_t^i - \bar{b}^i, \Delta x_t^i \right\rangle + \mathbb{E}\left\langle \mathcal{E}, \Delta x_t^i \right\rangle$$

$$=\mathbb{E}_{\mathcal{F}_{t-1}}\left\langle (\bar{A}^i x_t^i - \bar{b}^i) - (\bar{A}^i x_*^i - \bar{b}^i), \Delta x_t^i \right\rangle + \mathbb{E}\left\langle \mathcal{E}, \Delta x_t^i \right\rangle$$

$$=\mathbb{E}\left\langle \bar{A}^i \Delta x_t^i, \Delta x_t^i \right\rangle + \mathbb{E}\left\langle \mathcal{E}, \Delta x_t^i \right\rangle$$

$$\geq \lambda^i \mathbb{E}\|\Delta x_t^i\|^2 - \frac{1}{2}\cdot\frac{\lambda^i}{4}\mathbb{E}\|\Delta x_t^i\|^2 - \frac{1}{2}\cdot\frac{4}{\lambda^i}\mathbb{E}\|\mathcal{E}\|^2$$

$$=\tfrac{7}{8}\lambda_{\min}(\bar{A}^i)\mathbb{E}\|\Delta x_t^i\|^2 - 2(\lambda^i)^{-1}\mathbb{E}\|\mathcal{E}\|^2.$$

$$\square$$

For the variance, we inspect the importance-corrected aggregated update direction and biased-corrected local update direction separately.

**Lemma G.2** (Federated variance reduction). *The variance of the importance-corrected aggregated update direction satisfies*

$$\mathbb{E}\|g_t^{c \rightleftarrows i}(x_t^c; \theta_t^c, \eta_t^i)\|^2 \leq 24 G_A^2 \mathbb{E}\|\Delta x_t^c\|^2 + 40 \mathbb{E}\|\Delta \theta_t^c\|^2 + 64 \sigma^2 (n^{-1} + 2\delta_{\text{cen}}^i) + 2\mathbb{E}\|\mathcal{E}\|^2.$$

*Proof.* Similar to (28), we decompose the importance-corrected aggregated update direction as the direction that uses the true density ratio plus an estimation error term:

$$\mathbb{E}\|g_t^{c \rightleftarrows i}(x_t^c; \theta_t^c, \eta_t^i)\|^2 \leq 2\mathbb{E}\left\|g_t^{c \rightleftarrows i}(x_t^c; \theta_t^c, \eta_*^i)\right\|^2 + 2\mathbb{E}\|\mathcal{E}\|^2.$$

We can then focus on the direction with true density ratio, which again can be decomposed into local variances and covariances:

$$\mathbb{E}\|g_t^{c \rightleftarrows i}(x_t^c; \theta_t^c, \eta_*^i)\|^2 = \mathbb{E}\left\|\frac{1}{n}\sum_{i=1}^n \rho^i(s_t^i) g_t^{c \to j}(x_t^c)\right\|^2$$

$$= \underbrace{\frac{1}{n^2}\sum_{j=1}^n \mathbb{E}\|\rho^i(s_t^j) g_t^{c \to j}(x_t^c)\|^2}_{H_1} + \underbrace{\frac{1}{n^2}\sum_{j \neq k} \mathbb{E}\langle \rho^i(s_t^j) g_t^{c \to j}(x_t^c), \rho^i(s_t^k) g_t^{c \to k}(x_t^c)\rangle}_{H_2}.$$

Different from federated variance reduction using data sampled from i.i.d. distributions, $H_1$ also depends on how close the agents' heterogeneous environment distributions are. Suppose $\mathcal{F}_{t-1}$-a.s. that $\|g_t^{c \to j}(x_t^c)\| \leq H_3$ for all $j \in [n]$. Conditioned on $\mathcal{F}_{t-1}$, we then have

$$H_1 \leq \frac{H_3^2}{n^2}\sum_{j=1}^n \mathbb{E}_{\mu^j}|\rho^i(s)|^2$$

$$\leq \frac{2H_3^2}{n^2}\left(n + \sum_{j=1}^n \mathbb{E}_{\mu^j}|1 - \rho^i(s)|^2\right)$$

$$\leq \frac{2H_3^2}{n}\left(1 + \mathbb{E}_{\mu^0}\left|1 - \frac{\mu^i(s)}{\mu^0(s)}\right|^2\right)$$

$$\leq \frac{2H_3^2}{n}\left(1 + \chi^2(\mu^i, \mu^0)\right),$$

where $\chi^2$ is the chi-squared divergence. By Lemma D.3, we know that

$$\chi(\mu^i, \mu^0) \leq \max\left\{\|\rho^i\|_\infty, 1\right\} \cdot \|\mu^i - \mu^0\|_{\text{TV}} \leq \max\left\{\|\rho^i\|_\infty, 1\right\} \delta_{\text{env}}^i.$$

We notice that the essential supremum of the density ratio has a natural upper bound:

$$\|\rho^i\|_\infty = \sup_{s \in \mathcal{S}} \frac{\mu^i(s)}{\mu^0(s)} = \sup_{s \in \mathcal{S}} \frac{\mu^i(s)}{\frac{1}{n}\sum_{j=1}^n \mu^j(s)} \leq \sup_{s \in \mathcal{S}} \frac{\mu^i(s)}{\frac{1}{n}\mu^i(s)} = n,$$

where we use the convention that $0/0 = 0$. Combining the above two bounds together gives

$$H_1 \leq 2H_3^2(n^{-1} + \delta_{\text{env}}^i).$$

We now bound $H_3$. Conditioned on $\mathcal{F}_{t-1}$, we have

$$\|g_t^{c \to j}(x_t^c)\| = \|A_t^j(\Delta x_t^c + x_*^c) - \Phi_t^j(\Delta \theta_t^c + \theta_*^c)\| \leq \sigma + G_A\|\Delta x_t^c\| + \|\Delta \theta_t^c\|.$$

Thus, we set $H_3 = \sigma + G_A \mathbb{E}\|\Delta_t^c\| + \mathbb{E}\|\Delta \theta_t^c\|$. Plugging this back gives

$$H_1 \leq 2(2G_A^2 \mathbb{E}\|\Delta x_t^c\|^2 + 4\mathbb{E}\|\Delta \theta_t^c\|^2 + 4\sigma^2)(n^{-1} + \delta_{\text{env}}^i) \tag{29}$$

$$\leq 8G_A^2 \mathbb{E}\|\Delta x_t^c\|^2 + 16\mathbb{E}\|\Delta \theta_t^c\|^2 + 8\sigma^2(n^{-1} + \delta_{\text{env}}^i),$$

where we use the fact that $n^{-1}, \delta_{\text{env}}^i \leq 1$.

The covariances $H_2$ also needs special treatment for heterogeneous environments. Unlike the homogeneous case where the local update directions are *perpendicular* in the sense that their covariances are zero, here the importance correction alters the geometry and requires a more careful anatomy of the covariance terms. Conditioned on $\mathcal{F}_{t-1}$, we have

$$
\begin{aligned}
H_2 &= \frac{1}{n^2} \sum_{j=1}^{n} \left\langle \mathbb{E}_{\mu^j}[\rho^i(s)g^{c \to j}(x_t^c; s)], \sum_{k \neq j} \mathbb{E}_{\mu^k}[\rho^i(s)g^{c \to k}(x_t^c; s)] \right\rangle \\
&= \frac{1}{n^2} \sum_{j=1}^{n} \left\langle \mathbb{E}_{\mu^j}[\rho^i(s)g^{c \to j}(x_t^c; s)], \sum_{k=1}^{n} \mathbb{E}_{\mu^k}[\rho^i(s)g^{c \to k}(x_t^c; s)] - \mathbb{E}_{\mu^j}[\rho^i(s)g^{c \to j}(x_t^c; s)] \right\rangle \\
&= \frac{1}{n^2} \sum_{j=1}^{n} \left\langle \mathbb{E}_{\mu^j}[\rho^i(s)g^{c \to j}(x_t^c; s)], \sum_{k=1}^{n} \mathbb{E}_{\mu^k}[\rho^i(s)g^{c \to k}(x_t^c; s)] \right\rangle \\
&\quad \underbrace{- \frac{1}{n^2} \sum_{j=1}^{n} \left\| \mathbb{E}_{\mu^j}[\rho^i(s)g^{c \to j}(x_t^c; s)] \right\|^2}_{\leq 0} \\
&\leq \frac{1}{n^2} \left\langle \sum_{j=1}^{n} \mathbb{E}_{\mu^j}[\rho^i(s)g^{c \to j}(x_t^c; s)], \sum_{k=1}^{n} \mathbb{E}_{\mu^k}[\rho^i(s)g^{c \to k}(x_t^c; s)] \right\rangle \\
&= \frac{1}{n^2} \left\| \sum_{j=1}^{n} \mathbb{E}_{\mu^j}[\rho^i(s)g^{c \to j}(x_t^c; s)] \right\|^2 \\
&= \frac{1}{n^2} \left\| \sum_{j=1}^{n} \mathbb{E}_{\mu^j}[\rho^i(s)(A(s)x_t^c - \hat{b}_t^c(s))] \right\|^2 \\
&= \left\| \mathbb{E}_{\mu^0}[\rho^i(s)(A(s)x_t^c - \hat{b}_t^c(s))] \right\|^2 \\
&= \left\| \mathbb{E}_{\mu^i}[A(s)x_t^c - \hat{b}_t^c(s)] \right\|^2 \\
&= \left\| \mathbb{E}[g^{c \to i}(x_t^c)] \right\|^2 .
\end{aligned}
$$

Note that the only inequality above omits a term of $O(n^{-1})$, and thus the bound is tight when $n$ is large. Recall that $g^{c \to i}(x_t^c)$ corresponds to the bias in the aggregated update direction. Thus, we show that the covariance reduces nicely to the bias term, further showcasing the power of importance correction. The bias term (conditioned on $\mathcal{F}_{t-1}$) can be further decomposed as

$$
\begin{aligned}
\left\| \mathbb{E}[g^{c \to i}(x_t^c)] \right\|^2 &= \left\| \bar{A}^i(\Delta x_t^c + x_*^c - x_*^i + x_*^i) - \bar{\Phi}^i(\Delta \theta_t^c + \theta_*^c - \theta_*^i + \theta_*^i) \right\|^2 \\
&= \left\| \bar{A}^i(\Delta x_t^c + x_*^c - x_*^i) - \bar{\Phi}^i(\Delta \theta_t^c + \theta_*^c - \theta_*^i) \right\|^2 \\
&\leq 4 \left( G_A^2 \|\Delta x_t^c\|^2 + \|\bar{A}^i(x_*^c - x_*^i)\|^2 + \|\Delta \theta_t^c\|^2 + \|\bar{\Phi}^i(\theta_*^c - \theta_*^i)\|^2 \right) .
\end{aligned}
$$

Plugging in the bounds in Items (e) and (i) in Lemma D.1 gives

$$
\left\| \mathbb{E}[g^{c \to i}(x_t^c)] \right\|^2 \leq 4G_A^2 \|\Delta x_t^c\|^2 + 4\|\Delta \theta_t^c\|^2 + 32\sigma^2 (\delta_{\text{cen}}^i)^2.
$$

Removing the conditioning on $\mathcal{F}_{t-1}$ gives

$$
H_2 \leq 4G_A^2 \mathbb{E}\|\Delta x_t^c\|^2 + 4\mathbb{E}\|\Delta \theta_t^c\|^2 + 32\sigma^2 (\delta_{\text{cen}}^i)^2. \tag{30}
$$

Plugging (29) and (30) back gives the desired result:

$$
\mathbb{E}\|g_t^{c \to i}(x_t^c)\|^2 \leq 24G_A^2 \mathbb{E}\|\Delta x_t^c\|^2 + 40\mathbb{E}\|\Delta \theta_t^c\|^2 + 64\sigma^2(n^{-1} + 2\delta_{\text{cen}}^i) + 2\mathbb{E}\|\mathcal{E}\|^2,
$$

where we use the fact that $\delta_{\text{env}}^i \leq \delta_{\text{cen}}^i \leq 1$. □

We then inspect the variance of the bias-corrected local update direction.

**Lemma G.3** (Affinity-based variance reduction). *The variance of the bias-corrected local update direction satisfies*

$$\mathbb{E}\|g_t^i(x_t) - g_t^{c\to i}(x_t^c; \theta_t^c)\|^2 \leq 4G_A^2\mathbb{E}\|\Delta x_t^i\|^2 + 8G_A^2\mathbb{E}\|\Delta x_t^c\|^2 + 8\mathbb{E}\|\Delta\theta_t^c\|^2 + 16\sigma^2\tilde{\delta}_{\text{cen}}^i,$$

*where $\tilde{\delta}_{\text{cen}}^i = \min\{1, \nu\delta_{\text{cen}}^i\}$.*

*Proof.* Similarly, the variance term can be decomposed as the variance at the optimal solution plus the estimation error:

$$\begin{aligned}
\mathbb{E}\|g_t^i(x_t^i) - g_t^{c\to i}(x_t^c; \theta_t^c)\|^2 &= \mathbb{E}\|A_t^i(x_t^i - x_t^c) - (b^i(s_t^i) - \hat{b}_t^c(s_t^i))\|^2 \\
&= \mathbb{E}\left\|A_t^i(\Delta x_t^i + x_*^i - x_*^c - \Delta x_t^c) - \Phi_t^i(\theta_*^i - \theta_*^c - \Delta\theta_t^c)\right\|^2 \\
&\leq 4G_A^2\mathbb{E}\|\Delta x_t^i\|^2 + 8G_A^2\mathbb{E}\|\Delta x_t^c\|^2 + 8\mathbb{E}\|\Delta\theta_t^c\|^2 \qquad \text{(estimation)} \\
&\quad + 4\mathbb{E}\|A_t^i(x_*^i - x_*^c)\|^2 + 4\mathbb{E}\|\Phi_t^i(\theta_*^i - \theta_*^c)\|^2. \qquad \text{(affinity)}
\end{aligned}$$

For the affinity terms, by Lemma D.2,

$$\max\left\{\mathbb{E}\|A_t^i(x_*^i - x_*^c)\|^2, \mathbb{E}\|\Phi_t^i(\theta_*^i - \theta_*^c)\|^2\right\} \leq 2\sigma^2\tilde{\delta}_{\text{cen}}^i.$$

Combining the above bounds gives the desired result. □

We are now ready to prove the one-step progress of the local update in AffPCL.

**Corollary G.2** (AffPCL one-step progress). *Suppose $\alpha_t^i \leq \lambda^i/(40G_A^2)$. Then, for any time step $t$ and agent $i$, (27) satisfies*

$$\begin{aligned}
\mathbb{E}\|\Delta x_{t+1}^i\|^2 &\leq (1 - \tfrac{3}{2}\alpha_t^i\lambda^i)\mathbb{E}\|\Delta x_t^i\|^2 + 64(\alpha_t^i)^2 G_A^2\mathbb{E}\|\Delta x_t^c\|^2 + 96(\alpha_t^i)^2\mathbb{E}\|\Delta\theta_t^c\|^2 \\
&\quad + 4\alpha_t^i(\lambda^i)^{-1}\mathbb{E}\|\mathcal{E}\|^2 + 144(\alpha_t^i)^2\sigma^2(n^{-1} + 2\tilde{\delta}_{\text{cen}}^i) \\
&\leq (1 - \tfrac{3}{2}\alpha_t^i\lambda^i)\mathbb{E}\|\Delta x_t^i\|^2 \\
&\quad + (64(\alpha_t^i)^2 G_A^2 + 16\alpha_t^i(\lambda^i)^{-1}G_\rho^2 G_A^2)\mathbb{E}\|\Delta x_t^c\|^2 \\
&\quad + (96(\alpha_t^i)^2 + 16\alpha_t^i(\lambda^i)^{-1}G_\rho^2)\mathbb{E}\|\Delta\theta_t^c\|^2 \\
&\quad + 4\alpha_t^i(\lambda^i)^{-1}\sigma^2\mathbb{E}\|\Delta\eta_t^i\|^2 \\
&\quad + 144(\alpha_t^i)^2\sigma^2(n^{-1} + 2\tilde{\delta}_{\text{cen}}^i).
\end{aligned}$$

*Proof.* Combining Lemmas G.2 and G.3 gives

$$\begin{aligned}
\mathbb{E}\|\tilde{g}_t^i\|^2 &\leq 2(\mathbb{E}\|g_t^{c\rightleftarrows i}(x_t^c; \theta_t^c, \eta_t^i)\|^2 + \mathbb{E}\|g_t^i(x_t^i) - g_t^{c\to i}(x_t^c; \theta_t^c)\|^2) \\
&\leq 8G_A^2\mathbb{E}\|\Delta x_t^i\|^2 + 64G_A^2\mathbb{E}\|\Delta x_t^c\|^2 + 96\mathbb{E}\|\Delta\theta_t^c\|^2 + 144\sigma^2(n^{-1} + 2\tilde{\delta}_{\text{cen}}^i) + 4\mathbb{E}\|\mathcal{E}\|^2.
\end{aligned}$$

Combining the above bound with Corollary G.1 gives

$$\begin{aligned}
\mathbb{E}\|\Delta x_{t+1}^i\|^2 &= \mathbb{E}\|\Delta x_t^i\|^2 - 2\alpha_t\mathbb{E}\langle\tilde{g}_t^i, \Delta x_t^i\rangle + \alpha_t^2\mathbb{E}\|\tilde{g}_t^i\|^2 \\
&\leq \mathbb{E}\|\Delta x_t^i\|^2 - \tfrac{7}{4}\alpha_t^i\lambda^i\mathbb{E}\|\Delta x_t^i\|^2 + 2\alpha_t^i(\lambda^i)^{-1}\mathbb{E}\|\mathcal{E}\|^2 + 4(\alpha_t^i)^2\mathbb{E}\|\mathcal{E}\|^2 \\
&\quad + 8(\alpha_t^i)^2(G_A^2\mathbb{E}\|\Delta x_t^i\|^2 + 8G_A^2\mathbb{E}\|\Delta x_t^c\|^2 + 12\mathbb{E}\|\Delta\theta_t^c\|^2 + 18\sigma^2(n^{-1} + 2\tilde{\delta}_{\text{cen}}^i)) \\
&\leq (1 - \tfrac{7}{4}\alpha_t^i\lambda^i + 8(\alpha_t^i)^2 G_A^2)\mathbb{E}\|\Delta x_t^i\|^2 + 64(\alpha_t^i)^2 G_A^2\mathbb{E}\|\Delta x_t^c\|^2 + 96(\alpha_t^i)^2\mathbb{E}\|\Delta\theta_t^c\|^2 \\
&\quad + 2\alpha_t^i((\lambda^i)^{-1} + 2\alpha_t^i)\mathbb{E}\|\mathcal{E}\|^2 + 144(\alpha_t^i)^2\sigma^2(n^{-1} + 2\tilde{\delta}_{\text{cen}}^i).
\end{aligned}$$

Setting $\alpha_t^i \leq \lambda^i/(32G_A^2)$, which implies $2\alpha_t^i \leq (\lambda^i)^{-1}$, gives

$$\begin{aligned}
\mathbb{E}\|\Delta x_{t+1}^i\|^2 &\leq (1 - \tfrac{3}{2}\alpha_t^i\lambda^i)\mathbb{E}\|\Delta x_t^i\|^2 + 64(\alpha_t^i)^2 G_A^2\mathbb{E}\|\Delta x_t^c\|^2 + 96(\alpha_t^i)^2\mathbb{E}\|\Delta\theta_t^c\|^2 \\
&\quad + 4\alpha_t^i(\lambda^i)^{-1}\mathbb{E}\|\mathcal{E}\|^2 + 144(\alpha_t^i)^2\sigma^2(n^{-1} + 2\tilde{\delta}_{\text{cen}}^i).
\end{aligned}$$

Finally, we expand $\mathbb{E}\|\mathcal{E}\|^2$. By Assumption 1,

$$\begin{aligned}
\mathbb{E}\|\mathcal{E}(\Delta\eta_t^i, \Delta x_t^c, \Delta\theta_t^c)\|^2 &= \mathbb{E}\left\|\frac{1}{n}\sum_{i=1}^n \psi_t^i\Delta\eta_t^i(A_t^i\Delta x_t^c - \Phi_t^i\Delta_t^c + A_t^i x_*^c - \Phi_t^i\theta_*^c)\right\|^2 \\
&\leq 2\sigma^2\mathbb{E}\|\Delta\eta_t^i\|^2 + 4(G_A^2\mathbb{E}\|\Delta x_t^c\|^2 + \mathbb{E}\|\Delta\theta_t^c\|^2).
\end{aligned}$$

Plugging it back gives the desired result. □

## G.1 PROOF OF THEOREM 1

Invoking Lemmas D.4 and D.5 with Corollaries E.1, F.2 and G.2 gives us the main result. We restate a more general version of Theorem 1.

**Theorem 1.** *We synchronize the step sizes across all learning modules by setting $\alpha_t = \alpha_t^i \lambda^i = \alpha_t^b \lambda^b = \alpha_t^c \lambda^c$. Then, with a constant step size $\alpha_\tau \equiv \ln t/(\lambda t)$, $\tau = 0, \ldots, t$, AffPCL with various learning modules satisfies*

$$\mathbb{E}\|x_t^i - x_*^i\|^2 = O\left(\frac{\sigma^2 \ln t}{(\lambda^i)^2 t} \cdot \delta\right),$$

*where*

$$
\delta = \begin{cases}
\max\{n^{-1}, \tilde{\delta}_{\text{cen}}^i\} & \text{for (27) + (22-1) + (20);} \\
\tilde{\delta}_{\text{cen}}^i + \max\left\{1, \frac{\lambda^i}{\lambda^b \lambda^c}\right\}^2 \cdot n^{-1} & \text{for (27) + (22-2) + (20);} \\
\frac{\sigma^2}{(\lambda^\rho)^2} \cdot \tilde{\delta}_{\text{cen}}^i + \frac{\sigma^2}{(\min\{\lambda^b, \lambda^c, \lambda^\rho\})^2} \cdot n^{-1} & \text{for (27) + (22-1) + (20) + (13);} \\
\frac{\sigma^2}{(\lambda^\rho)^2} \cdot \tilde{\delta}_{\text{cen}}^i + \max\left\{\frac{\sigma}{\min\{\lambda^b, \lambda^c, \lambda^\rho\}}, \frac{\lambda^i}{\lambda^b \lambda^c}\right\}^2 \cdot n^{-1} & \text{for (27) + (22-2) + (20) + (13).}
\end{cases}
$$

*where $\tilde{\delta}_{\text{cen}} = \min\{1, \nu \delta_{\text{cen}}\}$.*

Specifically, we highlight that AffPCL with access to the true density ratio (i.e., without (13)) achieves

$$\mathbb{E}\|x_t^i - x_*^i\|^2 = \widetilde{O}((\kappa^i)^2 t^{-1} \cdot \max\{n^{-1}, \tilde{\delta}_{\text{cen}}^i\}),$$

where $\kappa = \sigma/\lambda^i$ is the agent-specific condition number, which further recovers Theorem 1 in the main text by noting that $\tilde{\delta}_{\text{env}}^i \leq \delta_{\text{env}}$, $\delta_{\text{cen}}^i \leq \delta_{\text{env}} + \delta_{\text{obj}}$, and $\kappa^i \leq \kappa$.

On the other hand, AffPCL with DRE has a worst-case complexity bounded by

$$\mathbb{E}\|x_t^i - x_*^i\|^2 = O\left((\kappa^i \kappa^\rho)^2 t^{-1} \cdot \max\{\nu^\rho n^{-1}, \tilde{\delta}_{\text{cen}}^i\}\right),$$

where $\kappa^\rho := \sigma/\lambda^\rho$ and $\nu^\rho = \max\{\frac{\lambda^\rho}{\min\{\lambda^b, \lambda^c\}}, \frac{\lambda^i \lambda^\rho}{\sigma \lambda^b \lambda^c}\}^2$, which now depends on the conditioning of DRE.

*Proof.* We only prove the first and last cases, as the other two cases follow similarly. For the last case, similar to the proof of Corollary F.3, Corollaries E.1, F.2 and G.2 fit into Lemma D.4 with $z_t^1 = \theta_t^c$, $z_t^2 = x_t^c$, $z_t^3 = \eta_t^i$, and $z_t^4 = x_t^i$, along with

$$C^1 \asymp C^2 = O(\sigma^2 n^{-1}), \quad C^3 \asymp C^4 = O(\sigma^2(n^{-1} + \tilde{\delta}_{\text{cen}}^i))$$
$$C^{2,1} \asymp O((\lambda^c)^{-1}), \quad C^{3,1} = C^{3,2} = 0, \quad C^{4,1} \asymp C^{4,2} \asymp C^{4,3} = O((\lambda^i)^{-1} \sigma^2).$$

Then, the corresponding weights in Lemma D.4 are

$$
\begin{aligned}
w^4 &= 1, \\
w^3 &= 2C^{4,3}(\lambda^i)^{-1} = O(\sigma^2(\lambda^i)^{-2}), \\
w^2 &= 2C^{4,2}(\lambda^i)^{-1} + 0 = O(\sigma^2(\lambda^i)^{-2}), \\
w^1 &= 2C^{4,1}(\lambda^i)^{-1} + 2C^{2,1}(\lambda^c)^{-1} = O(\sigma^2(\lambda^i)^{-2} + (\lambda^c)^{-2}).
\end{aligned}
$$

Thus, the overall MSE with a constant step size $\ln t/t$ is

$$
\begin{aligned}
\mathbb{E}\|x_t^i - x_*^i\|^2 =& O\Bigg(\frac{\sigma^2 \ln t}{t}\Bigg(\Bigg(\frac{1}{(\lambda^i)^2} + \frac{\sigma^2}{(\lambda^i \lambda^\rho)^2}\Bigg) \cdot (n^{-1} + \tilde{\delta}_{\text{cen}}^i) \\
&+ \Bigg(\frac{\sigma^2}{(\lambda^i \lambda^c)^2} + \frac{\sigma^2}{(\lambda^i \lambda^b)^2} + \frac{1}{(\lambda^b \lambda^c)^2}\Bigg) \cdot n^{-1}\Bigg)\Bigg) \\
=& O\Bigg(\frac{\sigma^2 \ln t}{(\lambda^i)^2 t}\Bigg(\frac{\sigma^2}{(\lambda^\rho)^2} \cdot \tilde{\delta}_{\text{cen}}^i + \Bigg(\frac{\sigma^2}{(\min\{\lambda^\rho, \lambda^b, \lambda^c\})^2} + \frac{(\lambda^i)^2}{(\lambda^b \lambda^c)^2}\Bigg) \cdot n^{-1}\Bigg)\Bigg).
\end{aligned}
$$

For the first case without DRE, $\mathcal{E} = 0$. Thus, Corollaries E.1, F.1 and G.2 fit into Lemma D.4 with $z_t^1 = \theta_t^c$, $z_t^2 = x_t^c$, and $z_t^3 = x_t^i$, along with

$$C^1 \asymp C^2 = O(\sigma^2 n^{-1}), \quad C^3 = O(\sigma^2(n^{-1} + \tilde{\delta}_{\text{cen}}^i))$$
$$C^{2,1} = 0, \quad C^{3,1} \asymp C^{3,2} = O(\alpha_0(\lambda^i)^{-1}\sigma^2).$$

Then, the corresponding weights in Lemma D.4 are

$$w^3 = 1,$$
$$w^2 = 2C^{3,2}(\lambda^i)^{-1} = O(\alpha_0\sigma^2(\lambda^i)^{-2}),$$
$$w^1 = 2C^{3,1}(\lambda^i)^{-1} + 0 = O(\alpha_0\sigma^2(\lambda^i)^{-2}).$$

Thus, the overall MSE with a constant step size $\ln t / t$ is

$$\mathbb{E}\|x_t^i - x_*^i\|^2 = O\left(\frac{\sigma^2 \ln t}{(\lambda^i)^2 t}\left((n^{-1} + \tilde{\delta}_{\text{cen}}^i) + \frac{\alpha_0\sigma^2}{(\min\{\lambda^b, \lambda^c\})^2} \cdot n^{-1}\right)\right)$$
$$= O\left(\frac{\sigma^2 \ln t}{(\lambda^i)^2 t} \cdot \max\{n^{-1}, \tilde{\delta}_{\text{cen}}^i\}\right).$$

Similarly, by using a linearly diminishing step size and a convex combination of the iterates, we can remove the logarithmic factor in the numerator. $\square$

