# OpenReview forum: "Personalized Collaborative Learning with Affinity-Based Variance Reduction"
_ICLR.cc/2026/Conference — ICLR 2026 Poster_

### Official Review · Reviewer_AoD3 · 2025-10-25

**Soundness:** 2
**Presentation:** 4
**Contribution:** 2
**Rating:** 4
**Confidence:** 4

**Summary:**

This paper proposes an affinity-based variance reduction approach that adaptively interpolates between federated learning and independent learning according to the level of objective-function heterogeneity (or data-distribution heterogeneity) among agents. Theoretically, when the data distributions across agents are homogeneous (i.e., all agents share the same objective function and the local optimal solution coincides with the global optimal solution), the proposed approach achieves the same convergence rate as federated learning. In contrast, when the data distributions are heterogeneous, the proposed approach converges to each agent's local optimal solution with a convergence rate similar to that of independent learning (where agents train in isolation). Numerical simulations validate the effectiveness of the proposed approach.

My main concerns lie in: i) overly idealized assumptions (see Weakness 1 and 2); ii) extremely heavy communication burden (see Weakness 3); iii) lack of empirical validation on benchmark datasets (see Weakness 4); iv) the reliance on a DRE oracle contradicts the "adaptivity without prior knowledge" claim (see Problem 1).
While the theoretical analysis is sound under the given assumptions, the absence of real-world evaluations and restricted applicability of the proposed approach limit the contribution of the paper.

**Strengths:**

The paper is clearly presented, and the theoretical results are well explained. The personalized collaborative learning problem studied in this work is also an active research topic.

**Weaknesses:**

1. **Limited applicability:** The authors assume that all agents share a common and fixed feature extractor (i.e., $\bar{A}$; see the first formula in Section 3). This assumption is unrealistic in many real-world applications. For example, in end-to-end deep learning, both the backbone and task-specific heads are typically trained jointly. Therefore, the proposed approach is only applicable to "frozen feature + personalized head" settings, which limits its practical applicability.

2. **Overly simplified problem setting:** The linear system in Eq. (2) essentially corresponds to the optimization problem $\min F(x)=\frac{1}{n}\sum_{i=1}^{n}f_{i}(x)$, where each local objective function $f_{i}(x)=||x^{\top}\bar{A}x-(\bar{b}^{i})^{\top}x||^2$ is strongly convex and even quadratic. This formulation is considerably simpler than those used in most existing studies on federated learning or personalized collaborative learning, which typically consider convex or nonconvex objective functions.

3. **Heavy communication overhead:** According to Algorithm 1 in the Appendix (which provides a summarized version of the proposed approach presented in the main text), each agent is required to send $s_{t}^{i}$, $g_{t}^{i}(x_{t}^{c})$, $g_{t}^{i,b}(\theta_{t}^{c})$, $g_{t}^{c\rightarrow i}(x_{t}^{c})$ to the centralized server, while the server requires to broadcast $g_{t}^{0,c}(x_{t}^{c})$, $g_{t}^{0,b}(\theta_{t}^{c})$, $g_{t}^{c\rightrightarrows i}(x_{t}^{c})$ to all agents. All these shared variables are high-dimensional, as they depend on the dimensionality of the underlying data in machine learning applications.
Moreover, this bidirectional communication must occur at every iteration, which leads to an extremely heavy communication overhead. This contradicts the core motivation of FL, where communication is typically much more expensive than local computation.

4. **Weakness on empirical evaluation:** The experiments are performed only on synthetic linear systems, without any evaluation on real-world datasets (e.g., FeMNIST, Shakespeare). Moreover, there lacks comparison with existing personalized FL approaches such as pFedMe, Ditto, or Clustered FL.

**Questions:**

See the Weaknesses section above. In addition, I have a few additional questions:

1. The paper repeatedly states that the proposed approach "does not require prior knowledge" and "automatically adapts to unknown heterogeneity." However, the algorithm relies on a density ratio term. Although the authors mention using a density-ratio estimator when this ratio is unknown, Theorem 2 shows that estimating it is nontrivial and inevitably introduces error. Therefore, the claim of "no prior knowledge required" is overstated unless the authors either (i) prove that the estimation is unbiased or (ii) analyze convergence in the presence of this estimated error.

2. The definition of personalization in this paper differs from that in mainstream studies, as it emphasizes only local optimality while ignoring global generalization. In fact, personalized models must balance adaptation to local data with maintaining global generalization. The paper does not discuss this point, which is an omission. Please clarify this point.

3. Some notation is unclear (e.g., in Equation 1, "t" denotes the iteration, not sample size).

---

> ### Author Response · Authors · 2025-11-24
> **Response to Reviewer AoD3**
>
> Dear Reviewer AoD3,
>
> Thank you for your detailed and thoughtful review and for giving us the opportunity to address your concerns. We would first like to clarify the motivation and scope of our personalized collaborative learning (PCL) setup before addressing each of your specific weaknesses and questions.
>
> ## PCL vs. FL
>
> As detailed in Section 1, motivating and formalizing the PCL framework is one of the main contributions of our work.
> Although PCL shares the same centralized collaboration scheme as FL, its motivation is nearly the **opposite** of that in traditional FL: agents in PCL seek fully personalized solutions rather than a single unified one.
> In other words, we identify a key pitfall of FL, namely, the unified solution often fails to serve heterogeneous agents well, and propose the PCL problem in response.
> We highlight three defining features of PCL in the third paragraph: full personalization, collaboration benefits, and adaptive learning.
> While PCL and FL share some similarities, the two frameworks have fundamentally different motivations and emphases.
>
> Given this intricate relationship, we fully understand that the reviewer may examine our work from an FL perspective and have some concerns about the absence of certain FL characteristics.
> However, because PCL is a distinct framework, several motivations central to FL, e.g., global generalization or communication efficiency, do not directly apply to our setting.
> Moreover, as a first work in this novel framework, we choose to focus on establishing the core **theoretical foundations** of PCL.
> A stylized setting allows us to highlight the key ideas, and naturally, this means we have to leave for future work certain practical considerations and extensions one would expect works in an established framework like FL to address.
> As noted in Section 8, these include communication complexity and general (non-)convex optimization problems.

---

> ### Author Response · Authors · 2025-11-24
> **Response to Reviewer AoD3 [Continued]**
>
> We now respond to each of your specific weaknesses and questions.
>
> > **W1** The assumption that all agents share a common and fixed feature extractor limits its practical applicability.
>
> We thank the reviewer for bringing up this point and providing an example where heterogeneous feature extractors are expected.
> We fully agree that such scenarios exist. However, we respectfully disagree that assuming a shared feature extractor is unrealistic.
> In fact, our setup is directly motivated by several important real-world applications in which a common feature extractor is standard practice:
>
> 1. Parameter-efficient fine-tuning (e.g., LoRA) for foundation models, where a large pre-trained model is shared among all users and only the final layers are fine-tuned for personalization.
> 2. Collaborative reinforcement learning with shared state perception, where agents share a common state representation module (e.g., identical sensors) but have different reward functions and policies. See also Appendix C.6 for more discussion.
> 3. Classical user-item matrix factorization for recommendation systems, where the item embeddings are shared among all users while each user has a personalized preference vector.
> 4. User calibration in edge computing, where all edge devices share the same signal-processing front end but require personalized response models.
>
> We would be happy to provide more relevant examples or references if the reviewer is interested.
>
> We recognize that extending our framework to heterogeneous feature extractors is one of the important future directions, and have added it to the list in Section 8.
> A straightforward approach is to learn personalized feature extractors $A^{i}$. There is no fundamental difference between this task and Eq. (2), so our method can be directly applied, and one can expect that the complexity depends on the affinity between features.
> One potential issue is the dimensionality of $A^{i}$, which may be large when the system dimension $d$ is large.
> In such cases, learning personalized features *implicitly* in a model-free manner is a nontrivial challenge to be addressed in future work.

---

> ### Author Response · Authors · 2025-11-24
> **Response to Reviewer AoD3 [Continued]**
>
> > **W2** The linear system formulation is considerably simpler than convex or nonconvex objective functions typically considered in most existing studies.
>
> We fully agree with the reviewer that extending our framework to general (non-)convex optimization problems is an important future direction, as listed in Section 8.
>
> We would first like to clarify that our method is based on stochastic approximation for fixed-point problems, which is fundamentally different from gradient descent for optimization problems. Therefore, it's not accurate to interpret our linear system as arising from a quadratic objective function.
> For example, as discussed in Appendix C.6, temporal difference learning in reinforcement learning is a stochastic approximation method for solving a linear system, but the temporal difference error is not the gradient of any objective function.
>
> As discussed in the section "PCL vs. FL" above, our work presents a first step in studying PCL, while most existing works build on the mature FL framework.
> As highlighted in the paper, solving linear systems in the PCL setting already poses multiple novel and nontrivial challenges that do not arise in FL.
> For example, to tackle objective heterogeneity in PCL, we need to develop a new bias-correction technique to achieve affinity-based variance reduction.
> This bias correction is fundamentally different from those used in the heterogeneous FL literature, as discussed in Paragraph "Bias correction" in Section 3.
> For another example, to tackle environmental heterogeneity in PCL, we incorporate a novel importance correction on the server side, which has not been studied in FL.
> Additionally, importance sampling is known to potentially incur larger variance, but we manage to show that our importance correction mechanism delivers effective variance reduction in PCL.
>
> These challenges are unique to the PCL setting, and our solutions differ significantly from existing FL techniques.
> We believe it is important to first establish the core theoretical foundations of PCL in a stylized setting before extending the framework to general objectives.
> For these reasons, we respectfully argue that comparing the scope of our work directly to that of existing studies in the established FL framework would be too harsh and not reflective of the novelty of the PCL setting.
>
> We now comment on potential approaches to extend our framework to general non-linear/(non-)convex objectives and highlight potential challenges.
> First, consider a spacial class of non-linear systems $\bar{A}\_{x^{i}\_{\ast}}x^{i}\_{\ast}=\bar{b}^{i}$, where the nonlinearity comes from $x$-dependent $\bar{A}\_{x}$. If at the fixed point $x^i_{\ast}$ the system matrix $\bar{A}\_{x^{i}\_{\ast}} \succeq \lambda I$ and the variation of $\bar{A}\_{x}$ is small enough such that $\sup_{x,y}\\|\bar{A}\_{x}-\bar{A}\_{y}\\| \le \lambda$, then we expect our method still works with minor modifications to the analysis.
> This is because the dynamics are dominated by $\bar{A}_{x^{i}\_{\ast}}$, and thus behave similarly to a linear system near the fixed point.
> For general non-linear systems, fixed points may be non-unique or may not exist, and further investigation is needed.
>
> For optimization problems $\min f_{i}(x)$, there are two major challenges. First, the gradient information cannot be composed additively as in Eq. (5) or (8), which is enabled by the linearity of the system. This reflects a more fundamental question: what is the right "central-global decomposition" discussed in Section 3 for optimization problems?
> The second challenge under the optimization setting is that objective and environmental heterogeneity may no longer be cleanly separable as in the linear case.
> In addition, the geometry of the optimization landscape is also shaped by higher-order information beyond gradients. For example, we may need to characterize and analyze the Hessian heterogeneity across agents.
>
> More challenges arise when the objectives are not strongly convex.
> Under strong convexity, local gradients quickly pull each agent's iterate towards its local optimum, and the variance around the optima dominates the error. Thus, it suffices to analyze the stochastic systems at the agents' optima.
> For general convex or non-convex objectives, we also need to analyze the optimization dynamics along agents' distinct trajectories, which is substantially more intricate and difficult to control.

---

> ### Author Response · Authors · 2025-11-24
> **Response to Reviewer AoD3 [Continued]**
>
> > **W3** Heavy communication overhead contradicts the core motivation of FL.
>
> We thank the reviewer for raising this point.
> We have added Remark 1 in Appendix C.1 to analyze the communication overhead of `AffPCL` in detail.
> As discussed in the section "PCL vs. FL" above, PCL does not have the same set of motivations as FL.
> While minimizing communication is indeed a primary goal of FL, enabling effective **collaboration among highly heterogeneous agents** is another.
> Intuitively, in a highly heterogeneous setting, agents inevitably need to exchange more information to correctly account for their differences and achieve full personalization.
> Our work explores the trade-offs of collaboration and personalization at this frontier.
> In summary, FL methods are more suited for near-homogeneous settings or when a global generalizable model is desired, while `AffPCL` works better in highly heterogeneous settings where full personalization is desired.
>
> We agree that communication efficiency is important for practical deployment.
> In Remark 1, we note that this overhead can be substantially reduced by incorporating standard communication-efficiency techniques, thus striking a balance between personalization and communication cost.

---

> ### Author Response · Authors · 2025-11-24
> **Response to Reviewer AoD3 [Continued]**
>
> > **W4** The experiments lack evaluation on real-world datasets and lack comparison with existing personalized FL approaches.
>
> We thank the reviewer for suggesting additional real-world datasets and baselines. We have added all the requested personalized FL baselines to our experiments, including **`pFedMe`, `Ditto`, and Clustered FL**.
> We have also added experiments on the suggested **real-world `FEMNIST` dataset**.
> Section 7 and Appendix B have been updated accordingly to include these new comparisons. We kindly refer the reviewer to our general response `R2` and `R3` for a summary of the updated experimental results.

---

> ### Author Response · Authors · 2025-11-24
> **Response to Reviewer AoD3 [Continued]**
>
> > **Q1** For density ratio estimation, the authors need to (i) prove that the estimation is unbiased or (ii) analyze convergence in the presence of this estimated error.
>
> We thank the reviewer for these valuable suggestions.
> Due to the page limit, we defer the detailed discussion of DRE to Appendix C.4 and G.
> Specifically, Appendix C.4 presents **three practical unbiased DRE methods** and Appendix G provides the sample complexity analysis of `AffPCL` **in the presence of estimation error** from an asynchronous DRE module (see Theorem 1 in Appendix G).
>
> We summarize the three unbiased DRE methods in Appendix C.4 below. First, we show that the DRE problem can be cast as a variant of Eq. (2), and thus our method can be directly applied to learn density ratios in an online and unbiased manner. Second, Example 1 shows that when the difference between distributions is sparse, online $\ell_0$-constrained or $\ell_1$-regularized least squares can be used to solve DRE unbiasedly. Third, Example 2 shows that when the distributions are coupled, a simple online fixed-point iteration can be used to solve DRE unbiasedly.
>
> Notably, when additional affinity structures among distributions are present, the lower bound in Theorem 2 does not preclude effective affinity-based variance reduction for `AffPCL` when using DRE methods in Examples 1 and 2.

---

> ### Author Response · Authors · 2025-11-24
> **Response to Reviewer AoD3 [Continued]**
>
> > **Q2** The definition of personalization in this paper differs from that in mainstream studies, as it emphasizes only local optimality while ignoring global generalization.
>
> We assume the reviewer is referring to the mainstream personalized FL literature, where partial personalization is often sought to balance local adaptation and global generalization.
> In contrast, our work studies a novel PCL setup, which is fundamentally different from the FL setting, as discussed in the "PCL vs. FL" section above.
> In PCL, agents operate in distinct environments with distinct objectives, and collaborate only when doing so benefits each agent’s own goal.
> As a result, "global generalization" is not defined in the PCL setup, as there is no single global objective that any real agent aims to generalize to.
> The seeming deviation from mainstream FL studies proves the novelty of our PCL framework.
> Further, as we discuss throughout, this novelty is nontrivial and brings unique challenges. This may also help explain why fully personalized yet collaborative methods have seen limited attention in the FL literature.
> We hope the above discussion clarifies the distinct motivations and emphases of PCL compared with FL.
>
> That said, `AffPCL` does have a central decision learning (CDL) module, which learns a solution to a global model, and is shared among all agents; see Section 5.1 for details.
> This module is precisely the FL procedure reviewed in Section 2 and therefore learns a unified solution that one may interpret as having desirable "global generalization" properties in the FL sense.
> Each agent's fully personalized solution is the combination of this global solution and its own local adaptation; see Paragraph "Central-local decomposition" in Section 3 for details.
> Therefore, for users who do care about global generalization in the traditional FL sense, our method naturally provides it via the CDL module as a by-product.

---

> ### Author Response · Authors · 2025-11-24
> **Response to Reviewer AoD3 [Continued]**
>
> > **Q3** Some notation is unclear (e.g., in Equation 1, "t" denotes the iteration, not sample size).
>
> In our setup, each agent collects one sample per iteration, so $t$ is also the sample size for each agent. Please let us know if there is any other unclear notation.
>
> We thank the reviewer again for the detailed review and hope our responses clarify the contributions of our work. We would be happy to engage in further discussions if any queries remain.

---

### Official Review · Reviewer_vFEm · 2025-10-29

**Soundness:** 3
**Presentation:** 3
**Contribution:** 3
**Rating:** 6
**Confidence:** 3

**Summary:**

This paper introduces a personalized collaborative learning (PCL) framework tailored for heterogeneous agents, addressing both environmental and objective heterogeneity. The proposed method, AffPCL, is theoretically shown to reduce sample complexity by a factor of max(n^{-1},\sigma) and analysis shows agent may obtain linear speeedup with arbitraily disimilar agents. The simulation results shows improvement over compared methods.

**Strengths:**

- The topic of personalized collaborative learning for heterogeneous agents is relevant and important. The formulation of PCL is a meaningful contribution,
- The proposed framework and algorithm are clearly described and well-motivated. The formulation is broad enough to cover multiple domains such as supervised learning, reinforcement learning, and statistical decision-making.
- Simulation results convincingly demonstrate the advantages of AffPCL, particularly under high heterogeneity settings.

**Weaknesses:**

- While Theorem 2 provides a lower bound, the paper would be stronger if it integrated and analyzed a practical DRE method within the AffPCL framework.
- The quantification of objective heterogeneity is not clearly defined, which may hinder its application in real-world scenarios.
- The current experimental evaluation is limited to synthetic linear systems. Including experiments on real-world benchmarks would significantly strengthen the empirical validity of the method.
- Although the theoretical framework is applicable to reinforcement learning, the settings are not covered in the experiments.
- The framework considers asynchronous importance estimation, but this aspect is not evaluated or discussed in the simulation results.
- The proposed update rule incurs higher communication overhead compared to standard federated averaging, which may limit its scalability in certain practical settings.

**Questions:**

Please see the weakness part.

---

> ### Author Response · Authors · 2025-11-24
> **Response to Reviewer vFEm**
>
> Dear Reviewer vFEm,
>
> We appreciate your constructive feedback and would like to address each of your specific concerns in detail.
>
> > **W1** The paper would be stronger if it integrated and analyzed a practical DRE method within the `AffPCL` framework.
>
> We thank the reviewer for this valuable suggestion.
> Due to the page limit, we defer the detailed discussion of DRE to Appendix C.4 and G.
> In particular, Appendix G provides the sample complexity analysis of `AffPCL` equipped with an asynchronous DRE module (see Theorem 1 in Appendix G).
> Appendix C.4 presents **three practical DRE methods** that can be integrated into our framework.
> First, Appendix C.4 shows that the DRE problem can be cast as a variant of Eq. (2), allowing our method to be directly applied to learn density ratios in an online manner.
> Second, Example 1 shows that when the difference between distributions is sparse, online $\ell_0$-constrained or $\ell_1$-regularized least squares can be used to solve DRE with a sample complexity reduction going beyond Theorem 2.
> Third, Example 2 shows that when the distributions are coupled, a simple online fixed-point iteration can be used to solve DRE with a sample complexity reduction going beyond Theorem 2.
> Notably, with additional affinity structures among distributions, the lower bound in Theorem 2 does not preclude effective affinity-based variance reduction for `AffPCL` when using DRE methods in Examples 1 and 2.

---

> ### Author Response · Authors · 2025-11-24
> **Response to Reviewer vFEm [Continued]**
>
> > **W2** The quantification of objective heterogeneity is not clearly defined, which may hinder its application in real-world scenarios.
>
> Objective heterogeneity is clearly and mathematically defined in Definition 1 as the magnitude of the relative difference between agents’ objective functions.
> We argue that this definition is flexible and well suited to a wide range of scenarios.
> For example, in supervised learning, it captures differences in label distributions across agents, and in reinforcement learning, it captures differences in reward functions.
> In the latter case, this definition of *reward heterogeneity* has been widely adopted in the federated RL literature; see, e.g., cited Zhang et al. (2024) and Wang et al. (2024).

---

> ### Author Response · Authors · 2025-11-24
> **Response to Reviewer vFEm [Continued]**
>
> > **W3** Including experiments on real-world benchmarks would significantly strengthen the empirical validity of the method.
>
> We thank the reviewer for this suggestion. We have added experiments on the real-world `FEMNIST` dataset in Section 7 with detailed setup provided in Appendix B. We kindly refer the reviewer to our general response `R3` for a summary of the updated experimental results.

---

> ### Author Response · Authors · 2025-11-24
> **Response to Reviewer vFEm [Continued]**
>
> > **W6** The proposed update rule incurs higher communication overhead compared to standard federated averaging, which may limit its scalability in certain practical settings.
>
> We thank the reviewer for bringing up this practical consideration.
> We have added Remark 1 in Appendix C.1 to transparently analyze the communication overhead of `AffPCL` compared to `FedAvg` and thus clarify its scalability.
> The overhead is incurred by the components necessary for achieving full personalization, illustrating the tension between personalization and collaboration.
> We also note in the discussion that for practical applications, this overhead can be effectively managed by integrating well-established techniques from federated learning, thereby improving scalability.

---

> ### Comment · Reviewer_vFEm · 2025-11-26
>
> Thank you for the authors’ thorough response. I have carefully read the discussion above and have no additional concerns about the current response. I would be grateful if the authors could address my remaining question.

---

> ### Author Response · Authors · 2025-11-26
> **Response to Reviewer vFEm [Continued]**
>
> Thank you for the prompt reply! We are glad our responses helped clarify your concerns. We now address your remaining question.
>
> > **W4&W5** The reinforcement learning settings are not covered in the experiments.
> > Asynchronous importance estimation is not evaluated or discussed in the simulation results.
>
> Thank you for suggesting new experiments. We have implemented the `AffPCL` version of `SARSA`, a fundamental temporal difference policy optimization algorithm, as well as asynchronous density ratio estimation (DRE) under this reinforcement learning setting.
> The results have been added to Section 7, with detailed derivation and setup provided in Appendix C.6 and B.
> We have also updated the supplementary material to include the code implementation.
> Please refer to our general response `R4` for a summary of the updated experimental findings.
>
> Notebaly, our RL experiments follow the standard heterogeneous federated RL setup in cited Zhang et al. (2024),
> and our implementation of `AffPCL`-`SARSA` is as straightforward as in the other settings, yet it delivers significant performance gains. This showcases the practicality, robustness, and versatility of our framework.

---

### Official Review · Reviewer_iE7v · 2025-11-06

**Soundness:** 3
**Presentation:** 3
**Contribution:** 3
**Rating:** 6
**Confidence:** 3

**Summary:**

This paper introduces Personalized Collaborative Learning (PCL), a unified framework for learning across heterogeneous agents that strikes a balance between personalization and inter-agent collaboration. The core algorithm, Affinity-based PCL (AffPCL), dynamically interpolates between independent and collaborative updates based on inter-agent affinity, leveraging both bias correction and importance correction mechanisms. The authors provide thorough convergence guarantees under this framework, showing that collaborative training never hurts and yields linear speedup under high affinity. Application scope spans supervised learning, reinforcement learning, and general decision-making tasks, positioning PCL as a general approach for multi-agent systems.

**Strengths:**

The paper presents a principled and flexible learning framework that smoothly interpolates between personalized and collaborative training, depending on the agent's similarity. By using affinity-adjusted optimization, AffPCL intelligently adapts collaboration strength instead of relying on static weighting schemes. The theoretical results are rigorous, incorporating valid assumptions and proofs that explain the model’s behavior across a range of heterogeneous settings. The clarity of exposition and strong conceptual coherence contribute to the paper’s accessibility and impact.

**Weaknesses:**

The limited scope of empirical evaluation undermines the practical validity of these claims. The experiments are confined to small-scale synthetic setups and do not compare AffPCL with more diverse or state-of-the-art baselines (e.g., pFedMe, Ditto, SCAFFOLD), despite some of these being mentioned in the related work. The computational and communication overhead of AffPCL, particularly in large-scale federated deployments that require density ratio estimation and per-sample corrections, remains unclear. The method’s reliance on accurately estimating density ratios or heterogeneity measures may also restrict applicability.

**Questions:**

How does the computational and communication complexity of AffPCL scale when deployed in realistic settings with hundreds or thousands of agents? An analysis of per-iteration runtime or messaging overhead relative to methods like FedAvg would help clarify scalability.

Can the authors discuss or evaluate AffPCL on larger or more realistic datasets, beyond small-scale synthetic setups? This would help confirm whether the benefits observed in controlled environments persist under real-world data heterogeneity.

The method relies on estimating density ratios or affinity between agents. How might practitioners estimate these measures in practical federated learning scenarios where distributions are unknown or evolving?

---

> ### Author Response · Authors · 2025-11-24
> **Response to Reviewer iE7v**
>
> Dear Reviewer iE7v,
>
> Thank you for your thoughtful review and positive feedback on our work. Below, we provide detailed responses to your questions.
>
> > **W1&Q2** The experiments are confined to small-scale synthetic setups and do not compare AffPCL with more diverse or state-of-the-art baselines.
>
> We appreciate the reviewer’s suggestion to strengthen the empirical evaluation. We have added comparisons with all the requested state-of-the-art baselines, including **`pFedMe` and `Ditto`**, and experiments on the **real-world `FEMNIST` dataset**.
> We have also implemented `SCAFFOLD` (please see the supplementary material). However, `SCAFFOLD` is designed to mitigate client drift in FL with multiple local updates. Since our setting does not involve multiple local updates, `SCAFFOLD` has the same performance as `FedAvg` in our experiments, and we therefore omit its results for clarity.
> We have replotted Figure 1 to include the newly added baselines, added Figure 2 for the real-world experiments, and updated the detailed experimental setup in Appendix B.
> We kindly refer the reviewer to our general response `R2` and `R3` for a summary of the updated experimental results.

---

> ### Author Response · Authors · 2025-11-24
> **Response to Reviewer iE7v [Continued]**
>
> > **W2&Q1** The computational and communication overhead of `AffPCL` remains unclear.
> > An analysis of per-iteration runtime or messaging overhead relative to methods like `FedAvg` would help clarify scalability.
>
> We thank the reviewer for this excellent suggestion.
> We have added a detailed discussion in Remark 1 in Appendix C.1, which contains an analysis of `AffPCL`'s per-iteration runtime and messaging overhead relative to `FedAvg`. Our analysis quantifies the overhead, showing that the asynchronous learning modules and additional personalization mechanisms in `AffPCL` result in a constant-factor increase in both communication and computation complexity.
> This increase reflects the inherent trade-off between personalization and collaboration in heterogeneous settings.
> Furthermore, we discuss how our framework's scalability can be improved in practice by incorporating standard communication-saving techniques, such as multiple local updates and update compression.

---

> ### Author Response · Authors · 2025-11-24
> **Response to Reviewer iE7v [Continued]**
>
> > **W3&Q3** The method relies on estimating density ratios or affinity between agents. How might practitioners estimate these measures in practical federated learning scenarios where distributions are unknown or evolving?
>
> We first clarify that our method does not estimate affinity between agents. The heterogeneity level $\delta$ is introduced solely for presenting our analysis and results.
> Our method adaptively achieves affinity-based variance reduction without requiring knowledge of the affinity levels among agents.
>
> We thank the reviewer for attending to the important question regarding density ratio estimation (DRE).
> Due to the page limit, we have to relegate the detailed discussion of DRE to Appendix C.4 and G, where we
>
> 1. present **three online DRE methods**, which can learn density ratios between unknown distributions and can adapt to evolving distributions;
> 2. show that two of these methods can go beyond the limitations imposed by Theorem 2 to deliver effective sample complexity reduction when additional affinity structures among distributions are present; and
> 3. provide sample guarantees for `AffPCL` with an asynchronous DRE module.
>
> Specifically, most compatible with our setup, Appendix C.4 shows that the DRE problem can be cast as a variant of Eq. (2), and thus our method can be directly applied to learn density ratios in an online manner.
> Example 1 shows that when the difference between distributions is sparse, online $\ell_0$-constrained or $\ell_1$-regularized least squares can be used to solve DRE with a sample complexity reduction.
> Example 2 shows that when the distributions are coupled, DRE can be solved using a simple online fixed-point iteration with a sample complexity reduction.
> Since DRE is a plug-and-play module in our framework, practitioners may choose alternative DRE methods that best suit their application; see, e.g., cited Sugiyama et al. (2012).

---

### Official Review · Reviewer_QeMG · 2025-11-07

**Soundness:** 2
**Presentation:** 3
**Contribution:** 3
**Rating:** 4
**Confidence:** 3

**Summary:**

This paper introduces Personalized Collaborative Learning (PCL) for federated settings with heterogeneous agents, where each agent has unique data and goals but collaborates via a central server. The proposed algorithm, AffPCL, uses affinity-based variance reduction to balance personalization and collaboration so that agents benefit from similar peers while avoiding degradation from dissimilar ones. It achieves finite-sample convergence with a mean-squared error of order $O(t^{-1} \\max\\{n^{-1}, \\delta\\})$, adapting to agent similarity without needing prior clustering. Experiments show AffPCL outperforms independent training and matches or exceeds FedAvg across varying heterogeneity levels. Key contributions include the PCL formulation, AffPCL algorithm, convergence guarantees, and empirical validation.

**Strengths:**

1. This paper is clearly written, guiding the reader from the simplest FL algorithm to fully personalized FL with variance reduction method. The authors clearly motivate why this is needed (e.g., common global models can be suboptimal under heterogeneity).

2. The proposed algorithm *AffPCL* is conceptually simple yet effective. Each agent’s update is adjusted via principled bias correction and importance weighting to account for differences in objectives and data distributions.

**Weaknesses:**

1. The finite-sample bounds presented in this paper appear quite loose. For instance, Prop. 1 provides an error bound for the vanilla heterogeneous FL algorithm of $\\tilde{O}(\\kappa^2 t^{-1}n^{-1})$, while Prop. 2 states the error bound for AffPCL as $\\tilde{O}(\\kappa^2 t^{-1}\\max\\{n^{-1}, \\delta_{obj}\\})$, with $\\delta_{obj} \in [0,1]$ representing the effect of objective heterogeneity. Since $\\max\\{n^{-1}, \\delta_{obj}\\} \geq n^{-1}$, the bound in Prop. 2 is no tighter than Prop. 1. The same argument applies to the bound in Thm 1.

2. The experiments compare AffPCL *only* against FedAvg (which provides a single global model) and independent learning (no sharing). These are the very foundational algorithms without considering personalized collaborative learning, and the paper does not include any comparisons to existing personalized federated learning methods. There is a rich literature on approaches like per-user fine-tuning, model interpolation, or clustered federated learning.

**Questions:**

1. In the AffPCL algorithm, each agent must perform bias-corrected local updates and compute importance weights for others’ data contributions. How does the computational or communication overhead from this AffPCL algorithm affect its complexity?

2. I find the authors’ argument in Lines 240–249 unclear. Could you clarify how the affinity-based variance reduction specifically benefits the proposed AffPCL algorithm?

---

> ### Author Response · Authors · 2025-11-24
> **Response to Reviewer QeMG**
>
> Dear Reviewer QeMG,
>
> We sincerely appreciate your thoughtful comments and for giving us the opportunity to clarify our contributions. We would like to offer the following clarifications.
>
> > **W1&Q2** The bound in Prop. 2 and Thm. 1 are no tighter than Prop. 1. Could you clarify how the affinity-based variance reduction specifically benefits the proposed `AffPCL` algorithm?
>
> We thank the reviewer for this question, which allows us to clarify a crucial point about our results. The reviewer correctly observes that $\max\\{n^{−1},\delta\\} \ge n^{−1}$, but we respectfully clarify that the bounds in Prop. 1 and Prop. 2/Thm. 1 apply to two fundamentally different problems.
>
> Prop. 1 addresses the standard federated learning (FL) problem of finding a single unified solution.
> Its linear speedup ($n^{-1}$) is the **statistically maximally possible** speedup when you have information from $n$ agents, similar to variance reduction with $n$ i.i.d. samples.
> This result serves as a **lower bound** for any collaborative learning method.
>
> In contrast, `AffPCL` seeks fully personalized solutions for each agent.
> When agents are homogeneous ($\delta=0$), Prop. 2 and Thm. 1 show that `AffPCL` achieves the same optimal linear speedup ($n^{-1}$) as FL does, which cannot be improved upon.
>
> When agents are heterogeneous, a linear speedup is **information-theoretically impossible** for personalized solutions, as one agent cannot fully utilize the information from other agents due to heterogeneity.
> The appropriate baseline for this task is independent learning, which has a sample complexity of ${O}(t^{-1})$, which is tight without collaboration. The result in Prop.2 and Thm. 1, ${O}(t^{-1} \cdot \max\\{n^{-1}, \delta\\})$, demonstrates a **affinity-based speedup over this baseline** since $\max\\{n^{-1}, \delta\\} \le 1$.
>
> In summary, the complexities ${O}(t^{-1}n^{-1})$ of FL and ${O}(t^{-1})$ of independent learning are both **tight** for their respective settings: agents collaboratively learn a unified solution vs. agents independently learn without collaboration.
> `AffPCL` adaptively interpolates between these two extremes, delivering fully personalized solutions while achieving the "best-of-both-worlds": enjoying optimal speedup when possible and ensuring no performance degradation otherwise.
> In the intermediate regime, the speedup of `AffPCL` over independent learning depends on how similar the agents are, and thus is referred to as affinity-based variance reduction.

---

> ### Author Response · Authors · 2025-11-24
> **Response to Reviewer QeMG [Continued]**
>
> > **W2** The paper does not include any comparisons to existing personalized federated learning methods.
>
> We thank the reviewer for suggesting additional baselines. We have added all the requested personalized FL baselines to our experiments, including **per-user fine-tuning, model interpolation (regularization), and clustered federated learning**. Section 7 and Appendix B have been updated accordingly to include these new comparisons.
> We kindly refer the reviewer to our general response `R2` for a summary of the updated experimental results.

---

> ### Author Response · Authors · 2025-11-24
> **Response to Reviewer QeMG [Continued]**
>
> > **Q1** How does the computational or communication overhead from this `AffPCL` algorithm affect its complexity?
>
> We thank the reviewer for this question.
> We have added a detailed analysis of `AffPCL`'s communication and computation complexity in Remark 1 in Appendix C.1.
> In this analysis, we show that the personalization components of `AffPCL` lead to a constant-factor increase in communication and computation complexity compared to `FedAvg`.
> This increase reflects the inherent tension between personalization and collaboration in heterogeneous settings, as agents need to exchange more information to correctly account for their differences and achieve full personalization.
> We also note that our framework is fully compatible with standard techniques (e.g., multiple local updates) that reduce communication complexity in practice.

---

### Author Response · Authors · 2025-11-24
**General Response**

We would like to thank the reviewers for their valuable and insightful comments on our work.
We are glad that the reviewers found
our paper to be clearly written (Reviewers `QeMG`, `iE7v`, `vFEm`, `AoD3`),
our personalized collaborative learning (PCL) setup to be well-motivated (Reviewers `QeMG`, `vFEm`, `AoD3`),
our theoretical results to be sound and rigorous (Reviewers `iE7v`, `vFEm`, `AoD3`),
and our method to be principled and effective (Reviewers `QeMG`, `iE7v`).

We are also grateful for the suggestions for improvement. In what follows, we summarize the revisions we have made to the paper in response to the main comments. The modifications are highlighted in ${\color{orange}\text{orange}}$ in the revised manuscript. We have also **uploaded our implementation** code for the experiments.

> **R1**.  Communication and computation complexity analysis.

We have added a detailed complexity analysis in Remark 1 in Appendix C.1 "Implementation details" to clarify the scalability of our method.
Our analysis quantifies the per-iteration complexity of `AffPCL` relative to `FedAvg`, showing that the additional components needed for personalization and affinity-based variance reduction lead to a constant-factor increase in overhead.
This increase reflects an inherent tension between personalization and collaboration in heterogeneous settings.
Different from FL, PCL seeks fully personalized solutions in a model-free manner, so agents inevitably need to exchange more information to correctly account for their differences.

For practical deployments, we agree that efficiency is critical. As we now state in the appendix, our framework is fully compatible with standard communication-saving techniques like multiple local updates and update compression. Integrating these methods is a natural and important next step left for future work. In this work, we deliberately chose a stylized setting without these optimizations to cleanly isolate and analyze our core theoretical contribution: the affinity-based variance reduction. This approach allows us to transparently show the performance gains from our novel method, disentangled from the well-understood effects of communication-saving protocols.

> **R2**. More baseline methods in experiments.

We have added comparisons with additional baselines in our experiments, including **fine-tuning, regularized (`pFedMe` and `Ditto`), and clustered FL methods**.
Specifically, we have replotted Figure 1 to include these new baselines included and added the detailed setup in Appendix B.
We have increased the heterogeneity levels in our experiments from (0,0.05,0.2,0.5) to (0,0.05,0.3,0.8) to better illustrate the advantages of `AffPCL` under medium-to-high heterogeneity.

Below we summarize the key observations from the updated results in Figure 1.
First, `pFedMe` and `Ditto` exhibit nearly identical performance in our setup, which is expected since both use similar regularization mechanisms to balance personalization and collaboration (and we use the same regularization strength for both methods).
Both regularized and clustered FL methods provide only *partial* personalization (pFL), and thus their performance always lies between `FedAvg` and independent learning (IL).
In particular, in the homogeneous and low-heterogeneity regimes, $\mathrm{MSE}_{\mathrm{FedAvg}} < \mathrm{MSE}\_{\text{pFL}} \le \mathrm{MSE}\_{\mathrm{IL}}$; whereas in the medium- and high-heterogeneity regimes, $\mathrm{MSE}\_{\mathrm{IL}} \le \mathrm{MSE}\_{\text{pFL}} < \mathrm{MSE}\_{\mathrm{FedAvg}}$.
Across all heterogeneity levels, `AffPCL` consistently achieves the lowest MSE among all methods, and strictly outperforms every baseline in the intermediate heterogeneity regimes.
Finally, we note that fine-tuning has the same convergence plateau as independent learning, which aligns with theory: the effect of the initialization vanishes faster than the variance term.

> **R3**. Experiments on real-world datasets.

We have added experiments on the **real-world `FEMNIST` dataset** in Section 7 with detailed setup provided in Appendix B.
In this experiment, we introduce objective heterogeneity in a realistic setting, and train 10 users under four heterogeneity levels.
Consistent with our other experiments, `AffPCL` achieves the lowest test MSE across all heterogeneity levels, while the relative performance of other baselines varies with the heterogeneity level.

We hope our responses and revisions address the comments raised. We would be happy to engage in further discussions if any queries remain.

---

> ### Author Response · Authors · 2025-11-26
> **General Response [Continued]**
>
> > **R4**. Reinforcement learning experiments.
>
> We have implemented the `AffPCL` version of `SARSA`, a fundamental temporal difference policy optimization algorithm, and added the results to Section 7 with detailed derivation provided in Appendix C.6 and setup provided in Appendix B.
> We would like to highlight that `SARSA` solves the **non-linaer** policy optimization problem, showcasing the versatility of our framework beyond linear systems.
> In this experiment, we also test the incorporation of **asynchronous density ratio estimation (DRE)** within `AffPCL` in a RL setting.
> Consistent with our other experiments, `AffPCL` achieves the lowest MSE with respect to the optimal Q-value functions across all heterogeneity levels.
> Additionally, incorporating asynchronous DRE does not hinder the performance of `AffPCL`, demonstrating the practical viability of our framework with online DRE modules.

---

### Meta-Review · Area_Chair_8cFc · 2026-01-06

**Summary:**

The paper develops a learning-theory framing of personalized collaborative learning, where each agent targets its own personalized objective but can still benefit from collaboration through a central server. In the paper’s main setting, agents solve linear fixed-point problems under both objective heterogeneity and environment heterogeneity. The proposed method aggregates information at the server in a gradient-averaging style and introduces agent-specific correction terms to reduce the variance of the collaborative signal. The current development is limited to linear systems and linear stochastic approximation, which makes the theory crisp but also narrows the direct reach to more general non-linear or end-to-end learning settings.

A helpful way to situate the technical idea is that bias correction is not new in federated learning; for example, methods like SCAFFOLD use control variates to correct client drift while still optimizing a global objective. This paper uses correction in almost the opposite direction: it is designed to preserve full personalization and only borrow what is beneficial from collaboration. The problem formulation appears to be a crucial step toward developing a theory of personalized collaborative learning. The main remaining uncertainty is the practical impact.

**Reviewer Concerns:**

Several reviewers focused on the empirical side. The original submission did not sufficiently demonstrate that the method is competitive beyond simple baselines, and it did not provide a concrete analysis of the overhead associated with the extra bias correction. The revision addresses much of this by adding comparisons against common federated baselines. The paper also now includes a more explicit communication and computation accounting for the proposed update protocol.

The main concern that remains is the extent to which the most general version of the method is implementable and convincingly validated end-to-end. In addition, while the real-data experiment is a welcome step, it does not yet provide the same breadth of baseline comparisons as the synthetic section, which leaves the practical advantage over strong personalized baselines less settled than the theory and exposition would suggest.

**Reviewer Scores:**

Reviewer QeMG is likely to move from 4 to 6 after the added baselines and clarifications. Reviewers iE7v and vFEm likely stay at 6. Reviewer AoD3 could plausibly move from 4 to 6, though some broader practicality concerns remain.

---

### Decision · Program_Chairs · 2026-01-26

Accept (Poster)